# Generalization Performance of Hypergraph Neural Networks

## Abstract

Hypergraph neural networks have been promising tools for handling learning tasks involving higher-order data, with notable applications in web graphs, such as modeling multi-way hyperlink structures and complex user interactions. Yet, their generalization abilities in theory are less clear to us. In this paper, we seek to develop margin-based generalization bounds for four representative classes of hypergraph neural networks, including convolutional-based methods (UniGCN), set-based aggregation (AllDeepSets), invariant and equivariant transformations (M-IGN), and tensor-based approaches (T-MPHN). Through the PAC-Bayes framework, our results reveal the manner in which hypergraph structure and spectral norms of the learned weights can affect the generalization bounds, where the key technical challenge lies in developing new perturbation analysis for hypergraph neural networks, which offers a rigorous understanding of how variations in the model's weights and hypergraph structure impact its generalization behavior. Our empirical study examines the relationship between the practical performance and theoretical bounds of the models over synthetic and real-world datasets. One of our primary observations is the strong correlation between the theoretical bounds and empirical loss, with statistically significant consistency in most cases.

## Keywords

Graph Classification; Hypergraph Neural Networks; Learning Theory.

**ACM Reference Format:**

Anonymous Author(s). 2024. Generalization Performance of Hypergraph Neural Networks. In . ACM, New York, NY, USA, 29 pages. https://doi.org/10.1145/nnnnnnn.nnnnnnn

## 1 Introduction

The web represents a vast, interconnected system comprising various types of graphs, such as those formed by web pages [9], social networks [19], and hyperlink networks [34]. Analyzing such web graphs is crucial for tasks like search engine ranking [55], hyperlink prediction [70], community detection [72], and user behavior analysis [33]. Graph learning algorithms, such as Graph Neural Networks (GNNs), have proven powerful in a variety of real-world applications [2]. However, traditional GNNs are inherently limited to modeling pairwise relationships. Hypergraph Neural Networks (HyperGNNs) extend GNNs by modeling higher-order relationships

through hyperedges that capture complex multi-way interactions [72], making them more suitable for web-based applications.

In recent years, several advanced HyperGNN architectures have been developed, including HGNN [22], HyperSAGE [3], K-GNN [45], KP-GNN [21], and T-MPNN [59]. While empirical studies have demonstrated the strong performance of these HyperGNNs, rigorous theoretical analysis is necessary for gaining deeper insights into these models. A well-researched focus is on examining their expressiveness power, typically assessing the ability to distinguish between hypergraph structures or realize certain functions [4, 21, 22, 63]. However, the expressive power of HyperGNNs does not necessarily inform their generalization ability. To date, our understanding of the generalization performance of HyperGNNs is still limited, which is the gap we aim to fill in this work.

In this paper, we seek to provide the very first theoretical evidence of the generalization performance of HyperGNNs for hypergraph classification. Through the PAC-Bayes framework, we examine four representative HyperGNN structures: UniGCN [28], AllDeepSets [10], M-IGN [28, 46], and T-MPHN [59]. These models were selected for their unique architectural approaches: UniGCN employs convolutional-based methods, AllDeepSets utilizes set-based aggregation techniques, M-IGN incorporates invariant and equivariant transformations, and T-MPHN leverages tensor-based operations. While HyperGNNs often generalize GNNs in an immediate manner, techniques of PAC-Bayes for GNNs [32, 38, 56] cannot be directly applied in that feature aggregations in HyperGNNs must be performed over large and heterogeneous sets of nodes, leading to challenges in developing perturbation analysis. Consequently, new analytical techniques are required to accommodate the aggregation mechanisms inherent in HyperGNNs.

Building on the theoretical work, we conduct an empirical study to assess the consistency between theoretical bounds and empirical performance of HyperGNNs. Unlike previous studies that focus on numerical comparisons of generalization bounds, we aim to directly evaluate the alignment between theoretical bounds and model performance, examining how well these bounds explain HyperGNNs' behavior. Since the obtained theoretical results represent upper bounds on generalization performance, this investigation focuses on validating these findings and exploring whether they can offer practical guidance for improving HyperGNNs' performance.

The contributions can be summarized as follows.

- We develop a refined analysis on obtaining the perturbation bound for HyperGNNs by decomposing the output variation into two essential quantities: a) the upper bounds on the maximum node representation and b) the maximum variation of the layer's output caused by the perturbed weights.
- We derive generalization bounds that demonstrate the correlation between the model generalization capacity and several key model attributes, such as the spectral norm of the parameters, the maximum hyperedge size, and the maximum size of the hyperedges that share the same node.

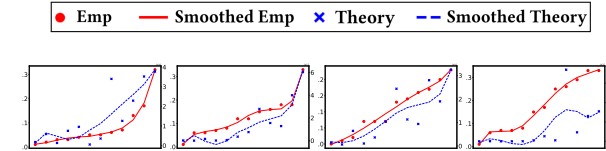

Figure 1: Consistency between empirical loss (Emp) and theoretical bounds (Theory). Each subgraph shows the empirical loss, theoretical bound, and their curves via the Savitzky-Golay filter [53] of 12 groups of datasets with UniGCN in different layers.

- We conduct a detailed analysis of the empirical loss and theoretical generalization bounds on datasets with varying hypergraph structures. Our results show a consistent positive correlation between the empirical loss and theoretical bounds, as depicted in Figure 1. Additionally, we observe that training significantly enhances this alignment. We further explore how different hypergraph structures impact both empirical performance and theoretical bounds.

**Organization.** Sec 2 introduces the related works. The preliminaries are provided in Sec 3. In Sec 4, we present the theoretical results. The empirical studies are given in Sec 5. Further discussions on technical proofs and additional details on the experiments can be found in the appendix. To support reproducibility, the source code and a subset of data are located in an anonymous repository[1].

## 2 Related Works

**HyperGNNs.** The existing HyperGNNs can be categorized into four classes:

- **HyperGCNs.** HyperGCNs bridge the gap between traditional GNNs and hypergraph structures by leveraging hypergraph Laplacians, enabling the application of well-established GCN techniques to capture higher-order interactions [24, 28, 65]. Bai et al. [5] proposed a HyperGCN model that learns from hypergraph structure and edge features. Feng et al. [22] uses a Chebyshev polynomial to approximate the hypergraph Laplacian, leveraging the spectral properties of hypergraphs. Additionally, Yadati et al. [65] proposed an efficient technique to approximate hypergraph Laplacians by focusing on clique expansion, reducing computational complexity while preserving the essential structure of the hypergraph.

- **HyperMPNNs.** HyperMPNNs are significant for their ability to directly model complex dependencies in hypergraphs through message-passing mechanisms, providing a flexible framework to capture multi-way node relationships that are not easily represented by simple graphs [28]. One popular example includes HyperSAGE, which employs a two-layer strategy combining both hyperedge-level and node-level aggregation [3]. Attention-based variants of HyperMPNNs use attention weights to prioritize messages from different nodes or hyperedges [5, 71]. Structure-based HyperMPNNs integrate structural features of hypergraphs directly into the model's embeddings [8, 21, 30, 49].

- **HyperGINs.** HyperGINs are distinguished by their strong expressive power, achieved by extending the concept of Graph Isomorphism Networks (GINs) [62] to hypergraphs and utilizing multiset functions, which preserves invariance or equivariance to input transformations [26, 41]. For instance, [46] combines the $k$-Weisfeiler-Lehman test with GINs to develop $k$-GNN, further boosting expressive power. However, despite their inherent expressiveness, the generalization performance of these models remains unclear and needs further investigation.

- **Tensor-based HyperGNNs.** These models leverage tensor operations that provide a structured and effective means of capturing the complexity of hypergraph interactions [14, 51, 60]. Gao et al. [25] introduced a tensor representation that allows dynamic adjustments of hypergraph components during learning. Building on this, Wang et al. [59] advanced the approach by encoding hypergraph structures using adjacency tensors and cross-node interaction tensors through T-product operations, enabling richer and more expressive data representations.

**Theoretical aspects.** The primary theoretical focus has been on the expressive ability of HyperGNNs. Inspired by the relationship between MPNNs and the 1-Weisfeiler-Leman (1-WL) test [62], a natural approach to designing more expressive HyperGNNs is to simulate higher-order WL tests [26, 28, 47]. One line of research aims to develop a unified framework to enhance the expressive power of HyperGNNs. For example, AllSets framework [10] extends the scope of existing HyperGNNs by employing two multiset functions, covering models like HCHA [5], HNHN [16], HyperSAGE [3], and HyperGCNs [65]. Additionally, motivated by the success of Subgraph GNNs [12], several studies have explored the structural generalization capacity of higher-order GNNs [40, 52, 69], demonstrating that certain HyperGNNs can generalize across graphs of varying sizes after being trained on a limited set of graphs.

**Generalization performance on GNNs.** Several studies have developed generalization bounds using classical statistical learning frameworks, such as Vapnik–Chervonenkis (VC) dimension and Rademacher complexity [20, 31, 54, 58]. Another approach leverages kernel learning techniques via the Neural Tangent Kernel (NTK) [6, 17], where the idea is to approximate a neural network using a kernel derived from its training dynamics. Recent research has focused on deriving norm-based bounds using the PAC-Bayes framework [38]. Although these methods have been effective for analyzing standard GNNs, they are not directly applicable to HyperGNNs due to the complex higher-order interactions and non-linear dependencies inherent in hypergraph structures.

## 3 Preliminaries

### 3.1 Hypergraphs

A hypergraph $\mathcal{G} = (\mathcal{V}, \mathcal{E})$ is given by a set $\mathcal{V} = \{v_1, v_2, ..., v_N\}$ of $N \in \mathbb{Z}^+$ nodes and a set $\mathcal{E} = \{e_1, e_2, ..., e_K\}$ of $K \in \mathbb{Z}^+$ hyperedges, where each hyperedge is a nonempty subset of $\mathcal{V}$. We use $M := \max_{k \in [K]} |e_k|$ to denote the maximum cardinality of hyperedges. For each node $v_i$, its neighbor set $\mathcal{N}_i$ consists of the nodes that share at least one common hyperedge with $v_i$, i.e., $\mathcal{N}_i = \{v_j \in \mathcal{V} \setminus \{v_i\}| \exists e \in \mathcal{E}, \{v_i, v_j\} \subseteq e\}$; the degree of node $v_i$ is

---

[1]https://doi.org/10.5281/zenodo.13906016

defined as $|\mathcal{N}_i|$, and let the maximum node degree be $D := \max_i |\mathcal{N}_i|$. We use $r_i = \{e \in \mathcal{E} | v_i \in e\}$ to denote the incident hyperedge set of node $v_i$; the maximum cardinality of incident hyperedge sets is denoted by $R := \max_i |r_i|$. The input hypergraph is associated with a node feature matrix $\mathbf{X} \in \mathbb{R}^{N \times d}$ and an edge feature matrix $\mathbf{Z} \in \mathbb{R}^{K \times d}$ with $d \in \mathbb{Z}^+$, where $\mathbf{X}[i,:]$ (resp., $\mathbf{Z}[k,:]$) denotes the feature associated with node $v_i$ (resp., hyperedge $e_k$). Following the common practice [48], we assume that $\|\mathbf{X}[i:]\|_2 \leq B^2$ and $\|\mathbf{Z}[k:]\|_2 \leq B^2$ for some constant $B \in \mathbb{R}^+$, where $\|\cdot\|_2$ denotes the $l_2$ norm. Furthermore, $\|\cdot\|$, $\|\cdot\|_F$, and $\|\cdot\|_\infty$ denote the spectral norm, Frobenius norm, and infinite norm for matrices, respectively. For the convenience of readers, a summary of notations is provided in Table 3 in the Appendix. The following operation will be frequently used for describing HyperGNNs.

**Definition 1** (Operation $\otimes$). *Given a matrix $\mathbf{A} \in \mathbb{R}^{a \times b}$ and a tensor $\mathbf{B} \in \mathbb{R}^{a \times b \times c}$, the resulting matrix $\mathbf{B} \otimes \mathbf{A} \in \mathbb{R}^{a \times c}$ is defined by $(\mathbf{B} \otimes \mathbf{A})[i,:] = \mathbf{A}[i,:]\mathbf{B}[I,:,:]$.*

### 3.2 Hypergraph classification

We focus on the hypergraph classification task with $C \in \mathbb{Z}^+$ labels $[C] := \{1, ..., C\}$, where the input domain $\mathcal{A}$ consists of triplets $A = (\mathcal{G}, \mathbf{X}, \mathbf{Z})$. Suppose that the input-label samples follow a latent distribution $\mathcal{D}$ over $\mathcal{A} \times [C]$. We consider classifiers in the form of $f_\mathbf{w} : \mathcal{A} \to \mathbb{R}^C$ parameterized by $\mathbf{w}$, where a prediction by $\arg\max_i f_\mathbf{w}(A)[i]$ for each input $A \in \mathcal{A}$. Given a parametric space $F$ of classifiers and a training set $S = \{(A_i, y_i)\}$ consisting of $m \in \mathbb{Z}^+$ iid samples from $\mathcal{D}$, our goal is to learn a classifier $f_\mathbf{w} \in F$ that can minimize the true error [48]:

$$\mathcal{L}_\mathcal{D}(f_\mathbf{w}) = \mathbb{E}_{(A,y) \sim \mathcal{D}} \left[ \mathbb{1}\left( f_\mathbf{w}(A)[y] \leq \max_{j \neq y} f_\mathbf{w}(A)[j] \right) \right], \quad (1)$$

where $\mathbb{1}(\cdot) \in \{0,1\}$ is the indicator function. The empirical loss $\mathcal{L}_{S,\gamma}(f_\mathbf{w})$ we consider is the common used multiclass margin loss [23, 35, 42, 48] with respect to a specified margin $\gamma \in \mathbb{R}^+$:

$$\mathcal{L}_{S,\gamma}(f_\mathbf{w}) = \frac{1}{m} \sum_{(A,y) \in S} \mathbb{1}\left( f_\mathbf{w}(A)[y] \leq \gamma + \max_{j \neq y} f_\mathbf{w}(A)[j] \right). \quad (2)$$

## 4 Generalization performance of HyperGNNs

In this section, we present the main results of this paper. We proceed by introducing the standard PAC-Bayes framework and then present the generalization bounds for a representative HyperGNN from each class discussed in Section 2.

### 4.1 The analytical framework

In the PAC-Bayes framework, given a prior distribution over the hypothesis space, which refers to the weight space of the model, the posterior distribution over model parameters is updated based on the training data. This framework provides a generalization bound for models that are drawn from the posterior distribution [43, 44]. Building on this foundation, recent work has developed margin-based generalization bounds for deterministic models by introducing controlled random perturbations [42, 48]. In particular, the posterior distribution can be represented as the learned parameters with an added random perturbation. As long as the Kullback-Leibler (KL) divergence between the prior and posterior

distributions remains tractable, the standard PAC-Bayesian bound can be derived, provided that the shift in the model's output caused by the perturbation is small [38, 42, 48].

Under such a learning framework, our analysis focuses on deriving generalization bounds for HyperGNNs by scrutinizing their unique message-passing schemes. The main challenge lies in designing suitable prior and posterior distributions that must meet three critical conditions: (a) a tractable KL divergence, (b) adherence to perturbation constraints, and (c) constructing a countable covering of the hypothesis space, due to the fact that the standard framework is typically tailored to one fixed model. The complex aggregation mechanisms and multi-way interactions in HyperGNNs necessitate a refined perturbation analysis to manage recursive dependencies and inequalities effectively. Additionally, achieving a finite covering is essential to making the union bound tractable in the PAC-Bayes analysis, as it allows for the approximation of the infinite set of possible weights with a finite subset. In addition, the perturbation bounds also influence the covering size, requiring a precise design that conforms to the specific format needed to derive the bound.

### 4.2 UniGCN

For HyperGCNs, we examine UniGCN [28] which adapts the standard GCN architecture for hypergraphs by integrating degree-based normalization for nodes and hyperedges. UniGCN takes the hypergraph $\mathcal{G}$ and node feature $\mathbf{X}$ as input, with the initial node representation $H^{(0)} = \mathbf{X} \in \mathbb{R}^{N \times d}$. Suppose that the model has $L \in \mathbb{Z}^+$ propagation steps. In each propagation step $l \in [L]$, the model computes the node representation $H^{(l)} \in \mathbb{R}^{N \times d_l}$ with $d_l \in \mathbb{Z}^+$ by

$$H^{(l)} = \mathbf{C}_4^\top \mathbf{C}_3^\top \text{ReLu}\left( \mathbf{C}_2^\top \left( \boldsymbol{\eta}^{(l)} \otimes (\mathbf{C}_1^\top H^{(l-1)}) \right) \right),$$

where $\boldsymbol{\eta}^{(l)} \in \mathbb{R}^{K \times d_{l-1} \times d_l}$ is defined by $\boldsymbol{\eta}^{(l)}[j,:] = \mathbf{W}^{(l)}$ for $j \in [K]$ with $\mathbf{W}^{(l)} \in \mathbb{R}^{d_{l-1} \times d_l}$ being the parameter in layer $l$. The matrices $\mathbf{C}_1 \in \mathbb{R}^{N \times K}$, $\mathbf{C}_2 \in \mathbb{R}^{K \times N}$, and $\mathbf{C}_3, \mathbf{C}_4 \in \mathbb{R}^{N \times N}$ encode the hypergraph structure, as follows.

$$\mathbf{C}_1[i,j] := \begin{cases} 1 & \text{if } v_i \in e_j \\ 0 & \text{otherwise} \end{cases}, \mathbf{C}_2[i,j] := \begin{cases} 1/\sqrt{d_{e_i}} & \text{if } e_i \in r_i \\ 0 & \text{otherwise} \end{cases},$$

$$\mathbf{C}_3[i,j] := \begin{cases} 1/\sqrt{|N_i|+1} & \text{if } i = j \\ 0 & \text{otherwise} \end{cases}, \mathbf{C}_4[i,j] := \begin{cases} 1 & \text{if } v_i \in N_j \\ 0 & \text{otherwise} \end{cases},$$

where $d_{e_i} = \frac{1}{|e_i|} \sum_{v_j \in e_i} |N_j| + 1$. The readout layer for the classification task is defined as

$$\text{UniGCN}_\mathbf{w}(A) = \frac{1}{N} \mathbf{1}_N H^{(L)} \mathbf{W}^{(L+1)},$$

where $\mathbf{W}^{(L+1)} \in \mathbb{R}^{d_L \times C}$ and $\mathbf{1}_N$ is an all-one vector. Let the maximum hidden dimension be $h := \max_{l \in [L]} d_l$.

In order to examine the generalization capacity of UniGCN, the following lemma, as a necessary step to establish the perturbation condition, shows that its perturbation can be bounded in terms of the spectral norm of the learned weights and hypergraph statistics.

LEMMA 1. *Consider* UniGCN$_\mathbf{w}$ *with $L + 1$ layers and parameters $\mathbf{w} = (\mathbf{W}^{(1)}, \ldots, \mathbf{W}^{(L+1)})$. For each $\mathbf{w}$, any perturbation $\mathbf{u} = (\mathbf{U}^{(1)}, \ldots, \mathbf{U}^{(L+1)})$ on $\mathbf{w}$ such that $\max_{i \in [L+1]} \frac{\|\mathbf{U}^{(i)}\|}{\|\mathbf{W}^{(i)}\|} \leq \frac{1}{L+1}$, and*

*each input $A \in \mathcal{A}$, we have*

$$\|\text{UniGCN}_{\mathbf{w+u}}(A) - \text{UniGCN}_{\mathbf{w}}(A)\|_2$$

$$\leq eB(DRM)^L \Big(\prod_{i=1}^{L+1} \|\mathbf{W}^{(i)}\|\Big)\Big(\sum_{i=1}^{L+1} \frac{\|\mathbf{U}^{(i)}\|}{\|\mathbf{W}^{(i)}\|}\Big).$$

PROOF SKETCH. The main part of the proof is to analyze the maximum change of the node representation $\Psi_l$ in $l$-layer caused by the perturbation of parameters. Due to the Lipschitz property of the ReLu function, $\Psi_l$ is bounded via a summation of two terms that are linear, respectively, in a) $\Psi_{l-1}$ and b) the maximum node representation in layer $l-1$, which we denote as $\Phi_{l-1}$. We then derive the following recursive formula.

$$\Psi_l \leq C\Psi_{l-1}\|\mathbf{W}^{(l)} + \mathbf{U}^{(l)}\| + C\Phi_{l-1}\|\mathbf{U}^{(l)}\|,$$

where $C = DRM$. Therefore, $\Psi_l$ can be recursively bounded if an analytical form of $\Phi_l$ is available. To this end, we observe that $\{\Phi_1, ..., \Phi_L\}$ forms a geometric sequence, where the common ratio depends on a) the spectral norm of weights on the previous layer and b) the number of link connections between layers, which are further decided by the hypergraph statistics. By solving the recursive, we have

$$\Phi_l \leq \|\mathbf{W}^{(l)}\|DRM\Phi_{l-1}.$$

Finally, combined with the mean readout function in the last layer, we have the perturbation bound for UniGCN. □

With the above result, we have the generalization bound as follows.

THEOREM 1. *For* $\text{UniGCN}_{\mathbf{w}}$ *with* $L+1$ *layers and each* $\delta, \gamma > 0$, *with probability at least* $1 - \delta$ *over a training set $S$ of size $m$, for any fixed* $\mathbf{w}$, *we have*

$$\mathcal{L}_{\mathcal{D}}(\text{UniGCN}_{\mathbf{w}}) \leq \mathcal{L}_{S,\gamma}(\text{UniGCN}_{\mathbf{w}})$$

$$+ O\Big(\sqrt{\frac{L^2 B^2 h \ln(Lh)(RMD)^L \mathcal{W}_1 \mathcal{W}_2 + \log\frac{mL}{\sigma}}{\gamma^2 m}}\Big), \quad (3)$$

*where* $\mathcal{W}_1 = \prod_{i=1}^{L+1} \|\mathbf{W}^{(i)}\|^2$ *and* $\mathcal{W}_2 = \sum_{i=1}^{L+1} \frac{\|\mathbf{W}^{(i)}\|_F^2}{\|\mathbf{W}^{(i)}\|^2}$.

PROOF SKETCH. Due to the homogeneity of ReLu, the perturbation bound will not change after weight normalization, and therefore, it suffices to consider the UniGCN where each $\mathbf{W}^{(i)}$ is normalized by a factor of $\beta/\|\mathbf{W}^{(i)}\|$ with $\beta = (\prod_{i=1}^{L+1}\|\mathbf{W}^{(i)}\|)^{1/L+1}$. The advantage of doing so is that the weight in each layer now has the same spectral norm, which is exactly $\beta$. For such normalized models, the generalization bound is proved by discussing cases depending on the position of $\beta$ relative to

$$[I_1, I_2] \coloneqq \Big[\Big(\frac{\gamma}{2B(DRM)^L}\Big)^{1/L+1}, \Big(\frac{\gamma\sqrt{m}}{2B(DRM)^L}\Big)^{1/L+1}\Big].$$

If $\beta \leq I_1$, the perturbation condition is satisfied trivially, thereby implying Equation 3. If $\beta \geq I_2$, Equation 3 follows from the observation that there exists prior $P$ and posterior $Q$ such that the regularization term in Equation 3 is always no less than one. For $\beta \in [I_1, I_2]$, we partition $[I_1, I_2]$ into sufficiently small sub-intervals such that each sub-interval admits $P$ and $Q$ that can make the perturbation condition in the standard framework satisfied, i.e., Lemma

5 in Appendix. Finally, Theorem 1 is proved by taking the union bound over the above cases. □

**Remark 1.** Considering other models within the HyperGCNs category, we observe that the proposed approach remains applicable due to the similarity in mechanisms with UniGCN. For example, the powerful HyperGCN model, HGNN [22], uses a truncated Chebyshev polynomial to approximate the hypergraph Laplacian, allowing for efficient spectral filtering and the capture of higher-order interactions within hypergraphs. We found that HGNN and UniGCN share a similar framework for modeling relationships between vertices and hyperedges. Due to space limitations, the main results for HGNN are provided in Sec C.10 in the Appendix.

## 4.3 AllDeepSets

For the HyperMPNNs, AllDeepSets is selected for its use of techniques from DeepSets [68], incorporating layer transformations that act as universal approximators for multiset functions. Given the hypergraph $\mathcal{G}$ and features $\mathbf{X}$ and $\mathbf{Z}$, the initial representation $H^{(0)} \in \mathbb{R}^{(N+K)\times d}$ is computed by $H^{(0)}[i,:] = \mathbf{X}[i,:]$ for $i \in [N]$ and $H^{(0)}[N+k,:] = \mathbf{Z}[k,:]$ for $k \in [K]$. Suppose that there are $L$ propagation steps. During each step $l \in [L]$, the model calculates the hidden representations $\bar{H}^{(l)} \in \mathbb{R}^{(N+K)\times d_{l-1}}$ and $H^{(l)} \in \mathbb{R}^{(N+K)\times d_l}$ by

$$\bar{H}^{(l)} = \text{ReLu}\big(\boldsymbol{\eta}_2^{(l)} \otimes \big(\mathbf{C}_e^\intercal\big(\text{ReLu}\big(\boldsymbol{\eta}_1^{(l)} \otimes H^{(l-1)}\big)\big)\big)\big) \quad \text{and}$$

$$H^{(l)} = \text{ReLu}\big(\boldsymbol{\eta}_4^{(l)} \otimes \big(\mathbf{C}_v^\intercal\big(\text{ReLu}\big(\boldsymbol{\eta}_3^{(l)} \otimes \bar{H}^{(l)}\big)\big)\big)\big),$$

where a) $\boldsymbol{\eta}_1^{(l)}, \boldsymbol{\eta}_2^{(l)}, \boldsymbol{\eta}_3^{(l)} \in \mathbb{R}^{(N+K)\times d_{l-1}\times d_{l-1}}$, and the shape of $\boldsymbol{\eta}_4^{(l)} \in \mathbb{R}^{(N+K)\times d_{l-1}\times d_l}$; b) $\boldsymbol{\eta}_i^{(l)}[k,:] = \boldsymbol{\eta}_i^{(l)}[j,:] = \mathbf{W}_i^{(l)}$ for $j \neq k$ and $i \in \{1,2,3,4\}$, with $\mathbf{W}_i^{(l)}$ being the learnable parameter; c) $\mathbf{C}_e, \mathbf{C}_v \in \{0,1\}^{(N+K)\times(N+K)}$ are the fixed matrices as follows.

$$\mathbf{C}_e[i,j] \coloneqq \begin{cases} 1 & \text{if } v_i \in e_{j-N} \text{ and } i = j \\ 0 & \text{otherwise} \end{cases}, \quad \text{and}$$

$$\mathbf{C}_v[i,j] \coloneqq \begin{cases} 1 & \text{if } e_{i-N} \in r_{i-N} \text{ and } i = j \\ 0 & \text{otherwise} \end{cases}.$$

The readout layer is given by

$$\text{AllDeepSets}(A) = \frac{1}{N+K}\mathbf{1}_{N+K}H^{(L)}\mathbf{W}^{L+1},$$

where $\mathbf{W}^{(L+1)} \in \mathbb{R}^{d_L \times C}$ and $\mathbf{1}_{N+K}$ is an all-one vector. In summary, the parameters are $\mathbf{W}^{(L+1)}$ and $\mathbf{W}_i^{(j)}$ for $i \in \{1,2,3,4\}$ and $j \in [L]$. Let the maximum hidden dimension be $h \coloneqq \max_{l \in [L]} d_l$. The generalization bound follows from the perturbation analysis.

LEMMA 2. *Consider* $\text{AllDeepSets}_{\mathbf{w}}$ *of $L$ propagation steps with parameters* $\mathbf{w} = \big(\mathbf{W}_i^{(j)}, \mathbf{W}^{(L+1)}\big)$. *For each* $\mathbf{w}$, *any perturbation* $\mathbf{u} = \big(\mathbf{U}_i^{(j)}, \mathbf{U}^{(L+1)}\big)$ *on* $\mathbf{w}$ *such that* $\max\big(\frac{\|\mathbf{U}_i^{(j)}\|}{\|\mathbf{W}_i^{(j)}\|}, \frac{\|\mathbf{U}^{(L+1)}\|}{\|\mathbf{W}^{(L+1)}\|}\big) \leq \frac{1}{4L+1}$, *and each input* $A \in \mathcal{A}$, *we have*

$$\|\text{AllDeepSets}_{\mathbf{w+u}}(A) - \text{AllDeepSets}_{\mathbf{w}}(A)\|_2$$

$$\leq C_A\Big(\prod_{j=1}^{L} \zeta_j\Big)\big(\|\mathbf{U}^{(L+1)}\|\big),$$

where $C_A = 30(4L + 2)eBL(M + 1)^L(R + 1)^L$ and $\zeta_j = \prod_{i=1}^{4} \|\mathbf{W}_i^{(j)}\|$.

**THEOREM 2.** *For* AllDeepSets$_\mathbf{w}$ *with $L$ propagation steps and each $\delta, \gamma > 0$, with probability at least $1 - \delta$ over a training set $S$ of size $m$, for any fixed* $\mathbf{w}$, *we have*

$$\mathcal{L}_{\mathcal{D}}(\text{AllDeepSets}_\mathbf{w}) \leq \mathcal{L}_{S,\gamma}(\text{AllDeepSets}_\mathbf{w})+$$

$$O(\sqrt{\frac{L^2 B^2 (RM)^L h \ln(hL)\mathcal{W}_1\mathcal{W}_2 + \log\frac{mL}{\sigma}}{\gamma^2 m}}),$$

*where* $\mathcal{W}_1 = \prod_{j=1}^{L}(\zeta_j)^2\|\mathbf{W}^{(L+1)}\|^2$, *and*

$$\mathcal{W}_2 = \sum_{j=1}^{L}\frac{\prod_{i=1}^{4}\|\mathbf{W}_i^{(j)}\|_F^2}{\prod_{i=1}^{4}\|\mathbf{W}_i^{(j)}\|^2} + \frac{\|\mathbf{W}^{(L+1)}\|_F^2}{\|\mathbf{W}^{(L+1)}\|^2}.$$

**Remark 2.** The challenge of obtaining the perturbation bound lies in the structure of AllDeepSet that employs two multiset functions. These two functions are represented as two types of aggregation mechanisms, causing compounded sensitivity response to the perturbation in weights. Consequently, the maximum change of the node representation in layer $l$, $\Psi_l$, is recursively through $\Psi_{l-1}$ and maximum node representation in layer $l - 1$, $\Phi_{l-1}$, where $\Psi_l$ is partially bounded by $\Phi_{l-1}$ via a factor involving $\zeta_l$. In addition, in applying the normalization trick in the proof of Theorem 1, we consider the AllDeepSet normalized by a factor of $\left(\|\mathbf{W}^{(L+1)}\| \cdot \prod_{j=1}^{L}\prod_{i=1}^{4}\|\mathbf{W}_i^{(j)}\|\right)^{1/(4L+1)}$. Accordingly, to obtain the generalization bound, the necessary discussion cases of the spectral norm of weights are decided by the interval as follows.

$$[I_1, I_2] := \left[\left(\frac{\gamma}{2B(M+1)(R+1)}\right)^{1/4L+1}, \left(\frac{\sqrt{m}\gamma}{2B(M+1)(R+1)}\right)^{1/4L+1}\right].$$

**Remark 3.** For the other models in HyperMPNNs, HyperSAGE can be analyzed using a similar approach to AllDeepSets because (a) both models utilize a propagation mechanism that involves a two-step aggregation between nodes and hyperedges, as seen in AllDeepSets, and (b) their activation functions (i.e., identity functions) are homogeneous which allows the normalization trick. In contrast, obtaining generalization bounds for attention-based aggregators (e.g., UniGAT [28] and AllSetTransformer [10]) presents two major challenges. First, the dynamic and highly nonlinear dependencies introduced by attention mechanisms require new techniques to derive solvable recursive formulas for perturbation bounds. Second, the use of the softmax function in these models, which is inherently non-homogeneous, prevents the application of standard normalization tricks. Therefore, our approach cannot be directly applied to attention-based HyperGNNs.

## 4.4 M-IGN

Regarding HyperGINs, we focus on analyzing the M-IGN model, which utilizes a set of fixed scalar values in each layer [28]. The initial hyperedges representation $H^{(0)} \in \mathbb{R}^{K \times d_0}$ is given by

$$\bar{H}^{(0)}[k, :] = \sum_{v_i \in e_k}(\mathbf{X}[v_i, :]) \quad \text{and} \quad H^{(0)} = \text{ReLu}(\bar{H}^{(0)}\mathbf{W}_0),$$

where $\mathbf{W}_0 \in \mathbb{R}^{d \times d_0}$ with $d_0 \in \mathbb{Z}^+$. Suppose that there are $L \in \mathbb{Z}^+$ propagation steps. In each step $l \in [L]$, a hyperparameter $\alpha^{(l)} \in [0, 1]$ is used to control the balance between the hyperedge feature and the aggregated node feature from their neighbors. The model computes the hidden hyperedge representation $H^{(l)} \in \mathbb{R}^{K \times d_l}$ with $d_l \in \mathbb{Z}^+$ by

$$\bar{H}^{(l)}[k, :] = (1 + \alpha^{(l)})H^{(l-1)}[k, :] + \sum_{e_i \in N(e_k)}H^{(l-1)}[i, :] \text{ and}$$

$$H^{(l)} = \text{ReLu}(\bar{H}^{(l)}\mathbf{W}^{(l)}),$$

where a) $\mathbf{W}^{(l)} \in \mathbb{R}^{d_{l-1} \times d_l}$, b) $N(e_k) \subseteq \mathcal{E}$ denotes the neighborhood of $e_k$, and c) $N(e_k) = \{e | e \cap e_k \neq \emptyset\}$. Through an aggregation process, the readout layer is computed by

$$\bar{H}^{(L+1)}[k, :] = \sum_{e_i \in N(e_k)}H^{(L)}[i, :] \quad \text{and}$$

$$\text{M-IGN}(A) = \frac{1}{K}\mathbf{1}_K(\text{ReLu}(\bar{H}^{(L+1)}\mathbf{W}^{(L+1)})),$$

where $\mathbf{W}^{(L+1)} \in \mathbb{R}^{d_{L+1} \times C}$ and $\mathbf{1}_K$ is an all-one vector. Let the maximum hidden dimension be $h := \max_{l \in [0, L+1]} d_l$. We now provide the perturbation bound.

**LEMMA 3.** *Consider the* M-IGN$_\mathbf{w}$ *of $L + 2$ layers with parameters* $\mathbf{w} = \left(\mathbf{W}^{(0)}, \ldots, \mathbf{W}^{(L+1)}\right)$. *For each* $\mathbf{w}$, *any perturbation* $\mathbf{u} = \left(\mathbf{U}^{(0)}, \ldots, \mathbf{U}^{(L+1)}\right)$ *on* $\mathbf{w}$ *such that* $\max_{i \in \{0\} \cup [L+1]}\left(\frac{\|\mathbf{U}^{(i)}\|}{\|\mathbf{W}^{(i)}\|}\right) \leq \frac{1}{L+2}$, *and for each input* $A \in \mathcal{A}$, *we have*

$$\|\text{M-IGN}_{\mathbf{w}+\mathbf{u}}(A) - \text{M-IGN}_\mathbf{w}(A)\|_2$$

$$\leq C_{I_1}\left(\prod_{i=0}^{L+1}\|\mathbf{W}^{(i)}\|\right)\left(\sum_{i=0}^{L+1}\frac{\|\mathbf{U}^{(i)}\|}{\|\mathbf{W}^{(i)}\|}\right),$$

*where* $C_{I_1} = 2e^2 M^{L+2}D^{L+1}BE^{(1,L)}$ *and* $E^{(i,j)} = \prod_{k=i}^{j}1 + \alpha^{(k)}$.

**THEOREM 3.** *For* M-IGN$_\mathbf{w}$ *with $L + 2$ layers and each $\delta, \gamma > 0$, with probability at least $1 - \delta$ over a training set $S$ of size $m$, for any fixed* $\mathbf{w}$, *we have*

$$\mathcal{L}_{\mathcal{D}}(\text{M-IGN}_\mathbf{w}) \leq \mathcal{L}_{S,\gamma}(\text{M-IGN}_\mathbf{w}) + O(\sqrt{\frac{C_{I_2}\mathcal{W}_1\mathcal{W}_2 + \log\frac{mL}{\delta}}{\gamma^2 m}}),$$

*where* a) $C_{I_2} = (MD)^L B^2 h \ln(Lh)(E^{(1,L)})^2$ *with* $E^{(i,j)}$ *being defined as* $\prod_{k=i}^{j}1 + \alpha^{(k)}$, b) $\mathcal{W}_1 = \prod_{i=0}^{L+1}\|\mathbf{W}^{(i)}\|^2$, *and* c) $\mathcal{W}_2 = \sum_{i=0}^{L+1}\frac{\|\mathbf{W}^{(i)}\|_F^2}{\|\mathbf{W}^{(i)}\|^2}$.

**Remark 4.** The main challenge with M-IGN lies in its use of layer-specific scalars for managing propagation, where parameter variations across different layers lead to non-uniform propagation behaviors. This non-uniformity complicates the derivation of the recursive inequality for $\Psi_l$, as the additional term introduced by these parameters needs to be carefully reordered to ensure compatibility with the other terms. We then derive the generalization bound by examining the interval of the spectral norm of weights, specified as follows.

$$[I_1, I_2] := \left[\left(\frac{\gamma}{2E^{(1,L)}M^{L+1}D^L B}\right)^{1/L+2}, \left(\frac{\sqrt{m}\gamma}{2E^{(1,L)}M^{L+1}D^L B}\right)^{1/L+2}\right].$$

**Remark 5.** The given approach can be extended to other Hyper-GINs. Given the similarity in their propagation mechanisms, we can leverage the same framework to analyze the generalization properties of these models, with adjustments to the recursive inequalities to accommodate their specific layer-wise characteristics. For example, in $k$-GNN [47], each layer aggregates structural information

from interactions between nodes or subgraphs with layer-specific scalars. These interactions result in a solvable recursive formula for the perturbation bound, where the linear factor is directly represented by the substructure properties. This allows for a consistent analysis of the model, similar to M-IGN.

## 4.5 T-MPHN

The tensor-based HyperGNN, T-MPHN [59] leverages high dimensional hypergraph descriptors and joint node interaction inherent in hyperedges for message passing. The model takes $\mathcal{G}$ and node feature $\mathbf{X}$ as input, and the initial hidden node representation $H^{(0)} \in \mathbb{R}^{N \times d_0}$ is computed by $H^{(0)} = \text{ReLu}(\mathbf{W}^{(0)}\mathbf{X})$ with $\mathbf{W}^{(0)} \in \mathbb{R}^{d \times d_0}$. Suppose that there are $L \in \mathbb{Z}^+$ propagation steps. For each node $v_i \in \mathcal{V}$, we denote its representation in step $l$ by $\mathbf{x}_{v_i}^{(l)} \in \mathbb{R}^{d_l}$. The model updates the node representation $\mathbf{x}_{v_i}^{(l')} \in \mathbb{R}^{2d_{l-1}}$ by

$$\mathbf{m}_{e^M(v_i)}^{(l)} \coloneqq \underset{\{\mathcal{U} \in \pi(\cdot)|\pi(\cdot) \in \pi(e^M(-v_i))\}}{\text{SUM}} \big(\text{CNI}_{\mathcal{U}}(H^{(l-1)})\big),$$

$$\mathbf{m}_{\mathcal{N}^M(v_i)}^{(l)} \coloneqq \underset{e^M \in E^M(v_i)}{\text{AVERAGE}}\big(a_e \mathbf{m}_{e^M(v_i)}^{(l-1)}\big), \quad \text{and}$$

$$\mathbf{x}_{v_i}^{(l')} = \text{CONCAT}\big(\mathbf{x}_{v_i}^{(l-1)}, \mathbf{m}_{\mathcal{N}^M(v_i)}^{(l)}\big),$$

where a) $e^M(v_i)$ represents the $M^{th}$-order hyperedge, b) $\pi(\cdot)$ denotes the sequence permutation function, c) $\text{CNI}_{\mathcal{U}}(\cdot)$ represents a matrix operation for an ordered sequence $\mathcal{U}$ of indexes, d) $E^M(v_i)$ indicates the $M^{th}$-order incident hyperedge of node $v_i$, e) $\mathcal{N}^M(v_i)$ is the $M^{th}$-order neighborhood of node $v_i$, and f) $a_e$ denotes adjacency value of hyperedge $e$. The definitions of the above terminologies are presented in Sec A.1 in the Appendix. Let $G^{(l)} = (\mathbf{x}_{v_1}^{(l')}, \mathbf{x}_{v_2}^{(l')}, \ldots, \mathbf{x}_{v_N}^{(l')})$. The model then calculates the hidden node representation $H^{(l)} \in \mathbb{R}^{N \times d_l}$ by adding a row-wise normalization:

$$\bar{H}^{(l)} = \text{ReLu}\big(\mathbf{W}^{(l)}G^{(l)}\big)$$

$$H^{(l)} = \left(\frac{\bar{H}^{(l)}[1,:]}{\|\bar{H}^{(l)}[1,:]\|_2}, \ldots, \frac{\bar{H}^{(l)}[N,:]}{\|\bar{H}^{(l)}[N,:]\|_2}\right),$$

where $\mathbf{W}^{(l)} \in \mathbb{R}^{2d_{l-1} \times d_l}$. The readout layer is defined as

$$\text{T-MPHN}(A) = \frac{1}{N}\mathbf{1}_N H^{(L)}\mathbf{W}^{(L+1)},$$

where $\mathbf{W}^{(L+1)} \in \mathbb{R}^{d_L \times C}$ and $\mathbf{1}_N$ is an all-one vector. Let the maximum hidden dimension be $h \coloneqq \max_{l \in [L]} d_l$. We have the following results for T-MPHN.

LEMMA 4. *Consider the* T-MPHN$_\mathbf{w}$ *of* $L + 1$ *layers with parameters* $\mathbf{w} = \big(\mathbf{W}^{(1)}, \ldots, \mathbf{W}^{(L+1)}\big)$. *For each* $\mathbf{w}$, *each perturbation* $\mathbf{u} = \big(\mathbf{U}^{(1)}, \ldots, \mathbf{U}^{(L+1)}\big)$, *and each input* $A \in \mathcal{A}$, *we have*

$$\|\text{T-MPHN}_{\mathbf{w}+\mathbf{u}}(A) - \text{T-MPHN}_{\mathbf{w}}(A)\|_2 \le 2\|\mathbf{W}^{(L+1)}\| + 3\|\mathbf{U}^{(L+1)}\|.$$

THEOREM 4. *For* T-MPHN$_\mathbf{w}$ *with* $L + 1$ *layers and each* $\delta, \gamma > 0$, *with probability at least* $1 - \delta$ *over a training set* $S$ *of size* $m$, *for any fixed* $\mathbf{w}$, *we have*

$$\mathcal{L}_{\mathcal{D}}(\text{T-MPHN}_{\mathbf{w}}) \le \mathcal{L}_{S,\gamma}(\text{T-MPHN}_{\mathbf{w}})+$$

$$O\left(\sqrt{\frac{L^2 h \ln h \sum_{i=1}^{L+1}\|\mathbf{W}\|_F^2 + \log\frac{mL}{\sigma}}{\gamma^2 m + \|\mathbf{W}^{(L+1)}\|^2 m}}\right).$$

**Table 1: Bounds comparison.** $L$ is the number of propagations. Note that $\mathcal{W}_p$ and $\mathcal{W}_s$ denote the parameter-dependent in the bound, respectively, with their specific definitions varying slightly depending on the model.

| Model | $D$ | $M$ | $R$ | $h$ | $\mathbf{w}$ |
|---|---|---|---|---|---|
| **UniGCN** | $O(D^L)$ | $O(M^L)$ | $O(R^L)$ | $O(h\ln(Lh))$ | $O(\mathcal{W}_p\mathcal{W}_s)$ |
| **AllDeepSet** | N/A | $O(M^L)$ | $O(R^L)$ | $O(h\ln(h))$ | $O(\mathcal{W}_p\mathcal{W}_s)$ |
| **M-IGN** | $O(D^L)$ | $O(M^L)$ | N/A | $O(h\ln(Lh))$ | $O(\mathcal{W}_p\mathcal{W}_s)$ |
| **T-MPHN** | N/A | N/A | N/A | $O(h\ln(h))$ | $O\big(\frac{\sum\|\mathbf{W}\|_F^2}{\|\mathbf{W}^{(L+1)}\|^2}\big)$ |
| **HGNN** | $O(D^{\frac{L}{2}})$ | $O(M^L)$ | $O(R^L)$ | $O(h\ln(Lh))$ | $O(\mathcal{W}_p\mathcal{W}_s)$ |

**Remark 6.** Due to the tensor-based representations involving complex node interactions through advanced computational operations, deriving the recursive formula for $\Psi_l$ in T-MPHN is challenging. However, we found that the row-wise normalization results in $\Psi_{l-1}$ being bounded by the Euclidean distance between two normalized vectors. Consequently, both $\Psi_{l-1}$ and $\Phi_{l-1}$ are upper-bounded by a constant. The perturbation bound follows immediately by incorporating the mean readout function in the last layer. Different from the previous models, the perturbation bound here is always satisfied without any assumption on the spectral norm of perturbations and weights. Altogether, the generalization bound is derived by considering three cases of the weights respective to $[I_1, I_2] \coloneqq \left[\frac{\gamma}{2}, \frac{\gamma\sqrt{m}}{2}\right]$.

**Remark 7.** For other tensor-based HyperGNNs, the underlying mechanisms can vary significantly from one model to another, making it difficult to apply our method uniformly. For instance, the TNHH [60] uses outer product aggregation with partially symmetric CP decomposition, which differs from the approach used in T-MPHN. The key challenge lies in obtaining a solvable recursive formula for the perturbation bound. In particular, the THNN's outer product pooling generates high-order tensor interactions among nodes representations, causing perturbations to propagate multiplicatively rather than additively; this results in perturbation effects that are non-linear and involve higher-degree terms, making linear approximations ineffective. And it prevents the recursive propagation of perturbations. Therefore, our current method is not directly applicable to these models, and each must be analyzed individually to account for their unique aggregation method.

## 4.6 Discussion

The generalization bounds of the discussed models depend reasonably on a) the properties on the hypergraphs (i.e., $D$, $R$, and $M$), b) the hidden dimension $h$, c) the number of propagation steps $L$, and d) the spectral norm of learned parameters. We summarize such relationships in Table 1. In addition, UniGCN can be seen as the application of GCN architecture. For a graph classification task, if we treat the given graphs as hypergraphs with $M = 1$ and $R = 1$, we can obtain the following term within the Big-O notation in the Theorem 1: $\left(\sqrt{\frac{L^2 B^2 h \ln(Lh)\mathcal{W}_1\mathcal{W}_2 + \log\frac{mL}{\sigma}}{\gamma^2 m}}\right)$, which aligns with the result in [38].

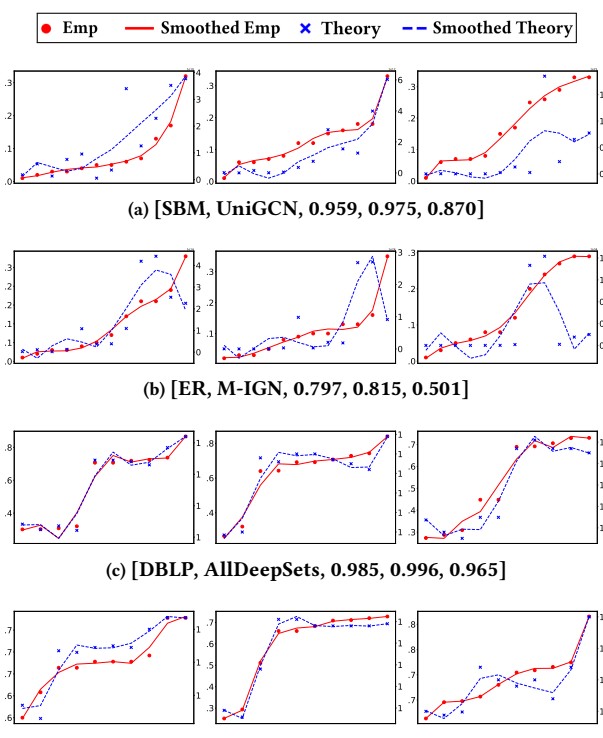

(a) [SBM, UniGCN, 0.959, 0.975, 0.870]

(b) [ER, M-IGN, 0.797, 0.815, 0.501]

(c) [DBLP, AllDeepSets, 0.985, 0.996, 0.965]

(d) [Collab, AllDeepSets, 0.937, 0.984, 0.916]

Figure 2: Consistency between empirical loss (Emp) and the-oretical bounds (Theory). Each subgroup labeled by [graph type, model, $r_2$, $r_4$, $r_6$] shows the empirical loss, theoretical bound, and their curves via Savitzky-Golay filter [53] and models (i.e., UniGCN, M-IGN, and AllDeepSets), where each figure plots the results of synthetic datasets (i.e., SBM) and real datasets (i.e., DBLP and Collab); the figures, from left to right, show the results with 2, 4 and 6 propagation steps, where $r_2$, $r_4$, and $r_6 \in [-1, 1]$ are the Pearson correlation coef-ficients between the two sets of points in each figure – higher $r$ indicating stronger positive correlation.

## 5 Experiments

Our empirical study explores three questions: Q1) Does the em-pirical error follow the trends given by the theoretical bounds? Q2) How does training influence the degree to which the empirical error consistently aligns with the theoretical trends? Q3) How do the properties of the hypergraphs and statistics on HyperGNNs influence the empirical performance?

### 5.1 Settings

We conduct hypergraph classification experiments over a) two real-world datasets DBLP-v1 [50] and COLLAB[66], b) twelve synthetic datasets generated based on the Erdos–Renyi (ER) [27] model, and c) twelve synthetic datasets generated based on the Stochastic Block Model (SBM) [1]. The synthetic datasets are generated with diverse hypergraph statistics (i.e., $N$, $M$, and $R$). For each dataset, we gen-erate a pool of input-label $(A, y)$ pairs using the HyperPA method [15, 36, 37], known for its ability to replicate realistic hypergraph

Table 2: Results of empirical loss (Emp) and theoretical bound (Theory). Each dataset is labeled by $[N, M, R, \max(D)]$.

| ER | L | [200,20,20,166] | | [200,20,40,166] | | [600,60,60,587] | |
|---|---|---|---|---|---|---|---|
| | | Emp | Theory | Emp | Theory | Emp | Theory |
| UniGCN | 2 | 0.01 | 6.46E+08 | 0.07 | 1.07E+09 | 0.17 | 2.80E+10 |
| | 4 | 0.08 | 2.79E+14 | 0.01 | 1.35E+15 | 0.18 | 3.81E+17 |
| | 6 | 0.08 | 5.66E+19 | 0.04 | 5.41E+20 | 0.12 | 8.79E+24 |
| M-IGN | 2 | 0.01 | 2.55E+12 | 0.03 | 2.10E+12 | 0.19 | 2.53E+14 |
| | 4 | 0.03 | 6.35E+19 | 0.03 | 7.74E+19 | 0.35 | 9.13E+23 |
| | 6 | 0.01 | 8.36E+26 | 0.03 | 7.07E+26 | 0.29 | 1.25E+33 |
| AllDeepSet | 2 | 0.00 | 1.04E+08 | 0.05 | 1.56E+09 | 0.26 | 1.39E+09 |
| | 4 | 0.03 | 5.21E+12 | 0.04 | 4.39E+12 | 0.27 | 1.01E+14 |
| | 6 | 0.05 | 4.82E+15 | 0.03 | 2.51E+16 | 0.25 | 2.18E+18 |

patterns, and the Wrap method [39, 64, 67] a standard method for generating label-specific features. The random train-test-valid split ratio is 0.5-0.3-0.2. The implementation of the considered models is adapted from their original code [11, 28, 59], with the propagation step $L$ being selected from $\{2, 4, 6, 8\}$. For each model on a given dataset, we examine the empirical loss and the theoretical gener-alization bound. The empirical loss is calculated either using the optimal Monte Carlo algorithm [13], which guarantees an estima-tion error of no more than $\epsilon = 10\%$ with a probability of at least $\delta = 90\%$, or by averaging over multiple runs. The details of the bound calculations can be found in Sec D.2 in the Appendix. In par-ticular, a model with random weights refers to one with randomly initialized parameters that have not undergone training, and under this setting, the empirical loss is calculated by averaging over five runs. The complete details are provided in Sec D.1 in the Appendix, including dataset statistics, sample generation, training settings, and test settings.

### 5.2 Results and observations

**Consistency between theory and practice.** Figure 2 together with Figures 4, 5, and 6 in Sec D.1 in the Appendix depicts the correlation between the empirical loss and theoretical bounds. We observe that the theoretical bounds indeed inform the empirical loss to a satisfactory extent with trained models; the Pearson corre-lation coefficients are mostly well above 0.0 and even close to 1.0 in many cases (e.g., Figures 2(a)-mid, (b)-mid, and (c)-mid). Such an observation is promising in the sense that it is arguably over-ambitious to expect that the empirical loss matches perfectly with the theoretical bounds. However, we also observed corners where such a correlation is not strong (e.g., Figures 4 (a)-left and (b)-right); interestingly, such cases are mostly associated with T-MPHN, and one possible reason is that the row-wise normalization can intro-duce scale variations across different layers. This scaling effect may lead to more stable outputs but also makes it challenging for the theoretical bounds to accurately reflect the empirical loss.

**The impact on training.** Figures 3 and 4 show the correlation obtained of the considered models with both trained and random weights. The results reveal an improvement in the alignment be-tween the theoretical bounds and the empirical loss of models with trained parameters, compared to those with random weights (e.g.,

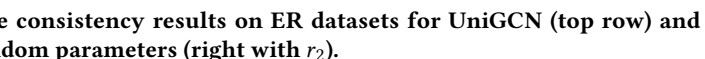

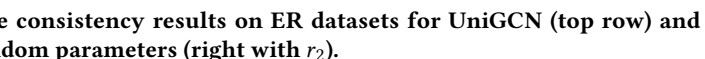

(a) [2, 0.800, 0.177]  (b) [4, 0.989, -0.091]  (c) [6, 0.959, -0.247]  (d) [8, 0.966, -0.547]

(e) [2, 0.786, -0.891]  (f) [4, 0.830, -0.083]  (g) [6, 0.875, -0.147]  (h) [8, 0.492, -0.192]

**Figure 3: Each subgroup, labeled by [$L$, $r_1$, $r_2$], presents the consistency results on ER datasets for UniGCN (top row) and AllDeepSets (bottom row) with trained (left with $r_1$) and random parameters (right with $r_2$).**

Figures 3 (d) and (e)). One possible reason for this improvement is the use of L2 regularization during training. For instance, the bound of UniGCN calculated by Equation 3, consists of two parts: the empirical loss (left term) and the complexity term (right term). The empirical loss (Equation 2), is typically smaller than the complexity term, which includes the KL divergence between the posterior and prior distributions over the model parameters and depends on the spectral norm of weights. L2 regularization reduces this complexity by penalizing large weights, thereby decreasing the KL divergence. This reduction enhances the alignment between the empirical loss and the theoretical bounds, resulting in a relatively small value of bounds that more accurately captures the model's generalization performance.

**Statistics on hypergraphs.** We examine the impact of hypergraph properties on the empirical performance and theoretical bounds. Table 2 reports the results on datasets associated with ER and SBM graphs. Results for other models and datasets can be found in Sec F in the Appendix. In general, the empirical results indicate clear patterns where changes in the complexity of hypergraphs significantly impact the model performance, echoing the theoretical bounds. We observe that for each of $R, M$, and $D$, when these values are smaller, their variation has less impact on the loss. For instance, the empirical loss across the three models shows minimal fluctuation on the first two datasets in Table 2, compared to the more significant variation observed on the last two datasets in Table 8. Regarding $R$, the results show that its impact on M-IGN is less than that of other models. Finally, larger $D$ often leads to larger empirical loss on each model (e.g., the last column in Table 2).

**The number of propagation steps.** We compare the performance of the considered HyperGNNs with different propagation steps. For datasets with smaller statistics, having more layers may result in larger loss increases (e.g., the results on UniGCN in the first dataset); in contrast, for datasets with larger statistics, complex models (i.e., larger $l$) produce better performance (e.g., the results on UniGCN in the last dataset in Table 2). Sharing the same spirit of the principle of Occam's razor, we see that a shallow model is sufficient for simpler tasks but lacks the ability to deal with complex hypergraphs, which has also been observed by existing works [7].

## 6 Conclusion and Futher Discussions

In this paper, we develop margin-based generalization bounds using the PAC-Bayes framework for four hypergraph models: UniGCNs, AllDeepSets, M-IGNs, and T-MPHN. These models were selected for their distinct approaches to leveraging hypergraph structures, enabling a comprehensive analysis of different architectures. Our empirical study reveals a positive correlation between the theoretical bounds and the empirical loss, suggesting that the bounds effectively capture the generalization behavior of these models.

**Node classification task.** Our study primarily focused on deriving generalization bounds for the hypergraph classification problem. Besides, the node classification task [18, 61] is important in hypergraph learning, with high relevance to web graph applications, i.e., user behavior prediction. While node classification focuses on predicting labels for individual nodes, it shares underlying principles with hypergraph classification, allowing our method to be naturally extended. In particular, we treat each node's output as a sub-neural network and analyze the generalization behavior of each node individually. The overall generalization bound for the model then follows from applying a union bound over all nodes and classes. More details including the problem statement and generalization bound on HyperGNNs can be found in Sec E in the Appendix.

**Future works.** Several directions for future research remain to be explored.

- Our paper focuses on HyperGNNs based on ReLu activation. One interesting future direction is to explore margin-based bound for HyperGNNs with non-homogeneous activation functions.
- Another important future direction is to derive generalization bounds for HyperGNNs using classical frameworks like the Vapnik–Chervonenkis (VC) dimension and Rademacher complexity.
- The experiments reveal a positive correlation between the theoretical bound and empirical performance, we can further investigate the degree of such correlation in theory, to systematically analyze the varying levels of consistency between empirical loss and theoretical bounds across different models, such as T-MPHN.

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

# Appendix

## A  Notations and Definitions

We summarize the notations used throughout the paper in Table 3.

**Table 3: Summary of notations.**

| Notations | Meaning |
|:---:|:---|
| $\mathcal{G}$ | the hypergraph |
| $\mathcal{V}$ | the node set |
| $N$ | the number of nodes |
| $\mathcal{E}$ | the hyperedge set |
| $K$ | the number of hyperedges |
| $\mathcal{N}_i$ | the neighbor set of node $v_i$ |
| $\mathbf{X}$ | the node features matrix |
| $\mathbf{Z}$ | the hyperedge features matrix |
| $d$ | the feature size of node and hyperedges |
| $B$ | the $l_2$ norm bound of node and hyperedge features |
| $M$ | the maximum hyperedge size |
| $R$ | the maximum incident hyperedge set size |
| $D$ | the maximum node degree |
| $\mathbf{X}[i,:]$ | the $i^{th}$ row of matrix $\mathbf{X}$ |
| $d_e$ | the degree of hyperedge $e$ |
| $\epsilon$ | the hyperparameter |
| $e^M$ | the $M^{th}$-order hyperedge |
| $\mathcal{E}^M(v_i)$ | the $M^{th}$-order incident hyperedge of node $v_i$ |
| $\mathcal{N}^M(v_i)$ | the $M^{th}$-order neighborhood of node $v_i$ |
| $a_e$ | the adjacency value of hyperedge $e$ |
| $\mathbf{T}$ | the matrix of the hyperedge weight |
| $\mathbf{J}$ | the incident matrix |
| $S$ | the set of training data |
| $m$ | the size of training data |
| $C$ | the number of classes |
| $y$ | the hypergraph label |
| $\mathcal{D}$ | the distribution over sample space |
| $A$ | the input $A = (\mathcal{G}, \mathbf{X}, \mathbf{Z})$ |
| $F$ | the hypothesis space |
| $\gamma$ | the margin |
| $\mathcal{L}_\mathcal{D}$ | the multiclass margin generalization loss |
| $\mathcal{L}_{S,\gamma}$ | the multiclass margin empirical loss |
| $P$ | the prior distribution over the learned parameters |
| $Q$ | the posterior distribution over the perturbed parameters |
| $h$ | the maximum hidden dimension |
| $L$ | the number of propagation steps |

### A.1  Terminologies used in T-MPHN

Recall that in the architecture of T-MPHN, each propagation step computes the hidden representation of nodes. We introduce the related terminologies in the T-MPHN in the following.

**Definition 2** ($M^{th}$-**order Hyperedge**[59]). Given a hypergraph $\mathcal{G} = (\mathcal{V}, \mathcal{E})$ with the order $M$, for any hyperedge $e \in \mathcal{E}$, its $M^{th}$-order hyperedge set $e^M$ is given by

$$e^M = \begin{cases} \{e\}, & if\,|e| = M \\ \text{span}^M(e), & if\,|e| < M \end{cases},$$

where

$$\text{span}^M(e) = \{e'|\text{unique}(e') = e, |e'| = M\},$$

where $\text{unique}(e') = e$ means the distinct elements in $e'$ is the same as $e$, and $|e'|$ is the number of (possibly nonunique) elements in $e'$. Notice that the size of $\text{span}^M(e)$ is equal to $\binom{M-1}{|e|-1}$.

**Definition 3** ($M^{th}$**-order Neighborhood of A Node**[59]). Given a hypergraph $\mathcal{G} = (\mathcal{V}, \mathcal{E})$ with the order $M$, for any node $v \in \mathcal{V}$, its $M^{th}$-order incidence edge set is

$$E^M(v) = \{e^M | e \in \mathcal{E}, v \in e\}.$$

Then the $M^{th}$-order neighborhood of $v$ is defined as

$$\mathcal{N}^M(v) := \{\pi(e^M(-v)) | e^M \in E^M(v)\},$$

where $e^M(-v)$ deletes exactly one node of $v$ from each $M^{th}$-order hyperedge in $e^M$, and $\pi(\cdot)$ represents permutation of the remaining nodes.

**Definition 4** (**Adjacency Value** $a_e$ [59]). Given an adjacency tensor of a hypergraph, the adjacency value $a_e$ associated with a hyperedge $e$ is a function of $(|e|, M)$:

$$a_e = \frac{|e|}{\sum_{i=0}^{|e|} (-1)^i \binom{|e|}{i}(|e|-i)^M}.$$

## B Analytical Framework of PAC-Bayes

This section discusses the theoretical framework underlying the derivation of margin-based generalization bounds. The following lemma, adapted from the work of Neyshabur et al. [48], establishes a probabilistic upper bound on the generalization error of a predictor parameterized by weights, using the PAC-Bayes framework. This result quantifies how the empirical performance of the model relates to its expected performance on unseen data while incorporating the impact of weight perturbations. The bound leverages the KL divergence between the prior and posterior distributions over the model parameters, along with a condition that ensures stability under perturbations.

LEMMA 5. [48] Consider a predictor $f_{\mathbf{w}}(A) : \mathcal{A} \to \mathbb{R}^C$ parameterized by $\mathbf{w}$. Let $P$ be any distribution over $\mathbf{w}$ that is independent of the training data, and $Q$ be any perturbation distribution over $\mathbf{u}$. For each $\gamma, \delta > 0$, with probability no less than $1 - \delta$ over a training set $S$ of size $m$, for each fixed $\mathbf{w}^*$, we have

$$\mathcal{L}_{\mathcal{D}}(f_{\mathbf{w}^*}) \le \mathcal{L}_{S,\gamma}(f_{\mathbf{w}^*}) + 4\sqrt{\frac{\text{KL}(\mathbf{w}^* + \mathbf{u}||\mathbf{w}) + \ln\frac{6m}{\delta}}{m-1}}, \quad (4)$$

where $\mathbf{u} \sim Q$ and $\mathbf{w} \sim P$, provided that we have the following perturbation condition for each $\mathbf{w}$

$$\Pr_{\mathbf{u} \sim Q}\left[\max_{A \in \mathcal{A}} \|f_{\mathbf{w}+\mathbf{u}}(A) - f_{\mathbf{w}}(A)\|_\infty < \frac{\gamma}{4}\right] \ge \frac{1}{2}.$$

## C Proofs

### C.1 Proof of Lemma 1

The proof of Lemma 1 includes two parts. We first analyze the maximum node representation among each layer except the readout layer. After adding the perturbation $\mathbf{u}$ to the weight $\mathbf{w}$, for each layer $l \in [L+1]$, we denote the perturbed weights $\mathbf{W}^{(l)} + \mathbf{U}^{(l)}$. We define $\boldsymbol{\theta} \in \mathbb{R}^{K \times d_{l-1} \times d_l}$ as the perturbation tensors. In particular $\boldsymbol{\theta}^{(l)}[k,:] = \boldsymbol{\theta}^{(l)}[j,:] = \mathbf{U}^{(l)} \in \mathbb{R}^{d_{l-1} \times d_l}$ when $j \ne k$. We then

can derive an upper bound on the $l_2$ norm of the maximum node representation in each layer. Let $w_l^* = \arg\max_{i \in [N]} \|H^{(l)}[i,:]\|_2$ and $\Phi_l = \|H^{(l)}[w_l^*,:]\|_2$.

$$\Phi_l = \|\mathbf{C}_4^\top \mathbf{C}_3^\top \text{ReLu}(\mathbf{C}_2^\top(\boldsymbol{\eta}^{(l)} \otimes (\mathbf{C}_1^\top H^{(l-1)})))[w_l^*,:]\|_2$$

$$= \|\sum_{i=1}^N \mathbf{C}_4^\top[w_l^*,i](\mathbf{C}_3^\top \text{ReLu}(\mathbf{C}_2^\top(\boldsymbol{\eta}^{(l)} \otimes (\mathbf{C}_1^\top H^{(l-1)}))))[i,:]\|_2$$

$$= \left\|\sum_{i=1}^N \mathbf{C}_4^\top[w_l^*,i]\left(\sum_{j=1}^N \mathbf{C}_3^\top[i,j]\text{ReLu}(\mathbf{C}_2^\top(\boldsymbol{\eta}^{(l)} \otimes (\mathbf{C}_1^\top H^{(l-1)})))\right.\right.$$
$$\left.\left.[j,:]\right)[i,:]\right\|_2$$

$$\le \left\|\sum_{i=1}^N \mathbf{C}_4^\top[w_l^*,i]\left(\sum_{j=1}^N \mathbf{C}_3^\top[i,j](\mathbf{C}_2^\top(\boldsymbol{\eta}^{(l)} \otimes (\mathbf{C}_1^\top H^{(l-1)})))\right.\right.$$
$$\left.\left.[j,:]\right)[i,:]\right\|_2$$

$$= \left\|\sum_{i=1}^N \mathbf{C}_4^\top[w_l^*,i]\left(\sum_{j=1}^N \mathbf{C}_3^\top[i,j]\left(\sum_{k=1}^N \mathbf{C}_2^\top[j,k](\boldsymbol{\eta}^{(l)}\right.\right.\right.$$
$$\left.\left.\left.\otimes (\mathbf{C}_1^\top H^{(l-1)}))[k,:]\right)[j,:]\right)[i,:]\right\|_2$$

$$= \left\|\sum_{i=1}^N \mathbf{C}_4^\top[w_l^*,i]\left(\sum_{j=1}^N \mathbf{C}_3^\top[i,j]\left(\sum_{k=1}^N \mathbf{C}_2^\top[j,k]\right.\right.\right.$$
$$\left.\left.\left.((\mathbf{C}_1^\top H^{(l-1)})\mathbf{W}^{(l)}[k,:]))[j,:]\right)[i,:]\right\|_2$$

$$= \left\|\sum_{i=1}^N \mathbf{C}_4^\top[w_l^*,i]\left(\sum_{j=1}^N \mathbf{C}_3^\top[i,j]\left(\sum_{k=1}^N\right.\right.\right.$$
$$\left.\left.\left.\mathbf{C}_2^\top[j,k](\mathbf{W}^{(l)}(\sum_{m=1}^N \mathbf{C}_1^\top[k,m]H^{(l-1)}\mathbf{W}^{(l)}[m,:])))[j,:]\right)[i,:]\right\|_2$$

Since $\mathbf{C}_1^\top[k,:]$ has at most $M$ of value 1 entries. We have

$$\Phi_l \le \left\|\sum_{i=1}^N \mathbf{C}_4^\top[w_l^*,i]\left(\sum_{j=1}^N \mathbf{C}_3^\top[i,j]\left(\sum_{k=1}^N \mathbf{C}_2^\top[j,k](\mathbf{W}^{(l)}M\Phi_{l-1})\right.\right.\right.$$
$$\left.\left.\left.\right)[j,:]\right)[i,:]\right\|_2$$

Since $\mathbf{C}_2^\top[j,:]$ has at most $R$ non-zero entries, which can be bounded by 1 based on the definition of $\mathbf{C}_2$. We have

$$\Phi_l \le \|\sum_{i=1}^N \mathbf{C}_4^\top[w_l^*,i](R(\mathbf{W}^{(l)}M\Phi_{l-1}))[i,:]\|_2$$
$$\tag{5}$$
$$\le \|\mathbf{W}^{(l)}\|DRM\Phi_{l-1} \le (DRM)^l B\prod_{i=1}^l \|\mathbf{W}^{(i)}\|$$

Let $C = DRM$. The second step is calculating the upper bound of the variation in the model's output given the perturbed parameters.

Let $H_{\mathbf{w}+\mathbf{u}}^{(l)}(A)$ and $H_{\mathbf{w}}^{(l)}(A)$ be the $l$-layer output with parameter $\mathbf{w}$ and perturbed parameter $\mathbf{w} + \mathbf{u}$, respectively. For $l \in [L]$, let $\Delta_l = \|H_{\mathbf{w}+\mathbf{u}}^{(l)}(A) - H_{\mathbf{w}}^{(l)}(A)\|_2$. We define $\Psi_l = \max_{i \in [N]} \|\Delta_l[i,:]\|_2 = \max_i \|\hat{H}^{(l)}[i,:] - H^{(l)}[i,:]\|_2$, where $\hat{H}^{(l)}$ is perturbed model. Let $v_{(l)}^* = \arg\max_i \|\Delta_l[i,:]\|_2$. Therefore, we have

$$
\begin{aligned}
\Psi_{(l)} &= \max_i \|\hat{H}^{(l)}[i,:] - H^{(l)}[i,:]\|_2 \\
&= \max_i \left\| \mathbf{C}_4^\mathsf{T}\mathbf{C}_3^\mathsf{T}\mathrm{ReLu}\big(\mathbf{C}_2^\mathsf{T}\big((\boldsymbol{\eta}^{(l)}+\boldsymbol{\theta}^{(l)})\otimes(\mathbf{C}_1^\mathsf{T}\hat{H}^{(l-1)})\big)\big)[i,:] \right. \\
&\qquad \left. - \mathbf{C}_4^\mathsf{T}\mathbf{C}_3^\mathsf{T}\mathrm{ReLu}\big(\mathbf{C}_2^\mathsf{T}\big(\boldsymbol{\eta}^{(l)}\otimes(\mathbf{C}_1^\mathsf{T}H^{(l-1)})\big)\big)[i,:] \right\|_2 \\
&= \max_i \left\| \mathbf{C}_4^\mathsf{T}\mathbf{C}_3^\mathsf{T}\mathrm{ReLu}\big(\mathbf{C}_2^\mathsf{T}\big((\boldsymbol{\eta}^{(l)}+\boldsymbol{\theta}^{(l)})\otimes(\mathbf{C}_1^\mathsf{T}\hat{H}^{(l-1)})\big) \right. \\
&\qquad \left. - \mathbf{C}_2^\mathsf{T}\big(\boldsymbol{\eta}^{(l)}\otimes(\mathbf{C}_1^\mathsf{T}H^{(l-1)})\big)\big)[i,:] \right\|_2
\end{aligned}
\tag{6}
$$

$$
\begin{aligned}
&\leq \max_i \left\| \mathbf{C}_4^\mathsf{T}\mathbf{C}_3^\mathsf{T}\big(\mathbf{C}_2^\mathsf{T}\big((\boldsymbol{\eta}^{(l)}+\boldsymbol{\theta}^{(l)})\otimes(\mathbf{C}_1^\mathsf{T}\hat{H}^{(l-1)})\big) \right. \\
&\qquad \left. - \mathbf{C}_2^\mathsf{T}\big(\boldsymbol{\eta}^{(l)}\otimes(\mathbf{C}_1^\mathsf{T}H^{(l-1)})\big)\big)[i,:] \right\|_2 \\
&\qquad\qquad\qquad \text{(Lipschitz property of ReLu)} \\
&= \max_i \left\| \mathbf{C}_4^\mathsf{T}\mathbf{C}_3^\mathsf{T}\mathbf{C}_2^\mathsf{T}\big(\big((\boldsymbol{\eta}^{(l)}+\boldsymbol{\theta}^{(l)})\otimes(\mathbf{C}_1^\mathsf{T}\hat{H}^{(l-1)})\big) \right. \\
&\qquad \left. - \big(\boldsymbol{\eta}^{(l)}\otimes(\mathbf{C}_1^\mathsf{T}H^{(l-1)})\big)\big)[i,:] \right\|_2 \\
&= \max_i \left\| \mathbf{C}_4^\mathsf{T}\mathbf{C}_3^\mathsf{T}\mathbf{C}_2^\mathsf{T}\big(\big((\mathbf{C}_1^\mathsf{T}\hat{H}^{(l-1)})\big)(\mathbf{W}^{(l)}+\mathbf{U}^{(l)}) \right. \\
&\qquad \left. - \big((\mathbf{C}_1^\mathsf{T}H^{(l-1)})\mathbf{W}^{(l)}\big)\big)[i,:] \right\|_2 \\
&\leq \max_i \|\mathbf{C}_4^\mathsf{T}\mathbf{C}_3^\mathsf{T}\mathbf{C}_2^\mathsf{T}\mathbf{C}_1^\mathsf{T}\big((\hat{H}^{(l-1)}-H^{(l-1)})(\mathbf{W}^{(l)}+\mathbf{U}^{(l)})[i,:]\|_2 \\
&\qquad + \|\mathbf{C}_4^\mathsf{T}\mathbf{C}_3^\mathsf{T}\mathbf{C}_2^\mathsf{T}\mathbf{C}_1^\mathsf{T}\big(H^{(l-1)}\mathbf{U}^{(l)}\big)[i,:]\|_2 \\
&\leq \max_i \|\mathbf{C}_4^\mathsf{T}\mathbf{C}_3^\mathsf{T}\mathbf{C}_2^\mathsf{T}\mathbf{C}_1^\mathsf{T}\big(\Delta_l(\mathbf{W}^{(l)}+\mathbf{U}^{(l)})[i,:]\|_2 + C\Phi_{l-1}\|\mathbf{U}^{(l)}\| \\
&\leq C\Psi_{l-1}\|\mathbf{W}^{(l)}+\mathbf{U}^{(l)}\| + C\Phi_{l-1}\|\mathbf{U}^{(l)}\|
\end{aligned}
\tag{7}
$$

We let $a_{l-1} = C\|\mathbf{W}^{(l)}+\mathbf{U}^{(l)}\|$ and $b_{l-1} = C\Phi_{l-1}\|\mathbf{U}^{(l)}\|$. Then $\Psi_l \leq a_{l-1}\Psi_{l-1} + b_{l-1}$ for $l \in [L]$. Using recursive, we have

$$
\Psi_l \leq \prod_{i=1}^{l-1} a_i \Psi_1 + \sum_{m=1}^{l-1} b_m \Big(\prod_{n=m+1}^{l-1} a_n\Big)
$$

Since $\Psi_0 = 0$, we can simplified the Equation 6 as follows,

$$
\begin{aligned}
\Psi_l &\leq \sum_{i=0}^{l-1} b_i \Big(\prod_{j=i+1}^{l-1} a_j\Big) \\
&= \sum_{i=0}^{l-1} C\Phi_i \|\mathbf{U}^{(i+1)}\| \Big(\prod_{j=i+2}^{l} C\|\mathbf{W}^{(j)}+\mathbf{U}^{(j)}\|\Big) \\
&= \sum_{i=0}^{l-1} C^{l-i}\Phi_i \|\mathbf{U}^{(i+1)}\| \Big(\prod_{j=i+2}^{l} \|\mathbf{W}^{(j)}+\mathbf{U}^{(j)}\|\Big)
\end{aligned}
$$

$$
\begin{aligned}
&\leq \sum_{i=0}^{l-1} C^{l-i}\Big(C^i B \prod_{k=1}^{i}\|\mathbf{W}^{(k)}\|\Big)\|\mathbf{U}^{(i+1)}\|\Big(\prod_{j=i+2}^{l}\|\mathbf{W}^{(j)}+\mathbf{U}^{(j)}\|\Big) \\
&\leq B\sum_{i=0}^{l-1} C^l \Big(\prod_{k=1}^{i}\|\mathbf{W}^{(k)}\|\Big)\|\mathbf{U}^{(i+1)}\|\Big(\prod_{j=i+2}^{l}\|\mathbf{W}^{(j)}\|\big(1+\frac{1}{L+1}\big)\Big) \\
&= B\sum_{i=0}^{l-1} C^l \Big(\prod_{k=1}^{i+1}\|\mathbf{W}^{(k)}\|\Big)\frac{\|\mathbf{U}^{(i+1)}\|}{\|\mathbf{W}^{(i+1)}\|}\Big(\prod_{j=i+2}^{l}\|\mathbf{W}^{(j)}\|\big(1+\frac{1}{L+1}\big)\Big) \\
&= BC^l \Big(\prod_{i=1}^{l}\|\mathbf{W}^{(i)}\|\Big)\sum_{i=1}^{l}\frac{\|\mathbf{U}^{(i)}\|}{\|\mathbf{W}^{(i)}\|}\big(1+\frac{1}{L+1}\big)^{l-i}
\end{aligned}
$$

Finally, we need to consider the readout layer. We have

$$
\begin{aligned}
|\Delta_{L+1}|_2 &= \left\| \frac{1}{N}\mathbf{1}_N \hat{H}^{(L)}(\mathbf{W}^{(L+1)}+\mathbf{U}^{(L+1)}) \right. \\
&\qquad \left. - \frac{1}{N}\mathbf{1}_N H^{(L+1)}(\mathbf{W}^{(L+1)}) \right\|_2 \\
&\leq \frac{1}{N}\left\| \mathbf{1}_N(H^{(L+1)\prime}-H^{(L+1)})(\mathbf{W}^{(L+1)} \right. \\
&\qquad \left. + \mathbf{U}^{(L+1)}) \right\|_2 + \frac{1}{N}\|\mathbf{1}_N H^{(L+1)}\mathbf{U}^{L+1}\|_2 \\
&\leq \frac{1}{N}\|\mathbf{1}_N\Delta_L(\mathbf{W}^{(L+1)}+\mathbf{U}^{(L+1)})\|_2 + \frac{1}{N}\|\mathbf{1}_N H^{(L)}\mathbf{U}^{(L+1)}\|_2 \\
&\leq \frac{1}{N}\|\mathbf{W}^{(L+1)}+\mathbf{U}^{(L+1)}\|\Big(\sum_{i=1}^{N}|\Delta_{L+1}[i,:]|_2\Big) \\
&\qquad + \frac{1}{N}\|\mathbf{U}^{(L+1)}\|\Big(\sum_{i=1}^{N}|H^{(L)}[i,:]|_2\Big) \\
&\leq \Psi_L\|\mathbf{W}^{(L+1)}+\mathbf{U}^{(L+1)}\| + \Phi_L\|\mathbf{U}^{(L+1)}\| \\
&\leq BC^L\Big(\prod_{i=1}^{L+1}\|\mathbf{W}^{(i)}\|\Big)\big(1+\frac{1}{L+1}\big)^{L+1}\Big[\sum_{i=1}^{L}\frac{\|\mathbf{U}^{(i)}\|}{\|\mathbf{W}^{(i)}\|} \\
&\qquad \big(1+\frac{1}{L+1}\big)^{-i} + \frac{\|\mathbf{U}^{(L+1)}\|}{\|\mathbf{W}^{(L+1)}\|}\big(1+\frac{1}{L+1}\big)^{-(L+1)}\Big] \\
&\leq eBC^L\Big(\prod_{i=1}^{L+1}\|\mathbf{W}^{(i)}\|\Big)\Big[\sum_{i=1}^{L+1}\frac{\|\mathbf{U}^{(i)}\|}{\|\mathbf{W}^{(i)}\|}\Big]
\end{aligned}
$$

Therefore, we conclude the bound in Lemma 1.

## C.2 Proofs of Theorem 1

Given the perturbation bound in Lemma 1, to obtain the generalization bounds by applying Lemma 5, we need to design the prior P and posterior Q by satisfying the perturbation condition for every possible $\mathbf{w}$. Due to the homogeneity of ReLu, the perturbation bound will not change after weight normalization. We consider a transformation of UniGCN with the normalized weights $\tilde{\mathbf{W}}^{(i)} = \frac{\beta}{\|\mathbf{W}^{(i)}\|}\mathbf{W}^{(i)}$, where $\beta = (\prod_{i=1}^{L+1}\|\mathbf{W}^{(i)}\|)^{1/(L+1)}$. Hence we have the norm equal across layers, i.e., $\|\mathbf{W}^{(i)}\| = \beta$. Therefore, the space of $\mathbf{w}$ is presented by all possible values of $\beta$. According to the generalization bound in Lemma 5, we find that the space of $\beta$ can be partitioned into three parts such that each part admits a finite design of $P$ and $Q$ that meets the perturbation condition. To see

this, we first recall the generalization bound in Lemma 5 as follows.

$$\mathcal{L}_{\mathcal{D}}(f_{\mathbf{w}}) \leq \mathcal{L}_{S,\gamma}(f_{\mathbf{w}}) + 4\sqrt{\frac{\mathrm{KL}(\mathbf{w}^* + \mathbf{u} || \mathbf{w}) + \ln \frac{6m}{\delta}}{m-1}}, \qquad (8)$$

There are two terms in the right hand. Since the $\mathcal{L}_{\mathcal{D}}(f_{\mathbf{w}})$ is in range $[0, 1]$, we then consider three cases: 1) $\mathcal{L}_{S,\gamma}(f_{\mathbf{w}}) = 1$, 2) $\sqrt{\frac{\mathrm{KL}(\mathbf{w}^*+\mathbf{u}||\mathbf{w})+\ln \frac{6m}{\delta}}{m-1}} > 1$, and 3) $\mathcal{L}_{S,\gamma}(f_{\mathbf{w}}) + 4\sqrt{\frac{\mathrm{KL}(\mathbf{w}^*+\mathbf{u}||\mathbf{w})+\ln \frac{6m}{\delta}}{m-1}} \in [0, 1]$. Significantly, the equation will be directly satisfied for values of $\beta$ that meet the first two cases. Therefore, for these values of $\beta$, we only need to specify one group of $P$ and $Q$ to satisfy the perturbation condition outlined in Lemma 5. However, when dealing with $\beta$ values that fall within the third case, it is necessary to determine the specific values of $P$ and $Q$ that satisfy the perturbation condition for each $\beta$. Altogether, we separate the proof into three parts for three ranges of $\beta$.

**First case.** We start from the first case with $\mathcal{L}_{S,\gamma}(f_{\mathbf{w}}) = 1$. In the proof of UniGCN's perturbation bound in Lemma 1, the maximum node representation can be bounded by $(DRM)^L B \prod_{i=1}^{L+1} \|\mathbf{W}^{(i)}\|$. If $\beta < \left(\frac{\gamma}{2B(DRM)^L}\right)^{1/L+1}$, then for any input $A$ and any $j \in [N]$, we have $\mathrm{UniGCN}(A)[j] \leq \frac{\gamma}{2}$. To see this, we have

$$\mathrm{UniGCN}_{\mathbf{w}}(A) = \mathbf{W}^{(L+1)} H^{(L)}$$

$$\leq \|\mathbf{W}^{(L+1)}\| \max_i \|H^{(L)}[i,:]\|_2 = \|\mathbf{W}^{(L+1)}\| \Phi_L$$

$$= B(DRM)^L \prod_{i=1}^{L+1} \|\mathbf{W}^{(i)}\| \leq B(DRM)^L \beta^{L+1} \leq \frac{\gamma}{2}$$

Therefore, by the definition in Equation 2, we always have $\mathcal{L}_{S,\gamma} = 1$. Then, we design distributions $P$ and $Q$ to satisfy the perturbation condition specified in Lemma 5 by using the perturbation bound in Lemma 1 such that

$$\Pr_{\mathbf{u}}\left[\max_{A \in \mathcal{A}} \|\mathrm{UniGCN}_{\mathbf{w}+\mathbf{u}}(A) - \mathrm{UniGCN}_{\mathbf{w}}(A)\|_2 < \frac{\gamma}{4}\right] \geq \frac{1}{2}.$$

Following that, we consider $P$ and $Q$ both follow the multivariate Gaussian distributions $\mathcal{N}(0, \sigma^2 I)$. Based on the work in [57], we have following bound for matrix $\mathbf{U}^i \in \mathbb{R}^{h \times h}$ where $\mathbf{U}^i \sim \mathcal{N}(0, \sigma^2 I)$ with $h \in \mathbb{Z}^+$,

$$\Pr_{\mathbf{U}^i \sim \mathcal{N}(0,\sigma^2 I)}\left[\|\mathbf{U}^i\| > t\right] \leq 2he^{-t^2/2h\sigma^2} \qquad (9)$$

One can obtain the spectral norm bound as follows by taking the union bound over all layers.

$$\Pr_{\mathbf{U}^i \sim Q}\left[\|\mathbf{U}^1\| \leq t \,\&\, \|\mathbf{U}^2\| \leq t \ldots \&\, \|\mathbf{U}^{L+1}\| \leq t\right]$$

$$\geq 1 - \sum_{i=1}^{L+1} \Pr\left[\|\mathbf{U}^i\|\right]$$

$$\geq 1 - 2(L_1)he^{\frac{-t^2}{2h\sigma^2}}$$

Setting $1 - 2(L+1)he^{\frac{-t^2}{2h\sigma^2}} = \frac{1}{2}$, we have $t = \sigma\sqrt{2h\ln(4h(L+1))}$. This implies that the probability that the spectral norm of the perturbation of any layer no larger than $\sigma\sqrt{2h\ln(4h(L+1))}$ is at least

$\frac{1}{2}$. Combining with Lemma 1, let $\mathcal{I} = eB(DRM)^L$, we have

$$\max_{A \in \mathcal{A}} \|\mathrm{UniGCN}_{\mathbf{w}+\mathbf{u}}(A) - \mathrm{UniGCN}_{\mathbf{w}}(A)\|_2$$

$$\leq \mathcal{I} \prod_{i=1}^{L_1} \|\mathbf{W}^{(i)}\| \left(\sum_{i=1}^{L_1} \frac{\|\mathbf{U}^{(i)}\|}{\|\mathbf{W}^{(i)}\|}\right) \leq \mathcal{I} \beta^{L+1}\left(\sum_{i=1}^{L+1} \frac{\|\mathbf{U}^{(i)}\|}{\beta}\right)$$

$$= \mathcal{I} \beta^L \left(\sum_{i=1}^{L_1} \|\mathbf{U}^{(i)}\|\right) \mathcal{I} \beta^L (L+1)\sigma\sqrt{2h\ln(4h(L+1))} < \frac{\gamma}{4},$$

By letting $\sigma = \frac{\gamma}{4\mathcal{I}\beta^L(L+1)\sqrt{2h\ln(4h(L+1))}}$, we hold the perturbation condition in Lemma 5. Remember that we have one conditions in Lemma 1 which is $\frac{\|\mathbf{U}^{(i)}\|}{\|\mathbf{W}^{(i)}\|} < \frac{1}{L_1}$ for $i \in [L_1]$. This assumption also holds when $\beta < \left(\frac{\gamma}{2B(DRM)^L}\right)^{1/L+1}$, where

$$\|\mathbf{U}\|^{(i)} \leq \sigma\sqrt{2h\ln 4h(L+1)} \leq \frac{\beta}{L+1} \implies \frac{\gamma}{4BD^LR^LM^L} \leq \beta^{L+1}$$

Therefore, we can calculate the KL divergence between $\mathbf{w} + \mathbf{u}$ and $P$.

$$\mathrm{KL}(\mathbf{w} + \mathbf{u} || P) \leq \frac{|\mathbf{w}|^2}{2\sigma^2}$$

$$\leq O\left(\frac{B^2 D^L R^L M^L L^2 h\ln(hL) \prod_{i=1}^{L+1} \|\mathbf{W}^{(i)}\|^2}{\gamma^2} \sum_{i=1}^{L+1} \frac{\|\mathbf{W}^{(i)}\|_F^2}{\|\mathbf{W}^{(i)}\|^2}\right).$$

Therefore, following the Lemma 5, we have

LEMMA 6. *Given a* $\mathrm{UniGCN}_{\mathbf{w}}(A) : \mathcal{A} \to \mathbb{R}^C$ *with* $L + 1$ *layers parameterized by* $\mathbf{w}$. *Given the training set of size* $m$, *for each* $\delta, \gamma > 0$, *for any* $\mathbf{w}$ *such that* $\beta < \left(\frac{\gamma}{2B(DRM)^L}\right)^{1/L+1}$, *we have*

$$\mathcal{L}_{\mathcal{D}}(\mathrm{UniGCN}_{\mathbf{w}}) \leq \mathcal{L}_{S,\gamma}(\mathrm{UniGCN}_{\mathbf{w}}) + \qquad (10)$$

$$O\left(\sqrt{\frac{L^2 B^2 h\ln(Lh)(RMD)^L \mathcal{W}_1 \mathcal{W}_2 + \log\frac{m}{\sigma}}{\gamma^2 m}}\right), \qquad (11)$$

*where* $\mathcal{W}_1 = \prod_{i=1}^{L+1} \|\mathbf{W}^{(i)}\|^2$ *and* $\mathcal{W}_2 = \sum_{i=1}^{L+1} \frac{\|\mathbf{W}^{(i)}\|_F^2}{\|\mathbf{W}^{(i)}\|^2}$.

**Second Case.** We then consider the values of $\beta$ that satisfy $\sqrt{\frac{\mathrm{KL}(\mathbf{w}^*+\mathbf{u}||\mathbf{w})+\ln \frac{6m}{\delta}}{m-1}} > 1$. In order to obtain the KL term, we first need to construct the distribution of $P$ and $Q$. Following the strategy used in the first case, We choose $P$ and $Q$ both following the multivariate Gaussian distributions $\mathcal{N}(0, \sigma^2 I)$. And to satisfy the perturbation condition in Lemma 5, we have

$$\sigma = \frac{\gamma}{4\mathcal{I}\beta^L(L+1)\sqrt{2h\ln(4h(L+1))}}.$$

Therefore, when calculating the term inside the big-O notation in the Equation 10, if $\beta > \left(\frac{\gamma\sqrt{m}}{2B(DRM)^L}\right)^{1/L+1}$ we have

$$\sqrt{\frac{L^2 B^2 h\ln(Lh)(RMD)^L \prod_{i=1}^{L+1} \|\mathbf{W}^{(i)}\|_2^2 \sum_{i=1}^{L+1} \frac{\|\mathbf{W}^{(i)}\|_F^2}{\|\mathbf{W}^{(i)}\|^2} + \log\frac{mL}{\sigma}}{\gamma^2 m}}$$

$$> \sqrt{\frac{L^2 h\ln(Lh)(RMD)^L \sum_{i=1}^{L+1} \frac{\|\mathbf{w}^{(i)}\|_F^2}{\|\mathbf{w}^{(i)}\|^2}}{4}} \geq 1,$$

where $\|\mathbf{W}^{(i)}\|_F^2 > \|\mathbf{W}^{(i)}\|^2$ and we typically choose $h \geq 2$ and $L \geq 2$. Therefore, we have

**LEMMA 7.** *Given a* $\mathrm{UniGCN}_{\mathbf{w}}(A) : \mathcal{A} \to \mathbb{R}^C$ *with* $L + 1$ *layers parameterized by* $\mathbf{w}$. *Given the training set of size* $m$, *for each* $\delta, \gamma > 0$, *for any* $\mathbf{w}$ *such that* $\beta > \left(\frac{\gamma\sqrt{m}}{2B(DRM)^L}\right)^{1/L+1}$, *we have*

$$\mathcal{L}_{\mathcal{D}}(\mathrm{UniGCN}_{\mathbf{w}}) \leq \mathcal{L}_{S,\gamma}(\mathrm{UniGCN}_{\mathbf{w}}) +$$

$$O\left(\sqrt{\frac{L^2 B^2 h \ln{(Lh)}(RMD)^L \mathcal{W}_1 \mathcal{W}_2 + \log{\frac{m}{\sigma}}}{\gamma^2 m}}\right),$$

*where* $\mathcal{W}_1 = \prod_{i=1}^{L+1} \|\mathbf{W}^{(i)}\|^2$ *and* $\mathcal{W}_2 = \sum_{i=1}^{L+1} \frac{\|\mathbf{W}^{(i)}\|_F^2}{\|\mathbf{W}^{(i)}\|^2}$.

**Third Case.** We now consider the case where $\beta$ in the following range, denoted as $\mathcal{B}$.

$$\left\{\beta \mid \beta \in \left[\left(\frac{\gamma}{2B(DRM)^L}\right)^{1/L+1}, \left(\frac{\gamma\sqrt{m}}{2B(DRM)^L}\right)^{1/L+1}\right]\right\}.$$

We observe that if $\beta$ falls in a grid of size $\frac{1}{2L+2}\left(\frac{\gamma}{2B(DRM)^L}\right)^{1/L+1}$, one group of $P$ and $Q$ is able to make the perturbation condition in Lemma 5 satisfied. To see this, for a $\beta \in \mathcal{B}$, we assume a $\tilde{\beta}$ such that $\tilde{\beta} \in \mathcal{B}$ and $|\beta - \tilde{\beta}| \leq \frac{1}{L+1}\beta$. Then we have

$$|\beta - \tilde{\beta}| \leq \frac{1}{L+1}\beta$$

$$\implies (1 - \frac{1}{L+1})\beta \leq \tilde{\beta} \leq (1 + \frac{1}{L+1})\beta$$

$$\implies (1 - \frac{1}{L+1})^{L+1}\beta^{L+1} \leq \tilde{\beta}^{L+1} \leq (1 + \frac{1}{L+1})^{L+1}\beta^{L+1} \quad (12)$$

$$\implies \frac{1}{e}\beta^{L+1} \leq \tilde{\beta}^{L+1} \leq e\beta^{L+1}$$

Given $\tilde{\beta}$, suppose that $\tilde{\mathbf{w}}$ and $\tilde{\mathbf{u}}$ are the corresponding parameters and perturbation in UniGCN. Therefore, to satisfy the perturbation condition, we have

$$\max_{A \in \mathcal{A}} \|\mathrm{UniGCN}_{\tilde{\mathbf{w}}+\tilde{\mathbf{u}}}(A) - \mathrm{UniGCN}_{\tilde{\mathbf{w}}}(A)\|_2$$

$$\leq \mathcal{I} \prod_{i=1}^{L+1} \|\tilde{\mathbf{W}}^{(i)}\| \left(\sum_{i=1}^{L_1} \frac{\|\tilde{\mathbf{U}}^{(i)}\|}{\|\tilde{\mathbf{W}}^{(i)}\|}\right) \leq \mathcal{I}\tilde{\beta}^{L+1}\left(\sum_{i=1}^{L+1} \frac{\|\tilde{\mathbf{U}}^{(i)}\|}{\tilde{\beta}}\right)$$

$$= \mathcal{I}\tilde{\beta}^L\left(\sum_{i=1}^{L+1} \|\tilde{\mathbf{U}}^{(i)}\|\right) \leq e\mathcal{I}\beta^L(L+1)\tilde{\sigma}\sqrt{2h\ln{(4h(L+1))}} \leq \frac{\gamma}{4},$$

where we let $\tilde{\sigma} \leq \frac{\gamma}{4e\mathcal{I}\beta^L(L+1)\sqrt{2h\ln{(4h(L+1))}}}$, to make the perturbation condition is satisfied. Same for $\beta$, we have

$$\max_{A \in \mathcal{A}} \|\mathrm{UniGCN}_{\tilde{\mathbf{w}}+\tilde{\mathbf{u}}}(A) - \mathrm{UniGCN}_{\tilde{\mathbf{w}}}(A)\|$$

$$\leq \mathcal{I} \prod_{i=1}^{L+1} \|\tilde{\mathbf{W}}^{(i)}\| \left(\sum_{i=1}^{L_1} \frac{\|\tilde{\mathbf{U}}^{(i)}\|}{\|\tilde{\mathbf{W}}^{(i)}\|}\right)$$

$$\leq \mathcal{I}\beta^L(L+1)\sigma\sqrt{2h\ln{(4h(L+1))}} \leq \frac{\gamma}{4},$$

where we let $\sigma \leq \frac{\gamma}{4\mathcal{I}\beta^L(L+1)\sqrt{2h\ln{(4h(L+1))}}}$ to satisfy the perturbation condition. We find that when $\beta$ and $\tilde{\beta}$ satisfying $|\beta - \tilde{\beta}| \leq \frac{1}{L+1}\beta$, they can share same $\sigma$ with value $\frac{\gamma}{4e\mathcal{I}\beta^L(L+1)\sqrt{2h\ln{(4h(L+1))}}}$ and obtain same generalization bound by simply apply the Lemma 5. Therefore, if we can find a covering size of $\mathcal{B}$ within the grid

$\frac{1}{2L+2}\left(\frac{\gamma}{2B(DRM)^L}\right)^{1/L+1}$, then we can get a bound which holds for all $\beta \in \mathcal{B}$. The grid size is given by $|\beta - \tilde{\beta}| \leq \frac{1}{L+1}\beta$ hold is $|\beta - \tilde{\beta}| \leq \frac{1}{L+1}\left(\frac{\gamma}{2B(DRM)^L}\right)^{1/L+1}$. Hence, dividing the range of $\mathcal{B}$ by the grid size, we have the covering size $n = ((\sqrt{m}DR)^{\frac{1}{L+1}} - 1)(2L+1)$. We denote the event of the generalization bound in one grid as $E_i$. We then have

$$\Pr[E_1 \& \ldots \& E_n] = 1 - \Pr[\exists_i, \neg E_i] \geq 1 - \sum_i^n \Pr[\neg E_i] \geq 1 - n\delta.$$

We obtain the following lemma within third case.

**LEMMA 8.** *Given a* $\mathrm{UniGCN}_{\mathbf{w}}(A) : \mathcal{A} \to \mathbb{R}^C$ *with* $L + 1$ *layers parameterized by* $\mathbf{w}$. *Given the training set of size* $m$, *for each* $\delta, \gamma > 0$, *with probability at least* $1 - \delta$ *over the training set of size* $m$, *for any* $\mathbf{w}$ *such that* $\beta \in \mathcal{B}$, *we have*

$$\mathcal{L}_{\mathcal{D}}(\mathrm{UniGCN}_{\mathbf{w}}) \leq \mathcal{L}_{S,\gamma}(\mathrm{UniGCN}_{\mathbf{w}}) +$$

$$O\left(\sqrt{\frac{L^2 B^2 h \ln{(Lh)}(RMD)^L \mathcal{W}_1 \mathcal{W}_2 + \log{\frac{mLDR}{\sigma}}}{\gamma^2 m}}\right),$$

*where* $\mathcal{W}_1 = \prod_{i=1}^{L+1} \|\mathbf{W}^{(i)}\|^2$ *and* $\mathcal{W}_2 = \sum_{i=1}^{L+1} \frac{\|\mathbf{W}^{(i)}\|_F^2}{\|\mathbf{W}^{(i)}\|^2}$.

Altogether, combining three cases, we conclude the theorem 1.

## C.3 Proof of Lemma 2

Similar to the proof of Lemma 1, this proof includes two parts. We add perturbation $u$ to the weight $w$ and denote the perturbed weights by $\mathbf{W}_i^{(l)} + \mathbf{U}_i^{(l)}$ where $i \in \{1, 2, 3, 4\}$. We can drive an upper bound on the $l_2$ norm of the maximum tuple representation among each node in the layer. For $l \in [L]$, let

$$\bar{w}_l^* = \arg\max_{i \in (N+K)} |\bar{H}^{(l)}[i, :]|_2, \qquad \bar{\Phi}_l = \|H^{(l')}[\bar{w}_l^*, :]\|_2,$$

$$w_l^* = \arg\max_{i \in (N+K)} |H^{(l)}[i, :]|_2, \text{ and } \Phi_l = \|H^{(l)}[w_l^*, :]\|_2.$$

We then have

$$\bar{\Phi}_l = \|\mathrm{ReLu}\left(\boldsymbol{\eta}_2^{(l)} \otimes \left(C_e\left(\mathrm{ReLu}\left(\boldsymbol{\eta}_1^{(l)} \otimes H^{(l-1)}\right)\right)\right)\right)[\bar{w}_l^*, :]\|_2$$

$$= \|\mathrm{ReLu}\left(\boldsymbol{\eta}_2^{(l)} \otimes \left(C_e\left(\mathrm{ReLu}\left(\boldsymbol{\eta}_1^{(l)} \otimes H^{(l-1)}\right)\right)\right)\right)[\bar{w}_l^*, :]\|_2$$

$$\leq \|\boldsymbol{\eta}_2^{(l)} \otimes \left(C_e\left(\mathrm{ReLu}\left(\boldsymbol{\eta}_1^{(l)} \otimes H^{(l-1)}\right)\right)\right)[\bar{w}_l^*, :]\|_2$$

$$= \|\left(C_e\left(\mathrm{ReLu}\left(\boldsymbol{\eta}_1^{(l)} \otimes H^{(l-1)}\right)\right)\right)\mathbf{W}_2^{(l)}[\bar{w}_l^*, :]\|_2$$

$$\leq \|\left(C_e\left(\boldsymbol{\eta}_1^{(l)} \otimes H^{(l-1)}\mathbf{W}_2^{(l)}[\bar{w}_l^*, :, :]\right)\right)\|_2$$

$$\leq \|C_e\left(\boldsymbol{\eta}_1^{(l)} \otimes H^{(l-1)}\mathbf{W}_2^{(l)}\right)[\bar{w}_l^*, :, :]\|_2$$

$$\leq \|\sum_{k=1}^{N+K} C_e[\bar{w}_l^*, k]\left(\boldsymbol{\eta}_1^{(l)} \otimes H^{(l-1)}\mathbf{W}_2^{(l)}[k, :]\right)\|_2$$

$$= \|\sum_{k=1}^{N+K} C_e[\bar{w}_l^*, k]\left(H^{(l-1)}\mathbf{W}_1^{(l)}\mathbf{W}_2^{(l)}[k, :]\right)\|_2$$

$$(13)$$

$$\leq \| \sum_{k=1}^{N+K} C_e[\bar{w}_l^*, k] (\Phi_{l-1} \mathbf{W}_1^{(l)} \mathbf{W}_2^{(l)}) \|_2$$

$$\leq \| (P+1) (\Phi_{l-1} \mathbf{W}_1^{(l)} \mathbf{W}_2^{(l)}) \|_2$$

$$\leq (P+1) \|\mathbf{W}_1^{(l)}\| \|\mathbf{W}_2^{(l)}\| \Phi_{l-1} \quad (14)$$

Similarly, we have

$$\Phi_l = \|\mathrm{ReLu}\big(\boldsymbol{\eta}_4^{(l)} \otimes \big(C_v(\mathrm{ReLu}(\boldsymbol{\eta}_3^{(l)} \otimes H^{(l-1)})))\big)[w_l^*, :]\|_2 \quad (15)$$

$$\leq \| \sum_{k=1}^{N+K} C_v[w_l^*, k] (H^{(l-1)} \mathbf{W}_3^{(l)} \mathbf{W}_4^{(l)}[k, :]) \|_2$$

$$\leq \| \sum_{k=1}^{N+K} C_v[w_l^*, k] (\Phi_{l-1} \mathbf{W}_3^{(l)} \mathbf{W}_4^{(l)}) \|_2$$

$$\leq \| (M+1) (\bar{\Phi}_{l-1} \mathbf{W}_3^{(l)} \mathbf{W}_4^{(l)}) \|_2$$

$$\leq (M+1) \|\mathbf{W}_3^{(l)}\| \|\mathbf{W}_4^{(l)}\| \bar{\Phi}_l \quad (16)$$

Combined with Equation 13, we have

$$\Phi_l \leq (M+1)(P+1) \|\mathbf{W}_4^{(l)}\| \|\mathbf{W}_3^{(l)}\| \|\mathbf{W}_2^{(l)}\| \|\mathbf{W}_1^{(l)}\| \Phi_{l-1}$$

$$\leq (C_A)^l B \prod_{i=1}^{l} \|\mathbf{W}_4^{(i)}\| \|\mathbf{W}_3^{(i)}\| \|\mathbf{W}_2^{(i)}\| \|\mathbf{W}_1^{(i)}\|, \quad (17)$$

where $C_A = (M+1)(P+1)$. Let $\zeta_l = \|\mathbf{W}_4^{(l)}\| \|\mathbf{W}_3^{(l)}\| \|\mathbf{W}_2^{(l)}\| \|\mathbf{W}_1^{(l)}\|$. Given the input $A$, we use $\Delta_l$ to denote the change in the output. We define $\Psi_l = \max_{i \in (N+K)} |\Delta_l[i, :]|_2 = \max_i |\hat{H}^{(l)}[i, :] - H^{(l)}[i, :]|_2$, where $\hat{H}^{(l)}$ is perturbed model. Let $v_{(l)}^* = \arg\max_i |\Delta_{(l)}[i, :]|_2$. We define $\boldsymbol{\theta}_1^{(l)}, \boldsymbol{\theta}_2^{(l)}, \boldsymbol{\theta}_3^{(l)} \in \mathbb{R}^{(N+K) \times d_{l-1} \times d_{l-1}}$ and $\boldsymbol{\theta}_4^{(l)} \in \mathbb{R}^{(N+K) \times d_{l-1} \times d_l}$ where $\boldsymbol{\theta}_i^{(l)}[k, :] = \boldsymbol{\theta}_i^{(l)}[j, :] = \mathbf{U}_i^{(l)}$ when $j \neq k$, where $i \in \{1, 2, 3, 4\}$. We then bound the max change of the tuple representation before the readout layers.

$$\Psi_l = \max_i \|\hat{H}^{(l)}[i, :] - H^{(l)}[i, :]\|_2$$

$$= \Big\| \mathrm{ReLu}\big((\boldsymbol{\eta}_4^{(l)} + \boldsymbol{\theta}_4^{(l)}) \otimes (C_v^\mathsf{T}(\mathrm{ReLu}((\boldsymbol{\eta}_3^{(l)} + \boldsymbol{\theta}_3^{(l)}) \otimes \hat{H}^{(l)}))))$$

$$[v_l^*, :] - \mathrm{ReLu}\big(\boldsymbol{\eta}_4^{(l)} \otimes (C_v^\mathsf{T}(\mathrm{ReLu}(\boldsymbol{\eta}_3^{(l)} \otimes \bar{H}^{(l)}))))[v_l^*, :] \Big\|_2$$

$$\leq \Big\| \big((\boldsymbol{\eta}_4^{(l)} + \boldsymbol{\theta}_4^{(l)}) \otimes (C_v^\mathsf{T}(\mathrm{ReLu}((\boldsymbol{\eta}_3^{(l)} + \boldsymbol{\theta}_3^{(l)}) \otimes \hat{H}^{(l)}))))[v_l^*, :]$$

$$- \big(\boldsymbol{\eta}_4^{(l)} \otimes (C_v^\mathsf{T}(\mathrm{ReLu}(\boldsymbol{\eta}_3^{(l)} \otimes \bar{H}^{(l)}))))[v_l^*, :] \Big\|_2$$

$$\text{(Lipschitz property of ReLu)}$$

$$\leq \Big\| \big((C_v^\mathsf{T} \hat{H}^{(l')}[v_l^*, :]) (\mathbf{W}_2^{(l)} + \mathbf{U}_2^{(l)})(\mathbf{W}_1^{(l)} + \mathbf{U}_1^{(l)}) -$$

$$(C_v^\mathsf{T} \bar{H}^{(l)}[v_l^*, :]) \mathbf{W}_2^{(l)} \mathbf{W}_1^{(l)}) \Big\|_2$$

Let us replace the $\hat{H}^{(l)}$ to $\hat{H}^{(l-1)}$ and $\bar{H}^{(l)}$ to $H^{(l-1)}$, we have

$$\Psi_l \leq \Big\| \big((C_v^\mathsf{T} C_e^\mathsf{T} \hat{H}^{(l-1)}[v_l^*, :])(\mathbf{W}_4^{(l)} + \mathbf{U}_4^{(l)})(\mathbf{W}_3^{(l)} + \mathbf{U}_3^{(l)})$$

$$(\mathbf{W}_2^{(l)} + \mathbf{U}_2^{(l)})(\mathbf{W}_1^{(l)} + \mathbf{U}_1^{(l)})$$

$$- (C_v^\mathsf{T} C_e^\mathsf{T} H^{(l-1)}[v_l^*, :]) \mathbf{W}_4^{(l)} \mathbf{W}_3^{(l)} \mathbf{W}_2^{(l)} \mathbf{W}_1^{(l)}) \Big\|_2$$

$$\leq \Big\| \big((C_v^\mathsf{T} C_e^\mathsf{T} \hat{H}^{(l-1)}[v_l^*, :]) - (C_v^\mathsf{T} C_e^\mathsf{T} H^{(l-1)}[v_l^*, :]))$$

$$(\mathbf{W}_4^{(l)} + \mathbf{U}_4^{(l)})(\mathbf{W}_3^{(l)} + \mathbf{U}_3^{(l)})(\mathbf{W}_2^{(l)} + \mathbf{U}_2^{(l)})(\mathbf{W}_1^{(l)} + \mathbf{U}_1^{(l)})$$

$$+ (C_v^\mathsf{T} C_e^\mathsf{T} H^{(l-1)}[v_l^*, :]) \mathbf{U}_4^{(l)} \mathbf{U}_3^{(l)} \mathbf{U}_2^{(l)} \mathbf{U}_1^{(l)} \Big\|_2 + \frac{14}{4L+1} \Phi_{l-1} \zeta_l$$

$$\leq (M+1)(P+1)\Psi_{l-1} \|\mathbf{W}_4^{(l)} + \mathbf{U}_4^{(l)}\| \|\mathbf{W}_3^{(l)} + \mathbf{U}_3^{(l)}\|$$

$$\|\mathbf{W}_2^{(l)} + \mathbf{U}_2^{(l)}\| \|\mathbf{W}_1^{(l)} + \mathbf{U}_1^{(l)}\|$$

$$+ (M+1)(P+1)\Phi_{l-1} \|\mathbf{U}_4^{(l)}\| \|\mathbf{U}_3^{(l)}\| \|\mathbf{U}_2^{(l)}\| \|\mathbf{U}_1^{(l)}\|$$

$$+ \frac{14}{4L+1}(M+1)(P+1)\Phi_{l-1} \zeta_l$$

To simplify, we let

$$\lambda_l = \|\mathbf{W}_4^{(l)} + \mathbf{U}_4^{(l)}\| \|\mathbf{W}_3^{(l)} + \mathbf{U}_3^{(l)}\| \|\mathbf{W}_2^{(l)} + \mathbf{U}_2^{(l)}\| \|\mathbf{W}_1^{(l)} + \mathbf{U}_1^{(l)}\|$$

$$\kappa_l = \|\mathbf{U}_4^{(l)}\| \|\mathbf{U}_3^{(l)}\| \|\mathbf{U}_2^{(l)}\| \|\mathbf{U}_1^{(l)}\|$$

We define $a_{l-1} = C_A \lambda_l$ and $b_{l-1} = C_A \Phi_{l-1} \kappa_l + \frac{14}{4L+1} C_A \Phi_{l-1} \zeta_l$. Then $\Psi_l \leq a_{l-1} \Psi_{l-1} + b_{l-1}$ for $l \in [L]$. Using recursive, we have $\Psi_l \leq \prod_{i=1}^{l-1} a_i \Psi_0 + \sum_{i=0}^{l-1} b_i \big(\prod_{j=i+1}^{l-1} a_j\big)$. In addition, we let $c = \frac{14}{4L+1}$. Since $\Psi_0 = 0$, we have

$$\Psi_l \leq \sum_{i=0}^{l-1} b_i \big(\prod_{j=i+1}^{l-1} a_j\big)$$

$$= B \sum_{i=0}^{l-1} \big(C_A \kappa_{i+1} \Phi_i + c C_A \zeta_{i+1} \Phi_i\big)\big(\prod_{j=i+1}^{l-1} C_A \lambda_{j+1}\big)$$

$$= B \sum_{i=0}^{l-1} \big(C_A^{i+1} \kappa_{i+1}(\prod_{j=1}^{i} \zeta_j) + c C_A^{i+1} \zeta_{i+1}(\prod_{j=1}^{i} \zeta_j)\big)\big(\prod_{j=i+2}^{l} C_A \lambda_j\big)$$

$$= B \sum_{i=0}^{l-1} \big(C_A^{i+1}(\prod_{j=1}^{i} \zeta_j) + c C_A^{i+1} \frac{\zeta_{i+1}}{\kappa_{i+1}}(\prod_{j=1}^{i} \zeta_j)\big)(\kappa_{i+1})\big(\prod_{j=i+2}^{l} C_A \lambda_j\big)$$

$$= B \sum_{i=0}^{l-1} \big(C_A^{i+1}(\prod_{j=1}^{i+1} \zeta_j) + c C_A^{i+1} \frac{\zeta_{i+1}}{\kappa_{i+1}}(\prod_{j=1}^{i+1} \zeta_j)\big)\big(\frac{\kappa_{i+1}}{\zeta_{i+1}}\big)\big(\prod_{j=i+2}^{l} C_A \lambda_j\big) \quad (18)$$

Since $\|\mathbf{U}\| \leq \frac{1}{4L+1} \|\mathbf{W}\|$, we have $\lambda_l \leq (1 + \frac{1}{4L+1})^4 \zeta_l$. Therefore

$$\Psi_l \leq B \sum_{i=0}^{l-1} \big(C_A^{i+1}(\prod_{j=1}^{i+1} \zeta_j) + c C_A^{i+1} \frac{\zeta_{i+1}}{\kappa_{i+1}}(\prod_{j=1}^{i+1} \zeta_j)\big)\big(\frac{\kappa_{i+1}}{\zeta_{i+1}}\big)\big(\prod_{j=i+2}^{l} \zeta_j\big)$$

$$\cdot \big(C_A^{l-i-2}(1 + \frac{1}{4L+1})^{4(l-i-2)}\big)$$

$$= B C_A^{l-1}(\prod_{i=1}^{l} \zeta_i) \sum_{i=1}^{l} \big(\frac{\kappa_i}{\zeta_i} + c\big)(1 + \frac{1}{4L+1})^{4(l-i-1)}$$

Since $\frac{\kappa_i}{\zeta_i} \leq (\frac{1}{4L+1})^4 < c$ with $L \geq 1$, therefore we have

$$\Psi_l \leq 2c B C_A^{l-1}(\prod_{i=1}^{l} \zeta_i) \sum_{i=1}^{l}(1 + \frac{1}{4L+1})^{4(l-i-1)}$$

$$\leq 2c e B l C_A^{l-1}(\prod_{i=1}^{l} \zeta_i) \qquad ((1 + \frac{1}{4L+1})^{4L+1} \leq e \text{ and } C_A > 1)$$

The last step is finding the readout layer's final perturbation bound. Let $N' = N + K$, then we have

$$|\Delta_{L+1}|_2 = \left\| \frac{1}{N'} \mathbf{1}_{N'} \hat{H}^{(L)} (\mathbf{W}^{(L+1)} + \mathbf{U}^{(L+1)}) \right. \tag{19}$$

$$\left. - \frac{1}{N'} \mathbf{1}_{N'} H^{(L)} (\mathbf{W}^{(L+1)}) \right\|_2$$

$$\leq \frac{1}{N'} \left\| \mathbf{1}_{N'} (H^{(L+1)'} - H^{(L+1)}) (\mathbf{W}^{(L+1)} \right.$$

$$\left. + \mathbf{U}^{(L+1)}) \right|_2 + \frac{1}{N'} \left| \mathbf{1}_{N'} H^{(L+1)} \mathbf{U}^{L+1} \right\|_2$$

$$\leq \frac{1}{N'} \left\| \mathbf{1}_{N'} \Delta_L (\mathbf{W}^{(L+1)} + \mathbf{U}^{(L+1)}) \right|_2 \tag{20}$$

$$+ \frac{1}{N'} \left| \mathbf{1}_{N'} H^{(L)} \mathbf{U}^{(L+1)} \right\|_2$$

$$\leq \frac{1}{N'} \| \mathbf{W}^{(L+1)} + \mathbf{U}^{(L+1)} \| (\sum_{i=1}^{N'} |\Delta_{L+1}[i,:]|_2)$$

$$+ \frac{1}{N'} \| \mathbf{U}^{(L)} \| (\sum_{i=1}^{N'} |H^{(L)}[i,:]|_2)$$

$$\tag{21}$$

$$\leq \Psi_L \| \mathbf{W}^{(L+1)} + \mathbf{U}^{(L+1)} \| + \Phi_L \| \mathbf{U}^{(L+1)} \|$$

$$\leq 2eBL(1 + \frac{1}{4L+1}) C_A^L (\prod_{i=1}^L \zeta_i) \| \mathbf{W}^{(L+1)} \| (c + \frac{\| \mathbf{U}^{(L+1)} \|}{\| \mathbf{W}^{(L+1)} \|})$$

$$\leq 2eBL(1 + \frac{1}{4L+1}) C_A^L (\prod_{i=1}^L \zeta_i) (15(4L+1)) \| \mathbf{U}^{(L+1)} \|$$

$$\text{(Equation 17)} \tag{22}$$

Therefore, we can conclude the bound in Lemma 2.

## C.4 Proof of Theorem 2

We first normalize AllDeepSets with $\beta = (\prod_{i=j}^L \zeta_j \cdot \| \mathbf{W}^{L+1} \|)^{1/L_2}$, where $L_2 = 4L + 1$. Again, we have the norm equal across layers as $\beta$. We consider the prior distribution $P$ following the normal distribution $\mathcal{N}(0, \sigma^2 I)$. Similarly, random perturbation $\mathbf{U} \sim \mathcal{N}(0, \sigma^2 I)$, denoted as distribution $Q$. We want the value of $\sigma$ to be based on $\beta$. We choose to use some approximation $\tilde{\beta}$ of $\beta$ and guarantee that each value of $\beta$ can be covered by some $\tilde{\beta}$. Specifically, we let $|\beta - \tilde{\beta}| \leq \frac{1}{L_2} \beta$. According to Lemma 5, we wish to have

$$\Pr_{\mathbf{u}} \left[ \max_{A \in \mathcal{A}} \| \text{AllDeepSets}_{\mathbf{w+u}}(A) - \text{AllDeepSets}_{\mathbf{w}}(A) \|_2 \geq \frac{1}{2} \right].$$

Based on the work in [57], let $1 - 2L_2 h e^{\frac{-t^2}{2h\sigma^2}} = 1 - \frac{1}{2}$. Then we have $t = \sigma \sqrt{2h \ln (4hL_2)}$. Combining with Lemma 2, we will show that with probability at least $\frac{1}{2}$, for any input, we aim to have $\max_{A \in \mathcal{A}} \| \text{AllDeepSets}_{\mathbf{w+u}}(A) - \text{AllDeepSets}_{\mathbf{w}}(A) \|_2 < \frac{\gamma}{4}$. Let $\mathcal{I} = 2eBL(1 + \frac{1}{L_2})(15L_2) C_A^L$, where $C_A = (R+1)(M+1)$. Following Equation 19 we have

$$\max_{A \in \mathcal{A}} | \text{AllDeepSets}_{\mathbf{w+u}}(A) - \text{AllDeepSets}_{\mathbf{w}}(A) |_2$$

$$\leq \mathcal{I} (\prod_{j=1}^L \zeta_j) \| \mathbf{U}^{(L+1)} \| \leq e\mathcal{I} \tilde{\beta}^{L_2-1} \sigma \sqrt{2h \ln (4hL_2)} < \frac{\gamma}{4},$$

Then we let $\sigma = \frac{\gamma}{4e\mathcal{I} \tilde{\beta}^{L_2-1} \sqrt{2h \ln (4hL_2)}}$ to satisfy the condition in Lemma 5. We first consider the case of a fixed $\beta$. Given a $\beta$, we can calculate the KL divergence with the distribution $P$ and $Q$ and obtain the PAC-Bayes bound for $F$ as follows.

$$\text{KL}(\mathbf{w} + \mathbf{u} \| P) \leq \frac{|\mathbf{w}|^2}{2\sigma^2}$$

$$\leq O \left( \frac{B^2 C_A^{2L} L^4 h \ln (hL_2) \prod_{i=1}^{L_2} \| \mathbf{W}^{(i)} \|^2}{\gamma^2} \sum_{i=1}^{L_2} \frac{\| \mathbf{W}^{(i)} \|_F^2}{\| \mathbf{W}^{(i)} \|^2} \right).$$

Following the Lemma 5, for a fixed $\tilde{\beta}$ where $|\beta - \tilde{\beta}| \leq \frac{1}{l} \beta$, given training data $S$ with size $R$, then with probability at least $1 - \delta$, for $\delta, \gamma > 0$ and any $\mathbf{w}$, we have the following bound.

$$\mathcal{L}_{\mathcal{D},0}(F_{\mathbf{w}}) \leq \hat{\mathcal{L}}_{S,\gamma}(F_{\mathbf{w}}) + \tag{23}$$

$$O \left( \sqrt{\frac{L^4 B^2 h \ln (hL_2) C_A^{2L} \prod_{i=1}^{L_2} \| \mathbf{W}^{(i)} \|^2 \sum_{i=1}^{L_2} \frac{\| \mathbf{W}^{(i)} \|_F^2}{\| \mathbf{W}^{(i)} \|^2} + \log \frac{m}{\sigma}}{\gamma^2 m}} \right), \tag{24}$$

As we discussed above, we need to consider all possible choices of $\tilde{\beta}$ such that it can cover any value of $\beta$. Then we can obtain the generalization bound. We found that we only need to consider values of $\beta$ in the following range,

$$\left( \frac{\gamma}{2BC_A^L} \right)^{1/L_2} \leq \beta \leq \left( \frac{\sqrt{m}\gamma}{2BC_A^L} \right)^{1/L_2} \tag{25}$$

If $\beta < \left( \frac{\gamma}{2BC^L} \right)^{1/L_2}$, then for any input instance $A \in \mathcal{A}$, we have $\| F(A)[i] \|_2 \leq \frac{\gamma}{2}$ based on the Equation 17. We stated it in the following.

$$\text{AllDeepSets}_{\mathbf{w}}(A) = \mathbf{W}^{(L+1)} H^{(L)} \leq \| \mathbf{W}^{(L+1)} \| \max_i |H^{(L)}[i,:]|$$

$$= \| \mathbf{W}^{(L+1)} \| \Phi_L$$

$$= BC_A^L \| \mathbf{W}^{(L+1)} \| \prod_{j=1}^L \zeta_j \leq BC_A^L \beta^{L_2} \leq \frac{\gamma}{2}$$

Following the definition of margin loss in Equation 1, we will always have $\mathcal{L}_{S,\gamma}(f_{\mathbf{w}}) = 1$. In addition, regarding the assumption in Lemma 2 that $\frac{\| \mathbf{U}^{(i)} \|}{\| \mathbf{W}^{(i)} \|} < \frac{1}{L_2}$ for $i \in [L_2]$. This assumption will be satisfied when this lower bound holds, where

$$\| \mathbf{U}^{(i)} \| \leq \sigma \sqrt{2h \ln 4h} \leq \frac{\beta}{L_2} \implies \frac{\gamma}{30eBL_2 C_A^L} \leq \beta^{L_2}$$

Since we have the lower bound that $\frac{\gamma}{2BC_A^L} \leq \beta^{L_2}$, the above statement will always be satisfied. Regarding the upper bound, if $\beta^{L_2} > \frac{\sqrt{m}\gamma}{2BC_A^L}$, it is easily to get $\mathcal{L}_{S,\gamma}(f_{\mathbf{w}}) \geq 1$ when calculates the term inside the big-O notation in Equation 10.

$$\sqrt{\frac{L^4 B^2 h \ln h C_A^{2L} \prod_{i=1}^{L_2} \| \mathbf{W}^{(i)} \|^2 \sum_{i=1}^{L_2} \frac{\| \mathbf{W}^{(i)} \|_F^2}{\| \mathbf{W}^{(i)} \|^2} + \log \frac{mL}{\sigma}}{\gamma^2 m}}$$

$$\geq \sqrt{\frac{L^4 h \ln h \sum_{i=1}^{L_2} \frac{\| \mathbf{W}^{(i)} \|_F^2}{\| \mathbf{W}^{(i)} \|^2}}{4}} \geq 1,$$

where $\|\mathbf{W}^{(i)}\|_F^2 > \|\mathbf{W}^{(i)}\|^2$. Therefore, $\mathcal{L}_{\mathcal{D},0}(F_{\mathbf{w}})$ is always bounded by 1. As a result, we should only consider $\beta$ in the above range. Since $|\beta - \tilde{\beta}| \leq \frac{1}{l}\beta$, we have $|\beta - \tilde{\beta}| \leq \frac{1}{L_2}\left(\frac{\sqrt{m}\gamma}{2BC_A^L}\right)^{1/L_2}$. Thus, we use a cover of size $\frac{l}{2}\left(m^{1/2(L_2)} - 1\right)$ with radius $\frac{1}{L_2}\left(\frac{\gamma}{2BC_A^L}\right)^{1/L_2}$. Therefore, by taking a union bound over all possible $\tilde{\beta}$, we conclude the bound in Theorem 2.

## C.5 Proof of Lemma 3

Let $A_e \in \{0,1\}^{K\times K}$ be the adjacency matrix between hyperedges. Let $B$ be a diagonal matrix with size $K$.

$$\mathbf{A}_e[i,j] := \begin{cases} 1 & \text{if } e_i \in N(e_j) \\ 0 & \text{otherwise} \end{cases}. \qquad \mathbf{B}[i,j] := \begin{cases} 1 & \text{if } i = j \\ 0 & \text{otherwise} \end{cases}.$$

For the layer $l \in [1,L]$, we can rewrite $H^{(l)}$ as follows,

$$H^{(l)} = \text{ReLu}\left(\mathbf{W}^{(l)}\left((1+\alpha^{(l)})B + A_e\right)H^{(l-1)}\right) \qquad (26)$$

We can drive an upper bound on the $l_2$ norm of the maximum node representation. For $l \in [0,L]$, let $w_l^* = \arg\max_{i\in[K]} |H^{(l)}[i,:]|_2$ and $\Phi_t = |H^{(l)}[w_l^*,:]|_2$. Let $L^{(l)} = (1+\alpha^{(l)})B + A_e$, we have

$$\begin{aligned} \Phi_l &= \|\text{ReLu}(L^{(l)}H^{(l-1)}\mathbf{W}^{(l)})[w_l^*,:]\|_2 \\ &= \|\text{ReLu}(L^{(l)}H^{(l-1)}\mathbf{W}^{(l)}[w_l^*,:])\|_2 \\ &\leq \|L^{(l)}H^{(l-1)}\mathbf{W}^{(l)}[w_l^*,:]\|_2 \leq MD(1+\alpha^{(l)})\Phi_{l-1}\|\mathbf{W}^{(l)}\| \\ &\leq (MD)^{l-1}\Phi_1\prod_{i=1}^{l}(1+\alpha^{(i)})\|\mathbf{W}^{(i)}\| \\ &\leq E^{(1,l)}M^lD^{l-1}B\prod_{i=0}^{l}\|\mathbf{W}^{(i)}\|, \end{aligned} \qquad (27)$$

where $E^{(i,j)} = \prod_{k=i}^{j}(1+\alpha^k)$. For $l \in [L]$, Let

$$v_l^* = \arg\max_{i\in[K]}\|\hat{H}^{(l)}[i,:] - H^{(l)}[i,:]\|_2,$$

where $\hat{H}^{(l)}$ denotes the layer output with perturbed weights. Then let $\Psi_l = \|\hat{H}^{(l)}[v_l^*,:] - H^{(l)}[v_l^*,:]\|_2$. We then bound the max change of the node representation before the readout layers as

$$\begin{aligned} \Psi_l &= \left\|\text{ReLu}(L^{(l)}\hat{H}^{(l-1)}\mathbf{W}^{(l)})[v_l^*,:] - \right. \\ &\qquad \left. \text{ReLu}(L^{(l)}H^{(l-1)}\mathbf{W}^{(l)})[v_l^*,:]\right\|_2 \\ &= \left\|\text{ReLu}(L^{(l)}\hat{H}^{(l-1)}\mathbf{W}^{(l)}[v_l^*,:] - L^{(l)}H^{(l-1)}\mathbf{W}^{(l)}[v_l^*,:])\right\|_2 \\ &\leq \left\|(L^{(l)}\hat{H}^{(l-1)}[v_l^*,:] - L^{(l)}H^{(l-1)}[v_l^*,:])(\mathbf{W}^{(l)} + \mathbf{U}^{(l)})\right. \\ &\qquad \left. + L^{(l)}H^{(l-1)}\mathbf{U}^{(l)}[v_t^*,:]\right\|_2 \\ &\leq MD(1+\alpha^{(l)})\Psi_{l-1}\|\mathbf{W}^{(l)} + \mathbf{U}^{(l)}\| + MD(1+\alpha^{(l)})\Phi_{l-1}\|\mathbf{U}_2^{(l)}\| \end{aligned}$$

To simplify, we let $a_{l-1} = MD(1+\alpha^{(l)})\|\mathbf{W}^{(l)} + \mathbf{U}^{(l)}\|$ and $b_{l-1} = MD(1+\alpha^{(l)})\Phi_{l-1}\|\mathbf{U}^{(l)}\|$. We have $\Psi_l \leq a_{l-1}\Psi_{l-1} + b_{l-1}$. For $l \in [0,L]$, $\Psi_l \leq \prod_{i=0}^{l-1}a_i\Psi_0 + \sum_{i=0}^{l-1}b_i\left(\prod_{j=i+1}^{l-1}a_j\right)$. Since $\Psi_0 \leq$

$\|\mathbf{U}^{(0)}\bar{H}^{(0)}\|_2 \leq BM\|\mathbf{U}^{(0)}\|$ and input difference is 0, we have

$$\begin{aligned} \prod_{i=0}^{l-1}a_i\Psi_1 &\leq M^lD^{l-1}E^{(1,l)}B\|\mathbf{U}^{(0)}\|\prod_{i=1}^{l-1}\|\mathbf{W}^{(i)} + \mathbf{U}^{(i)}\| \\ &\leq M^lD^{l-1}E^{(1,l)}B\|\mathbf{U}^{(0)}\|\prod_{i=1}^{l-1}(1+\frac{1}{l})\|\mathbf{W}^{(i)}\| \end{aligned}$$

Since $1 + \frac{1}{L+2} > 1$ and $\|\mathbf{U}\| \leq (1+\frac{1}{L+2})\|\mathbf{W}\|$, we have

$$\prod_{i=0}^{l-1}a_i\Psi_1 \leq eM^lD^{l-1}E^{(1,l)}B\prod_{i=0}^{l-1}\|\mathbf{W}^{(i)}\| \qquad (28)$$

For the second term in $\Psi_l$, we have

$$\begin{aligned} \sum_{i=0}^{l-1}b_i\left(\prod_{j=i+1}^{l-1}a_j\right) &= \sum_{i=0}^{l-1}b_i\left(\prod_{j=i+1}^{l-1}a_j\right) \\ &= \sum_{i=0}^{l-1}MD(1+\alpha^{(i+1)})\Phi_i\|\mathbf{U}^{(i+1)}\|\left(\prod_{j=i+1}^{l-1}MD(1+\alpha^{(j+1)})\right. \\ &\qquad \left. \|\mathbf{W}^{(j+1)} + \mathbf{U}^{(j+1)}\|\right) \\ &\leq M^{l+1}D^lBE^{(1,l)}\sum_{i=0}^{l-1}\left(\prod_{j=0}^{i}\|\mathbf{W}^{(i)}\|\right)\|\mathbf{U}^{(i+1)}\| \\ &\qquad \left(\prod_{j=i+1}^{l-1}\|\mathbf{W}^{(j+1)} + \mathbf{U}^{(j+1)}\|\right) \\ &\leq M^{l+1}D^lBE^{(1,l)}\sum_{i=0}^{l-1}\left(\prod_{j=0}^{i+1}\|\mathbf{W}^{(j)}\|\right)\frac{\|\mathbf{U}^{(i+1)}\|}{\|\mathbf{W}^{(i+1)}\|} \\ &\qquad \prod_{j=i+2}^{l}(1+\frac{1}{L+2})\|\mathbf{W}^{(j)}\| \\ &= M^{l+1}D^lBE^{(1,l)}\prod_{i=0}^{l}\|\mathbf{W}^{(i)}\|\sum_{i=1}^{l}\frac{\|\mathbf{U}^{(i)}\|}{\|\mathbf{W}^{(i)}\|}(1+\frac{1}{L+2})^{l-i} \qquad (29) \end{aligned}$$

Combining, we have

$$\Psi_l \leq eM^{l+1}D^lBE^{(1,l)}\prod_{i=0}^{l}\|\mathbf{W}^{(i)}\|\sum_{i=1}^{l}\frac{\|\mathbf{U}^{(i)}\|}{\|\mathbf{W}^{(i)}\|}(1+\frac{1}{L+2})^{l-i}$$

Let $C = eM^{L+1}D^LBE^{(1,L)}$. we can bound the change of M-IGN's output with and without the weight perturbation as follows.

$$\begin{aligned} \Psi_{L+1} &= \left|\frac{1}{K}\mathbf{1}_KA_e\hat{H}^{(L)}(\mathbf{W}^{l+1} + \mathbf{U}^{(L+1)}) - \frac{1}{K}\mathbf{1}_KA_eH^{(L+1)}\mathbf{W}^{(L+1)}\right|_2 \\ &\leq \frac{1}{K}\|\mathbf{1}_KA_e(\hat{H}^{(L+1)} - H^{(L+1)})(\mathbf{W}^{(L+1)} + \mathbf{U}^{(L+1)})\|_2 \\ &\qquad + \frac{1}{K}\|\mathbf{1}_KA_eH^{(L+1)}\mathbf{U}^{(L+1)}\|_2 \\ &\leq MD\Psi_L\|\mathbf{W}^{(L+1)} + \mathbf{U}^{(L+1)}\| + MD\Phi_L\|\mathbf{U}^{(L+1)}\| \\ &\leq eMDC\prod_{i=0}^{L+1}\|\mathbf{W}^{(i)}\|\sum_{i=0}^{L}\frac{\|\mathbf{U}^{(i)}\|}{\|\mathbf{W}^{(i)}\|}. \end{aligned}$$

## C.6 Proof of Theorem 3

Similar to the proof process of the Theorem 1, this proof involves two parts. First, we aim to establish the maximum permissible perturbation that fulfills the specified margin condition $\gamma$. Second, based on Lemma 5, we then use the perturbation to calculate the KL-term and obtain the bound. We let $L_3 = L + 2$. We consider $\beta = (\prod_{i=0}^{L+1} \|\mathbf{W}^{(i)}\|)^{1/L_3}$. We normalize the weights as $\frac{\beta}{\|\mathbf{W}^{(i)}\|}\mathbf{W}^{(i)}$ for $i \in [L_3]$. Then we assume the same spectral norm across layers, i.e., $\|\mathbf{W}^{(i)}\| = \beta$ for $i \in [L_3]$. We again consider the prior distribution $P$ following the normal distribution $\mathcal{N}(0, \sigma^2 I)$. Similarly, random perturbation $\mathbf{U} \sim \mathcal{N}(0, \sigma^2 I)$. According to Lemma 5, we aim to have

$$\Pr_{\mathbf{u}}\left[\max_{A \in \mathcal{A}} \|\text{M-IGN}_{\mathbf{w}+\mathbf{u}}(A) - \text{M-IGN}_{\mathbf{w}}(A)\|_2 < \frac{\gamma}{4}\right] \geq \frac{1}{2}.$$

Based on the work in [57], we let $1 - 2lhe^{\frac{-t^2}{2h\delta^2}} = \frac{1}{2}$, we have $t = \sigma\sqrt{2h \ln 4hl}$. Combining with Lemma 2, we will show that with probability at least $\frac{1}{2}$, for any input, we aim to have

$$\max_{A \in \mathcal{A}} \|\text{M-IGN}_{\mathbf{w}+\mathbf{u}}(A) - \text{M-IGN}_{\mathbf{w}}(A)\|_2 < \frac{\gamma}{4}.$$

Let $\mathcal{I} = e^2 M^{L+2} D^{L+1} B E^{(1,L)}$, then we have

$$\max_{A \in \mathcal{A}} \|\text{M-IGN}_{\mathbf{w}+\mathbf{u}}(A) - \text{M-IGN}_{\mathbf{w}}(A)\|_2 \leq \frac{\gamma}{4}$$

$$\leq \mathcal{I} \prod_{i=0}^{L_3} \|\mathbf{W}^{(i)}\| \left(\sum_{i=0}^{L_3} \frac{\|\mathbf{U}^{(i)}\|}{\|\mathbf{W}^{(i)}\|}\right)$$

$$\leq \mathcal{I} \beta^{L_3} \left(\sum_{i=0}^{L_3} \frac{\|\mathbf{U}^{(i)}\|}{\beta}\right) = \mathcal{I} \beta^{L_3-1} \left(\sum_{i=0}^{L_3} \|\mathbf{U}^{(i)}\|\right)$$

$$\leq \mathcal{I} \beta^{L_3-1} L_3 \sigma \sqrt{2h \ln 4hL_3}$$

$$\leq e\mathcal{I} \tilde{\beta}^{L_3-1} L_3 \sigma \sqrt{2h \ln 4hL_3} < \frac{\gamma}{4},$$

Then we let $\sigma = \frac{\gamma}{4e\mathcal{I}\tilde{\beta}^{L_3-1}L_3\sqrt{2h \ln 4hL_3}}$ to satisfy the condition in Lemma 5. We first consider the case of a fixed $\beta$. Given a $\beta$, we can calculate the KL divergence with the distribution $P$ and $\mathbf{w} + \mathbf{u}$ and obtain the PAC-Bayes bound for $F$ as follows.

$$\text{KL}(\mathbf{w} + \mathbf{u}\|P) \leq \frac{|\mathbf{w}|^2}{2\sigma^2}$$

$$\leq O\left(\frac{C \prod_{i=0}^{L_3} \|\mathbf{W}^{(i)}\|^2}{\gamma^2} \sum_{i=0}^{L_3} \frac{\|\mathbf{W}^{(i)}\|_F^2}{\|\mathbf{W}^{(i)}\|^2}\right),$$

where $C = e^4(BL)^2(MD)^L(E^{(2,L)})^2 h \ln(hL)$. Following the Lemma 5, for a fixed $\tilde{\beta}$ where $|\beta - \tilde{\beta}| \leq \frac{1}{l}\beta$, given training data $S$ with size $R$, then with probability at least $1 - \delta$, for $\delta, \gamma > 0$ and any $\mathbf{w}$, we have the following bound.

$$\mathcal{L}_{\mathcal{D},0}(F_{\mathbf{w}}) \leq \hat{\mathcal{L}}_{S,\gamma}(F_{\mathbf{w}})$$

$$+ O\left(\sqrt{\frac{C \prod_{i=0}^{L_3} \|\mathbf{w}^{(i)}\|^2 \sum_{i=0}^{L_3} \frac{\|\mathbf{w}^{(i)}\|_F^2}{\|\mathbf{w}^{(i)}\|^2} + \log \frac{m}{\sigma}}{\gamma^2 m}}\right), \quad (30)$$

where $C = e^4(BL)^2(MD)^L(E^{(2,L)})^2 h \ln(hL)$. As we discussed above, we need to consider all possible choices of $\tilde{\beta}$ such that it can cover

any value of $\beta$. Then we can obtain the PAC-Bayes bound. We found that we only need to consider values of $\beta$ in the following range,

$$\left(\frac{\gamma}{2E^{(1,L)}M^{L+1}D^L B}\right)^{1/L_3} \leq \beta \leq \left(\frac{\sqrt{m}\gamma}{2E^{(1,L)}M^{L+1}D^L B}\right)^{1/L_3} \quad (31)$$

If $\beta < \left(\frac{\gamma}{2E^{(1,L)}M^{L+1}D^L B}\right)^{1/L_3}$, then for any input instance $A \in \mathcal{A}$, we have $|F(A)[i]|_2 \leq \frac{\gamma}{2}$ based on the Equation 27. We stated it in the following.

$$H^{(L_3)}(A) = A_e H^{(L_3-1)}\mathbf{W}^{(L_3)} \leq \|\mathbf{W}^{(L_3)}\|A_e \max_i \|H^{(L_3-1)}[i,:]\|_2$$

$$\leq E^{(1,L)}M^{L+1}D^L B \prod_{i=0}^{l} \|\mathbf{W}^{(i)}\| = E^{(1,L)}M^{L+1}D^L B\beta^{L_3} \leq \frac{\gamma}{2}$$

Following the definition of margin loss in Equation 1, we will always have $\mathcal{L}_{S,\gamma}(f_{\mathbf{w}}) = 1$. In addition, regarding the assumption in Lemma 2 that $\frac{\|\mathbf{U}^{(i)}\|}{\|\mathbf{W}^{(i)}\|} < \frac{1}{L_3}$ for $i \in [l]$. This assumption will be satisfied when this lower bound holds, where

$$\|\mathbf{U}^{(i)}\| \leq \sigma\sqrt{2h \ln 4hL_3} \leq \frac{\beta}{L_3}$$

Since we have the lower bound that $\frac{\gamma}{2E^{(1,L)}M^{L+1}D^L B} \leq \beta^{L_3}$, the above statement will always be satisfied. Regarding the upper bound, if $\beta^{L_3} > \frac{\sqrt{R}\gamma}{2E^{(1,L)}M^{L+1}D^L B}$, it is easily to get $\mathcal{L}_{S,\gamma}(f_{\mathbf{w}}) \geq 1$ when calculates the term inside the big-O notation in Equation 30. Therefore, $\mathcal{L}_{\mathcal{D},0}(f_{\mathbf{w}})$ is always bounded by 1. As a result, we should only consider $\beta$ in the above range. To hold our assumption that $|\beta - \tilde{\beta}| \leq \frac{1}{L_3}\beta$, we need to have $|\beta - \tilde{\beta}| \leq \frac{1}{L_3}\left(\frac{\sqrt{m}\gamma}{2B(MD)^L E^{(2,L)}}\right)^{1/L_3}$. We use a cover of size $\frac{l}{2}(m^{1/2L_3} - 1)$ with radius $\frac{1}{L_3}\left(\frac{\gamma}{2B(MD)^L E^{(2,L)}}\right)^{1/L_3}$. That is, given a fixed $\tilde{B}$, we have an event in Equation 10 that happened with probability $1 - \delta$. We need to cover all possible $\beta$ that is number of $\frac{l}{2}(m^{1/2L_3} - 1)$ such event happened. Therefore, by taking a union bound, we conclude the bound in Theorem 3.

## C.7 Proof of Lemma 4

Given the input $A$, for $l \in [L]$, let $H_{w+u}^{(l)}$ and $H_w^{(l)}(A)$ be the $l$-layer output with parameter $w$ and perturbed parameter $w + u$, respectively. Use $\Delta_l$ to denote the change in the output. We define $\Psi_l = \max_{i \in (N)} \|\Delta_l[i,:]\|_2 = \max_i \|\hat{H}^{(l)}[i,:] - H^{(l)}[i,:]\|_2$, where $\hat{H}^{(l)} := H_{\mathbf{w}+\mathbf{u}}^{(l)}(A)$ and $H^{(l)} := H_{\mathbf{w}}^{(l)}(A)$.

Let $v_{(l)}^* = \arg\max_i \|\Delta_{(l)}[i,:]\|_2$. Since after each propagation step, T-MPHN includes a row-wise normalization. We can first bound the $\Psi_l$ by 2 for any $l \in [L]$. In particular, let $A[i,:]$ and $B[i,:]$ be two non-zero row vectors. We define the normalized vectors $\hat{A}[i,:] = \frac{A[i,:]}{\|A[i,:]\|_2}$ and $\hat{B}[i,:] = \frac{B[i,:]}{\|B[i,:]\|_2}$. The expression of interest is $\|\hat{A}[i,:] - \hat{B}[i,:]\|_2$, which represents the Euclidean distance between these two unit vectors. The upper bound for this expression is 2, which is achieved when the vectors are completely opposite in

direction. Therefore, considering the last readout layer, we have

$$|\Delta_{L+1}|_2 = \|\frac{1}{N}\mathbf{1}_N\hat{H}^{(L)}(\mathbf{W}^{(L+1)} + \mathbf{U}^{(L+1)}) - \frac{1}{N}\mathbf{1}_N H^{(L)}(\mathbf{W}^{(L+1)})\|_2$$

$$\leq \frac{1}{N}\left\|\mathbf{1}_N(\hat{H}^{(L)} - H^{(L)})(\mathbf{W}^{(L+1)} + \mathbf{U}^{(L+1)})\right.$$

$$\left. + \frac{1}{N}\mathbf{1}_N H^{(L)}\mathbf{U}^{(L+1)}\right\|_2$$

$$\leq \frac{1}{N}\|\mathbf{W}^{(L+1)} + \mathbf{U}^{(L+1)}\|(\sum_{i=1}^{N}\|\Delta_L[i,:]\|_2)$$

$$+ \frac{1}{N}\|\mathbf{U}^{(L+1)}\|(\sum_{i=1}^{N}\|H^{(L)}[i,:]\|_2)$$

$$\leq \Psi_L\|\mathbf{W}^{(L+1)} + \mathbf{U}^{(L+1)}\| + \|\mathbf{U}^{(L+1)}\|$$

$$\leq 2\|\mathbf{W}^{(L+1)}\| + 3\|\mathbf{U}^{(L+1)}\|$$

## C.8 Proof of Theorem 4

The proof involves two parts. First, we aim to establish the maximum permissible parameter perturbation that fulfills the specified margin condition $\gamma$. Second, based on Lemma 5, we use the perturbation to calculate the KL-term and obtain the bound. To simplify, we let $L_4 = L+1$. We consider $\beta = \left(\prod_{i=1}^{L+1}\|\mathbf{W}^{(i)}\|\right)^{\frac{1}{L_4}}$. We normalize the weights as $\frac{\beta}{\|\mathbf{W}^i\|}\mathbf{W}^i$, where $i \in [L_4]$. Therefore, we assume the norm is equal across layers, i.e., $\|\mathbf{W}^i\| = \beta$.

Consider the prior distribution $P = \mathcal{N}(0, \sigma_{\mathbf{n}}^2 I)$ and the random perturbation $\mathbf{U} \sim \mathcal{N}(0, \sigma_{\mathbf{n}}^2 I)$, denoted as distribution $Q$. Notice that the prior and the perturbation are the same with the same $\sigma$. We want the value of $\sigma$ to be based on $\beta$. However, we cannot use the learned parameter $w$. We then choose to use some approximation $\tilde{\beta}$ of $\beta$ and guarantee that each value of $\beta$ can be covered by some $\tilde{\beta}$. Let $|\beta - \tilde{\beta}| \leq \frac{1}{L_4}\beta$. According to Lemma 5, for any input $A$, we have $\max_{A\in\mathcal{A}}|\max_{A\in\mathcal{A}}\|\text{T-MPHN}_{\mathbf{w}+\mathbf{u}}(A) - \text{T-MPHN}_{\mathbf{w}}(A)\|_2 < \frac{\gamma}{4}$. Then we have

$$\max_{A\in\mathcal{A}}|\text{T-MPHN}_{\mathbf{w}+\mathbf{u}}(A) - \text{T-MPHN}_{\mathbf{w}}(A)|$$

$$\leq 2\|\mathbf{W}^{(L+1)}\| + 3\|\mathbf{U}^{(L+1)}\| \leq \frac{2}{1 - \frac{1}{L_4}}\tilde{\beta} + 3\sigma\sqrt{2h\ln 4h} < \frac{\gamma}{4},$$

Then we let $\sigma = \frac{\gamma L - 8L_4\tilde{\beta}}{12L\sqrt{2h\ln 4h}}$ to satisfy the condition in Lemma 5. Remember that we have one conditions in Lemma 4 which is $\|\mathbf{U}^{(i)}\| \leq \gamma$ for $i \in [L_4]$. Given the value of $\sigma$, the assumption indicates that $\beta \geq 0$, which is always satisfied. We first consider the case of a fixed $\beta$. Given a $\beta$, we can calculate the KL divergence with the distribution $P$ and $\mathbf{w} + \mathbf{u}$ and obtain the PAC-Bayes bound for $F$ as follows.

$$KL(\mathbf{w} + \mathbf{u}\|P) \leq \frac{|\mathbf{w}|^2}{2\sigma^2} = \frac{144L^2h\ln(4h)}{(\gamma L + 8L_4\tilde{\beta})^2}\sum_{i=1}^{L_4}\|\mathbf{W}^{(i)}\|_F^2$$

$$\leq O\left(\frac{h\ln h}{(\gamma - \|\mathbf{W}^{(L_4)}\|)^2}\sum_{i=1}^{L_4}\|\mathbf{W}^{(i)}\|_F^2\right).$$

Following the Lemma 5, for a fixed $\tilde{\beta}$ where $|\beta - \tilde{\beta}| \leq \frac{1}{l}\beta$, given training data $S$ with size $R$, then with probability at least $1 - \delta$, for

$\delta, \gamma > 0$ and any $\mathbf{w}$, we have the following bound.

$$\mathcal{L}_{\mathcal{D},0}(f_{\mathbf{w}}) \leq \hat{\mathcal{L}}_{S,\gamma}(f_{\mathbf{w}}) + O\left(\sqrt{\frac{h\ln h\sum_{i=1}^{L_4}\|\mathbf{W}^{(i)}\|_F^2 + \log\frac{m}{\sigma}}{m(\gamma - \|\mathbf{W}^{(L_4)}\|)^2}}\right), \quad (32)$$

As we discussed above, we need to consider all possible choices of $\tilde{\beta}$ such that it can cover any value of $\beta$. Then we can obtain the PAC-Bayes bound. We found that we only need to consider values of $\beta$ in the following range.

$$\frac{\gamma}{2} \leq \beta \leq \frac{\gamma\sqrt{m}}{2}. \quad (33)$$

If $\beta < \frac{\gamma}{2}$, then for any input instance $A$, we have $|F(A)[i]|_2 \leq \frac{\gamma}{2}$ based on the Lemma 4. We stated it in the following.

$$\|\text{T-MPHN}(A)\|_2 = \|\mathbf{W}^{(L_4)}\| = \beta \leq \frac{\gamma}{2}$$

Following the definition of margin loss in Equation 1, we will always have $\hat{\mathcal{L}}_{S,\gamma}(f_{\mathbf{w}}) = 1$. Meanwhile, if $\beta \geq \frac{\gamma\sqrt{m}}{2}$, we can easily get $\mathcal{L}_{S,\gamma}(f_{\mathbf{w}}) \geq 1$ when calculating the term inside the big-O notation in Equation 10.

$$\sqrt{\frac{h\ln h\sum_{i=1}^{L_4}\|\mathbf{W}^{(i)}\|_F^2 + \log\frac{m}{\sigma}}{m(\gamma - \|\mathbf{W}^{(L_4)}\|)^2}} \geq \sqrt{\frac{h\ln h\sum_{i=1}^{L_4}\|\mathbf{W}^{(i)}\|^2 + \log\frac{m}{\sigma}}{m(\gamma)^2}}$$

$$\geq 1,$$

where $\|\mathbf{W}^{(i)}\|_F^2 > \|\mathbf{W}^{(i)}\|_2^2$. Therefore, $\mathcal{L}_{\mathcal{D},0}(f_{\mathbf{w}})$ is always bounded by 1. Since we have the upper bound, the above statement will always be satisfied. As a result, we should only consider $\beta$ in the above range, see Equation 33. We have an assumption that $|\beta - \tilde{\beta}| \leq \frac{1}{L_4}\beta$. Thus, we use a cover of size $(\sqrt{m} - 1)L_4$ with radius $\frac{\gamma}{2L_4}$. That is, given a fixed $\tilde{\beta}$, we have an event in Equation 32 that happened with probability $1 - \delta$. We need to cover all possible $\beta$ that is a number of $(\sqrt{m} - 1)L_4$ such event happened. Therefore, we conclude the bound in Theorem 4 by taking a union bound.

## C.9 Main results on HGNN+

HGNN+ performs spectral hypergraph convolution by using the hypergraph Laplacian, which models the relationships between vertices and hyperedges [24]. The model adopts various strategies to generate Hyperedge groups that capture different types of relationships or correlations in the data, (i.e., $k$-Hop neighbor group or feature-based group). Specifically, given $z \in \mathbb{Z}^+$ hyperedge groups $\{\mathcal{E}_1, \ldots, \mathcal{E}_z\}$ where $\mathcal{E}_i \subseteq \mathcal{E}$ for $i \in [z]$, the incident matrix $\mathbf{J} \in \{0, 1\}^{N \times p}$ is given by $\mathbf{J} = \mathbf{J}_1|\ldots|\mathbf{J}_z$, where $|$ is matrix concatenation operation, $\mathbf{J}_i \in \{0, 1\}^{N \times p_i}$ and $p = \sum_{i\in[z]}p_i$. For each $\mathbf{J}_i$, its entries $\mathbf{J}_i(v, e)$ equal to 1 if $v \in e$ for $v \in \mathcal{V}$ and $e \in \mathcal{E}_i$; Otherwise 0. The model takes $\mathcal{G}$, the node features $\mathbf{X}$ and the diagonal matrix of hyperedge weights $\mathbf{T} \in \mathbb{R}^{K \times K}$ as input. The initial representation is $H^{(0)} \in \mathbb{R}^{N \times d_0}$. Suppose that there are $L \in \mathbb{Z}^+$ propagation steps. In each step $l \in [L]$, the hidden representation $H^{(l)} \in \mathbb{R}^{d_{l-1} \times d_l}$ is calculated by

$$H^{(l)} = \text{ReLu}(\mathbf{D}_v^{-1}\mathbf{J}\mathbf{T}\mathbf{D}_e^{-1}\mathbf{J}^\top H^{(l)}\mathbf{W}^{(l)}),$$

where $\mathbf{W}^{(l)} \in \mathbb{R}^{d_{l-1} \times d_l}$. $\mathbf{D}_v \in \mathbb{R}^{N \times N}$ and $\mathbf{D}_e \in \mathbb{R}^{K \times K}$ are diagonal matrices of the vertex and hyperedge degree, respectively. The readout layer is defined as $\text{HGNN+}(A) = \frac{1}{N}\mathbf{1}_N H^{(L)}\mathbf{W}^{(L+1)}$, where

$\mathbf{W}^{(L+1)} \in \mathbb{R}^{d_L \times C}$ and $\mathbf{1}_N$ is an all-one vector. Let the maximum hidden dimension be $h \coloneqq \max_{l \in [L]} d_l$. We have the following results for HGNN+.

LEMMA 9. *Consider the* HGNN+ *with* $L + 1$ *layers and parameters* $\mathbf{w} = (\mathbf{W}^{(1)}, \dots, \mathbf{W}^{(L+1)})$. *For each* $\mathbf{w}$, *each perturbation* $\mathbf{u} = (\mathbf{U}^{(1)}, \dots, \mathbf{U}^{(L+1)})$ *on* $\mathbf{w}$ *such that* $\max_{i \in [L+1]} \frac{||U^{(i)}||}{||W^{(i)}||} \leq \frac{1}{L+1}$, *and each input* $A \in \mathcal{S}$, *we have*

$$\|\text{HGNN+}_{\mathbf{w+u}}(A) - \text{HGNN+}_{\mathbf{w}}(A)\|_2 \leq$$

$$\leq eB(CD^{\frac{1}{2}}RM)^L (\prod_{i=1}^{L+1} \|W^{(i)}\|)(\sum_{i=1}^{L+1} \frac{||\mathbf{U}^{(i)}||}{||\mathbf{W}^{(i)}||}),$$

*where* $C = \max_i \mathbf{T}_{ii}$.

PROOF. The proof includes two parts. We first analyze the maximum node representation among each layer except the readout layer. After adding the perturbation $\mathbf{u}$ to the weight $\mathbf{w}$, for each layer $l \in [L+1]$, we denote the perturbed weights $\mathbf{W}^{(l)} + \mathbf{U}^{(l)}$. We define $\boldsymbol{\theta} \in \mathbb{R}^{K \times d_{l-1} \times d_l}$ as the perturbation tensors. In particular $\boldsymbol{\theta}^{(l)}[k,:] = \boldsymbol{\theta}^{(l)}[j,:] = \mathbf{U}^{(l)} \in \mathbb{R}^{d_{l-1} \times d_l}$ when $j \neq k$. We then can derive an upper bound on the $l_2$ norm of the maximum node representation in each layer. Let $w_l^* = \arg\max_{i \in [N]} \|H^{(l)}[i,:]\|_2$ and $\Phi_l = \|H^{(l)}[w_l^*,:]\|_2$.

$$\Phi_l = \|\text{ReLu}(\mathbf{D}_v^{-1}\mathbf{J}\mathbf{T}\mathbf{D}_e^{-1}\mathbf{J}^\top H^{(l)}\mathbf{W}^{(l)})[w_l^*,:]\|_2$$

$$\leq \|\mathbf{D}_v^{-1}\mathbf{J}\mathbf{T}\mathbf{D}_e^{-1}\mathbf{J}^\top H^{(l)}\mathbf{W}^{(l)}[w_l^*,:]\|_2$$

$$= \|\mathbf{D}_v^{-1}\mathbf{J}\mathbf{T}\mathbf{D}_e^{-1}\mathbf{J}^\top H^{(l)}[w_l^*,:]\|_2 \|\mathbf{W}^{(l)}\|$$

$$= \|\sum_{i=1}^{N} \mathbf{D}_v^{-1}\mathbf{J}\mathbf{T}\mathbf{D}_e^{-1}\mathbf{J}^\top[w_l^*,i]H^{(l-1)}[i,:]\|_2 \|\mathbf{W}^{(l)}\|$$

$$\leq \sum_{i=1}^{N} \mathbf{D}_v^{-1}\mathbf{J}\mathbf{T}\mathbf{D}_e^{-1}\mathbf{J}^\top[w_l^*,i]\|H^{(l-1)}[i,:]\|_2 \|\mathbf{W}^{(l)}\|$$

$$\leq \sum_{i=1}^{N} \mathbf{D}_v^{-1}\mathbf{J}\mathbf{T}\mathbf{D}_e^{-1}\mathbf{J}^\top[w_l^*,i]\Phi_{l-1} \|\mathbf{W}^{(l)}\|$$

Since we have

$$\|\mathbf{D}_v^{-1}\mathbf{J}\mathbf{T}\mathbf{D}_e^{-1}\mathbf{J}^\top\|_\infty = \max_{i \in N} \sum_{j=1}^{N} |\mathbf{D}_v^{-1}\mathbf{J}\mathbf{T}\mathbf{D}_e^{-1}\mathbf{J}^\top[i,j]|$$

We denote $\mathbf{A} = \mathbf{D}_v^{-1}\mathbf{J}\mathbf{T}\mathbf{D}_e^{-1}\mathbf{J}^\top$ and for each entry, we have

$$\mathbf{A}_{ij} = [\mathbf{D}_v^{-1}\mathbf{J}\mathbf{T}\mathbf{D}_e^{-1}\mathbf{J}^\top]_{ij}$$

$$= \frac{1}{d_v(i)}[\mathbf{J}\mathbf{T}\mathbf{D}_e^{-1}\mathbf{J}^\top]_{ij}$$

$$= \frac{1}{d_v(i)}\sum_{e=1}^{p} \mathbf{J}_{ie}\left(\frac{t(e)}{d_e(e)}\right)\mathbf{J}_{je},$$

where a) $\mathbf{J}_{ie} = 1$ if vertex $i$ is incident to hyperedge $e$, and 0 otherwise; b) $\mathbf{J}_{je} = 1$ if vertex $j$ is incident to hyperedge $e$, and 0 otherwise. The product $\mathbf{J}_{ie}\mathbf{J}_{je}$ equals 1 if both vertices $i$ and $j$ are incident to hyperedge $e$, and 0 otherwise. Thus,

$$\mathbf{A}_{ij} = \frac{1}{d_v(i)}\sum_{e \in \mathcal{E}_{ij}} \frac{t(e)}{d_e(e)},$$

where $\mathcal{E}_{ij} = \{e \in \mathcal{E} \mid i \in e \text{ and } j \in e\}$. We calculate the sum over all columns $j$ for a fixed row $i$.

$$\sum_{j=1}^{N} \mathbf{A}_{ij} = \frac{1}{d_v(i)}\sum_{j=1}^{N}\sum_{e \in \mathcal{E}_{ij}} \frac{t(e)}{d_e(e)}$$

$$= \frac{1}{d_v(i)}\sum_{e \ni i} \frac{t(e)}{d_e(e)} \cdot d_e(e)$$

$$= \frac{1}{d_v(i)}\sum_{e \ni i} t(e).$$

The infinity norm of the matrix $\mathbf{A}$ is the maximum absolute row sum.

$$\|\mathbf{A}\|_\infty = \max_{i \in N} \sum_{j=1}^{N} |\mathbf{A}_{ij}|$$

$$= \max_{i \in N} \sum_{j=1}^{N} \mathbf{A}_{ij} \quad (\text{since } \mathbf{A}_{ij} \geq 0)$$

$$= \max_{i \in N} \left(\frac{1}{d_v(i)}\sum_{e \ni i} t(e)\right) \leq D^{\frac{1}{2}} \max_i \mathbf{T}_{ii}.$$

Let $C = D^{\frac{1}{2}}RM \max_i \mathbf{T}_{ii}$. Therefore, we have

$$\Phi_l \leq \sum_{i=1}^{N} \mathbf{D}_v^{-1}\mathbf{J}\mathbf{T}\mathbf{D}_e^{-1}\mathbf{J}^\top[w_l^*,i]\Phi_{l-1}\|\mathbf{W}^{(l)}\|$$

$$\leq (D^{\frac{1}{2}}RM \max_i \mathbf{T}_{ii})^l B \prod_{i=1}^{l} \|\mathbf{W}^{(i)}\|$$

Finally, we need to consider the readout layer. We have

$$|\Delta_{L+1}|_2 = \left\|\frac{1}{N}\mathbf{1}_N\hat{H}^{(L)}(\mathbf{W}^{(L+1)} + \mathbf{U}^{(L+1)})\right.$$

$$\left. - \frac{1}{N}\mathbf{1}_N H^{(L+1)}(\mathbf{W}^{(L+1)})\right\|_2$$

$$\leq \frac{1}{N}\left\|\mathbf{1}_N(H^{(L+1)'} - H^{(L+1)})(\mathbf{W}^{(L+1)}\right.$$

$$\left. + \mathbf{U}^{(L+1)})\right\|_2 + \frac{1}{N}\|\mathbf{1}_N H^{(L+1)}\mathbf{U}^{L+1}\|_2$$

$$\leq \frac{1}{N}\|\mathbf{1}_N\Delta_L(\mathbf{W}^{(L+1)} + \mathbf{U}^{(L+1)})\|_2 + \frac{1}{N}\|\mathbf{1}_N H^{(L)}\mathbf{U}^{(L+1)}\|_2$$

$$\leq \frac{1}{N}\|\mathbf{W}^{(L+1)} + \mathbf{U}^{(L+1)}\|\left(\sum_{i=1}^{N} |\Delta_{L+1}[i,:]|_2\right)$$

$$+ \frac{1}{N}\|\mathbf{U}^{(L+1)}\|\left(\sum_{i=1}^{N} |H^{(L)}[i,:]|_2\right)$$

$$\leq \Psi_L\|\mathbf{W}^{(L+1)} + \mathbf{U}^{(L+1)}\| + \Phi_L\|\mathbf{U}^{(L+1)}\|$$

$$\leq BC^L\left(\prod_{i=1}^{L+1} \|\mathbf{W}^{(i)}\|\right)\left(1 + \frac{1}{L+1}\right)^{L+1}\left[\sum_{i=1}^{L} \frac{||\mathbf{U}^{(i)}||}{||\mathbf{W}^{(i)}||}\right.$$

$$\left(1 + \frac{1}{L+1}\right)^{-i} + \frac{||\mathbf{U}^{(L+1)}||}{||\mathbf{W}^{(L+1)}||}\left(1 + \frac{1}{L+1}\right)^{-(L+1)}\right]$$

$$\leq eBC^L \left(\prod_{i=1}^{L+1} \|\mathbf{W}^{(i)}\|\right)\left[\sum_{i=1}^{L+1} \frac{\|\mathbf{U}^{(i)}\|}{\|\mathbf{W}^{(i)}\|}\right]$$

Therefore, we conclude the bound in Lemma 9. □

**Theorem 5.** *For* HGNN+ *parameted by* $\mathbf{w}$ *with* $L+1$ *layers and each* $\delta, \gamma > 0$, *with probability at least* $1-\delta$ *over a training set* $S$ *of size* $m$, *for any fixed* $\mathbf{w}$, *we have*

$$\mathcal{L}_\mathcal{D}(\text{HGNN+}_\mathbf{w}) \leq \mathcal{L}_{S,\gamma}(\text{HGNN+}_\mathbf{w})+$$

$$O\left(\sqrt{\frac{L^2 B^2 h \ln(Lh)(CD^{\frac{1}{2}}RM)^L \mathcal{W}_1 \mathcal{W}_2 + \log\frac{mL}{\sigma}}{\gamma^2 m}}\right),$$

*where* $\mathcal{W}_1 = \prod_{i=1}^{L+1} \|\mathbf{W}^{(i)}\|^2$, $\mathcal{W}_2 = \sum_{i=1}^{L+1} \frac{\|\mathbf{W}^{(i)}\|_F^2}{\|\mathbf{W}^{(i)}\|^2}$.

**Proof.** We consider a transformation of HGNN+ with the normalized weights $\tilde{\mathbf{W}}^{(i)} = \frac{\beta}{\|\mathbf{W}^{(i)}\|}\mathbf{W}^{(i)}$, where

$$\beta = \left(\prod_{i=1}^{L+1} \|\mathbf{W}^{(i)}\|\right)^{1/(L+1)}.$$

Therefore we have the norm equal across layers, i.e., $\|\mathbf{W}^{(i)}\| = \beta$. The proof can be separated into three parts for three ranges of $\beta$.

- **First case.** $\beta$ that satisfy $\mathcal{L}_{S,\gamma}(f_\mathbf{w}) = 1$.
- **Second case.** $\beta$ that satisfy $\sqrt{\frac{\text{KL}(\mathbf{w}^*+\mathbf{u}\|\mathbf{w})+\ln\frac{6m}{\delta}}{m-1}} > 1$.
- **Third case.** $\beta$ in the following range, denoted as $\mathcal{B}$.

$$\left\{\beta \Big| \beta \in \left[\left(\frac{\gamma}{2B(D^{\frac{1}{2}}RMt_{\max})^L}\right)^{1/L+1}, \left(\frac{\gamma\sqrt{m}}{2B(D^{\frac{1}{2}}RMt_{\max})^L}\right)^{1/L+1}\right]\right\}.$$

Similar to the idea in the proof of the Theorem 1, combining three cases, we conclude the theorem 5. □

## C.10 Main results on HGNN

HGNN performs spectral hypergraph convolution by using the hypergraph Laplacian. Given the hypergraph $\mathcal{G}$, the incident matrix $\mathbf{J} \in \{0,1\}^{N \times K}$ is defined as $\mathbf{J}(v,e)$ equal to 1 if $v \in e$ for $v \in \mathcal{V}$ and $e \in \mathcal{E}$; Otherwise 0. The model takes $\mathcal{G}$, the node features $\mathbf{X}$ and the diagonal matrix of hyperedge weights $\mathbf{T} \in \mathbb{R}^{K \times K}$ as input. The initial representation is $H^{(0)} \in \mathbb{R}^{N \times d_0}$. Suppose that there are $L \in \mathbb{Z}^+$ propagation steps. In each step $l \in [L]$, the hidden representation $H^{(l)} \in \mathbb{R}^{d_{l-1} \times d_l}$ is calculated by

$$H^{(l)} = \text{ReLu}(\mathbf{D}_v^{-\frac{1}{2}}\mathbf{JTD}_e^{-1}\mathbf{J}^\top\mathbf{D}_v^{-\frac{1}{2}}H^{(l)}\mathbf{W}^{(l)}),$$

where $\mathbf{W}^{(l)} \in \mathbb{R}^{d_{l-1} \times d_l}$. $\mathbf{D}_v \in \mathbb{R}^{N \times N}$ and $\mathbf{D}_e \in \mathbb{R}^{K \times K}$ are diagonal matrices of the vertex and hyperedge degree, respectively. The readout layer is defined as $\text{HGNN}(A) = \frac{1}{N}\mathbf{1}_N H^{(L)}\mathbf{W}^{(L+1)}$, where $\mathbf{W}^{(L+1)} \in \mathbb{R}^{d_L \times C}$ and $\mathbf{1}_N$ is an all-one vector. Let the maximum hidden dimension be $h := \max_{l \in [L]} d_l$. We have the following results for HGNN.

**Lemma 10.** *Consider the* HGNN *with* $L+1$ *layers and parameters* $\mathbf{w} = (\mathbf{W}^{(1)}, \ldots, \mathbf{W}^{(L+1)})$. *For each* $\mathbf{w}$, *each perturbation* $\mathbf{u} = (\mathbf{U}^{(1)}, \ldots, \mathbf{U}^{(L+1)})$ *on* $\mathbf{w}$ *such that* $\max_{i \in [L+1]} \frac{\|\mathbf{U}^{(i)}\|}{\|\mathbf{W}^{(i)}\|} \leq \frac{1}{L+1}$, *and*

*each input* $A \in \mathcal{S}$, *we have*

$$\|\text{HGNN}_{\mathbf{w}+\mathbf{u}}(A) - \text{HGNN}_\mathbf{w}(A)\|_2 \leq$$

$$\leq eB(C^{1/2}D^{1/2}RM)^L \left(\prod_{i=1}^{L+1} \|W^{(i)}\|\right)\left(\sum_{i=1}^{L+1} \frac{\|\mathbf{U}^{(i)}\|}{\|\mathbf{W}^{(i)}\|}\right),$$

*where* $C = \max_i \mathbf{T}_{ii}$.

**Theorem 6.** *For* HGNN *parameted by* $\mathbf{w}$ *with* $L+1$ *layers and each* $\delta, \gamma > 0$, *with probability at least* $1-\delta$ *over a training set* $S$ *of size* $m$, *for any fixed* $\mathbf{w}$, *we have*

$$\mathcal{L}_\mathcal{D}(\text{HGNN}_\mathbf{w}) \leq \mathcal{L}_{S,\gamma}(\text{HGNN}_\mathbf{w})$$

$$+ O\left(\sqrt{\frac{L^2 B^2 h \ln(Lh)(CD^{\frac{1}{2}}RM)^L \mathcal{W}_1 \mathcal{W}_2 + \log\frac{mL}{\sigma}}{\gamma^2 m}}\right),$$

*where* $\mathcal{W}_1 = \prod_{i=1}^{L+1} \|\mathbf{W}^{(i)}\|^2$ *and* $\mathcal{W}_2 = \sum_{i=1}^{L+1} \frac{\|\mathbf{W}^{(i)}\|_F^2}{\|\mathbf{W}^{(i)}\|^2}$.

## D Additional materials for experiments

### D.1 Experiment settings

**Sample generation for real dataset.** Table 5 shows the statistics on real datasets. DBLP-v1 consists of bibliography data in computer science. The papers in DBLP-v1 are represented as a graph, where each node denotes a paper ID or a keyword and each edge denotes the citation relationship between papers or keyword relations. COL-LAB is a scientific collaboration dataset where in each graph, the researcher and its collaborators are nodes and an edge indicates collaboration between two researchers. Note that these datasets are benchmarks for graph classification tasks. To satisfy our experiment task, we construct the hyperedge using attribute-based hypergraph generation methods [29]. In particular, for DBLP-v1, we let each hyperedge include the node paper ID node and its related keywords nodes. For COLLAB, we let the hyperedge include all researchers who appear in the same work.

**Sample generation for synthetic dataset.** Table 4 shows the statistics on synthetic datasets. The basic graphs are randomly generated using the Erdos–Renyi (ER) [27] model and the Stochastic Block Model (SBM) [1] with 24 different settings, varying by the number of nodes, edge probability, and number of blocks. We generate a pool of 1000 basic graphs for each setting and form hypergraphs using a variation of the HyperPA method with different statistics (i.e., $N, M, R$) [15]. The number of classes is set to 3, and hypergraph labels are randomly assigned. Then we use the Wrap method to generate the label-specific features using the node structure information as input [67].

**Training settings.** All the experiments are trained with $2 \times$ NVIDIA RTX A4000. We use SGD as the optimizer for the model AllDeepSets and Adam as the optimizer for UniGCNs, M-GINs, T-MPHN, and HGNN+. For all datasets, the random train-test-valid split ratio is 0.5-0.3-0.2. We set the hidden dimension $h = 64$. The learning rate is chosen from $\{0.002, 0.01\}$. The batch size is 20 and the number of epochs is 100. We evaluate our models in four different propagation steps: 2, 4, 6, and 8. The $\gamma$ in margin loss is set to 0.25.

**Testing settings.** The empirical loss for UniGCN, M-IGN, and AllDeepSets on synthetic datasets is computed using the optimal

## Table 4: Statistics on the synthetic dataset

| Dataset | ER1 | ER2 | ER3 | ER4 | ER5 | ER6 | ER7 | ER8 | ER9 | ER10 | ER11 | ER12 |
|---|---|---|---|---|---|---|---|---|---|---|---|---|
| **N** | 200 | 200 | 200 | 200 | 400 | 400 | 400 | 400 | 600 | 600 | 600 | 600 |
| **max(K)** | 200 | 200 | 200 | 200 | 400 | 399 | 399 | 400 | 600 | 600 | 600 | 600 |
| **max(M)** | 20 | 20 | 40 | 40 | 40 | 40 | 60 | 60 | 60 | 60 | 80 | 80 |
| **max(R)** | 20 | 40 | 20 | 40 | 40 | 60 | 40 | 60 | 60 | 80 | 60 | 80 |
| **max(D)** | 166 | 166 | 199 | 199 | 375 | 376 | 399 | 399 | 587 | 584 | 599 | 599 |
| Dataset | SBM1 | SBM2 | SBM3 | SBM4 | SBM5 | SBM6 | SBM7 | SBM8 | SBM9 | SBM10 | SBM11 | SBM12 |
| **N** | 200 | 200 | 200 | 200 | 400 | 400 | 400 | 400 | 600 | 600 | 600 | 600 |
| **max(K)** | 200 | 200 | 200 | 200 | 400 | 399 | 400 | 400 | 600 | 600 | 597 | 600 |
| **max(M)** | 20 | 20 | 40 | 40 | 40 | 40 | 60 | 60 | 60 | 60 | 80 | 80 |
| **max(R)** | 20 | 40 | 20 | 40 | 40 | 60 | 40 | 60 | 60 | 80 | 60 | 80 |
| **max(D)** | 165 | 163 | 199 | 199 | 374 | 377 | 399 | 399 | 586 | 587 | 599 | 599 |

## Table 5: Statistics on the real dataset

| Dataset | max(N) | max(K) | max(M) | max(R) | max(D) | features | sample size | classes |
|---|---|---|---|---|---|---|---|---|
| **DBLP_v1** | 39 | 39 | 9 | 16 | 32 | 5 | 560 | 2 |
| **COLLAB** | 76 | 37 | 17 | 19 | 53 | 5 | 1000 | 3 |

Monte Carlo algorithm [13]. On real datasets, their empirical loss is calculated by averaging the results over five runs, due to the limited sample size. For T-MPHN, the empirical loss is calculated by averaging over five runs on both synthetic and real datasets. The results of all models with random parameters on synthetic data are obtained by averaging across five test datasets randomly selected from the sample pool. Additionally, for each subgraph shown in Figures 2 and 3, the corresponding experiments are conducted including synthetic and real datasets, with each subfigure displaying the results from twelve different datasets (synthetic graph datasets with models of trained parameters) and ten repeated runs (real graph datasets with models with random parameters). In particular, in Figure 3, we report the optimal results over ten repeated experiments in each subgraph on models with both trained and random parameters. In Figures 5 and 6, the experiments are performed on real datasets, with each subfigure depicting the results from a single dataset repeated ten times.

### D.2 Bounds calculation

We compute the bound values for the learned model saved at the end of the training. In particular, for the considered models, we compute the following quantities.

$$B_{\text{UniGCN}} = \mathcal{L}_{S,\gamma}(\text{UniGCN}_{\mathbf{w}}) +$$

$$\sqrt{\frac{32e^4 B^2 DRM^L (L_1)^2 h \ln (4h(L_1)) \mathcal{W}_1 \mathcal{W}_2 + \log \frac{m(\frac{L_1}{2}(m^{1/(L_1)} - 1))}{\delta}}{\gamma^2 m}}$$

$$B_{\text{AllDeepSets}} = \mathcal{L}_{S,\gamma}(\text{AllDeepSets}_{\mathbf{w}}) +$$

$$\sqrt{\frac{32e^4 B^2 C^{2L} (L_2)^2 h \ln (4h(L_2)) \mathcal{W}_1 \mathcal{W}_2 + \log \frac{m(\frac{L_2}{2}(m^{1/2(L_2)} - 1))}{\delta}}{\gamma^2 m}}$$

$$B_{\text{M-IGN}} = \mathcal{L}_{S,\gamma}(\text{M-IGN}_{\mathbf{w}}) +$$

$$\sqrt{\frac{32e^8 (MD)^{2L} B^2 E^{(2,L)} \mathcal{W}_1 \mathcal{W}_2 + \log \frac{m(\frac{L+2}{2}(m^{1/2(L+2)} - 1))}{\delta}}{\gamma^2 m}}$$

$$B_{\text{T-MPHN}} = \mathcal{L}_{S,\gamma}(\text{T-MPHN}_{\mathbf{w}}) + \sqrt{\frac{144L^2 h \ln h \sum_{i=1}^{L+1} \|\mathbf{W}\|_F^2 + \log \frac{mL}{\sigma}}{\gamma^2 m + \|\mathbf{W}^{(L+1)}\|^2 m}},$$

where $L + 1 = L + 1, L_2 = 2L + 1, C = \max(M, R)$. $\mathcal{W}_1 = \prod_{i=1}^{L^*} \|\mathbf{W}^{(i)}\|_2^2$, and $\mathcal{W}_2 = \sum_{i=1}^{L^*} \frac{\|\mathbf{W}^{(i)}\|_F^2}{\|\mathbf{W}^{(i)}\|_2^2}$.

## E Further discussion on node classification task

Consider the user behavior prediction problem in a social network, where the goal is to predict whether a user will take a specific action (e.g., share a post, like content, or make a purchase). Given pairs representing the initial status of each user, which refers to the set of features or attributes associated with each user at the beginning of the observation period, and their final behavior, we aim to learn a predictor from these pairs to forecast future user behavior based on new user status data. Formally, we follow the notation used in the hypergraph classification task, where we are given a hypergraph $\mathcal{G} = (\mathcal{V}, \mathcal{E})$ and node features $\mathbf{X}$, representing the initial status of the users. The input domain is defined as $\mathcal{A}$, containing all pairs $A = (\mathcal{G}, \mathbf{X})$, and the output domain is $\mathbb{R}^{N \times C}$, where $C \in \mathbb{Z}^+$ indicates the number of behavior types. Our objective is to learn a predictor $f : \mathcal{A} \to \mathbb{R}^{N \times C}$ from $m$ samples, such that for new user status, the true error $\mathcal{L}_{\mathcal{D}}$ is minimized, assuming the input pairs follow a distribution $\mathcal{D}$. We define the true error over distribution $\mathcal{D}$ as

$$\mathcal{L}_{\mathcal{D}}(f_{\mathbf{w}}) = \mathbb{E}_{A \sim \mathcal{D}} \left[ \frac{1}{N} \sum_{i}^{N} \mathbb{1} \left( f_{\mathbf{w}}(v)[y_v] \leq \max_{j \neq y_v} f_{\mathbf{w}}(v)[j] \right) \right],$$

and the empirical margin loss over the labeled nodes $S = \{(v_i, y_{v_i})\}$ as

$$\mathcal{L}_{S,\gamma}(f_{\mathbf{w}}) = \frac{1}{mN} \sum_{i=1}^{m} \sum_{i=1}^{N} \mathbb{1}\left(f_{\mathbf{w}}(v_i)[y_{v_i}] \leq \gamma + \max_{j \neq y_{v_i}} f_{\mathbf{w}}(v_i)[j]\right),$$

The following is the generalization bound on the behavior prediction task with UniGCN.

THEOREM 7. *For* UniGCN$_{\mathbf{w}}$ *with* $L+1$ *layers, and for each* $\delta, \gamma > 0$*, with probability at least* $1 - \delta$ *over a training set* $S$ *of size* $m$*, for any fixed* $\mathbf{w}$*, we have*

$$\mathcal{L}_{\mathcal{D}}(f_{\mathbf{w}}) \leq \mathcal{L}_{S,\gamma}(f_{\mathbf{w}})$$

$$+ O\left(\sqrt{\frac{L^2 B^2 h \ln(Lh)(RMD)^L \mathcal{W}_1 \mathcal{W}_2 + \ln(mN) + \ln\left(\frac{mL}{\sigma}\right)}{\gamma^2 m}}\right),$$

*where* $\mathcal{W}_1 = \prod_{i=1}^{L+1} \|\mathbf{W}^{(i)}\|^2$ *and* $\mathcal{W}_2 = \sum_{i=1}^{L+1} \frac{\|\mathbf{W}^{(i)}\|_F^2}{\|\mathbf{W}^{(i)}\|^2}$.

PROOF. The proof follows from extending the generalization bound for hypergraph classification to the node classification setting, accounting for the per-node analysis, and applying a union bound over all nodes and classes.

First, consider a class $\mathcal{F}$ consisting of functions UniGCN$_{\mathbf{w}}(v) : \mathcal{A} \to \mathbb{R}^C$. Each function in $\mathcal{F}$ corresponds to a node's prediction. Then we can apply the generalization bound results given by Theorem 1, we have

$$\mathcal{L}_{\mathcal{D}}(\text{UniGCN}_{\mathbf{w}}(v)) \leq \mathcal{L}_{S,\gamma}(\text{UniGCN}_{\mathbf{w}}(v))$$

$$+ O\left(\sqrt{\frac{L^2 B^2 h \ln(Lh)(RMD)^L \mathcal{W}_1 \mathcal{W}_2 + \log\frac{mL}{\sigma}}{\gamma^2 m}}\right),$$

where $\mathcal{W}_1 = \prod_{i=1}^{L+1} \|\mathbf{W}^{(i)}\|^2$ and $\mathcal{W}_2 = \sum_{i=1}^{L+1} \frac{\|\mathbf{W}^{(i)}\|_F^2}{\|\mathbf{W}^{(i)}\|^2}$.

Since we have $N$ nodes in each sample and $m$ samples, the total number of events (i.e., the bounds holding for each node in each sample) is $mN$. To ensure that the generalization bound holds simultaneously for all nodes across all samples with probability at least $1 - \delta$, we apply the union bound. We set the failure probability for each event to $\delta' = \frac{\delta}{mN}$. The failure probability $\delta'$ appears inside a logarithmic term in the concentration inequalities used to derive the generalization bound. Specifically, the $\ln\left(\frac{1}{\delta'}\right)$ term becomes

$$\ln\left(\frac{1}{\delta'}\right) = \ln\left(\frac{mN}{\delta}\right) = \ln(mN) + \ln\left(\frac{1}{\delta}\right).$$

Since the bound in single now holds for all nodes in all samples with probability at least $1 - \delta$, we can sum over all nodes to obtain the overall bound. Therefore, we have our results. □

## F Additional results

Table 6 shows the results on real datasets. Table 7 report the results of HGNN models on synthetic datasets. Table 8 shows the additional results on synthetic datasets. Table 9 and Table 10 report the results of T-MPHN models on synthetic datasets. Table 10 reports the results of T-MPHN models on synthetic datasets. Figure 4 displays the additional results on consistency between empirical loss and theoretical bounds. Figures 5 and 6 show the main results on the consistency of real datasets. We put all the tables in Sec

**Table 6: Main results on real datasets.**

| Model | L | DBLP | | Collab | |
|---|---|---|---|---|---|
| | | **Emp** | **Theory** | **Emp** | **Theory** |
| **UniGCN** | 2 | 0.27 ± 0.07 | 1.54e+10 ± 6.76e+08 | 0.23 ± 0.20 | 1.31e+11 ± 5.90e+09 |
| | 4 | 0.23 ± 0.14 | 5.19e+17 ± 7.94e+16 | 0.35 ± 0.04 | 4.07e+19 ± 5.06e+18 |
| | 6 | 0.15 ± 0.06 | 1.29e+25 ± 2.00e+24 | 0.31 ± 0.13 | 1.25e+28 ± 2.35e+27 |
| **AllDeepSets** | 2 | 0.24 ± 0.13 | 6.44e+09 ± 4.63e+08 | 0.17 ± 0.16 | 1.79e+10 ± 5.76e+08 |
| | 4 | 0.37 ± 0.25 | 8.77e+16 ± 7.33e+15 | 0.25 ± 0.16 | 5.36e+17 ± 7.54e+16 |
| | 6 | 0.27 ± 0.26 | 9.48e+23 ± 1.86e+26 | 0.18 ± 0.19 | 2.41e+25 ± 2.61e+24 |
| **M-GIN** | 2 | 0.29 ± 0.14 | 2.42e+06 ± 4.36e+05 | 0.09 ± 0.12 | 2.61e+06 ± 4.87e+05 |
| | 4 | 0.09 ± 0.11 | 1.52e+10 ± 6.50e+09 | 0.01 ± 0.01 | 6.69e+09 ± 2.73e+09 |
| | 6 | 0.09 ± 0.12 | 9.88e+12 ± 6.00e+12 | 0.18 ± 0.22 | 5.58e+12 ± 4.08e+12 |
| **T-MPHN** | 2 | 0.34 ± 0.04 | 1.00e+00 ± 1.38e-01 | 0.33 ± 0.05 | 1.03e+00 ± 6.73e-02 |
| | 4 | 0.27 ± 0.06 | 4.41e+00 ± 4.90e-01 | 0.30 ± 0.05 | 3.82e+00 ± 1.67e-01 |
| | 6 | 0.29 ± 0.01 | 2.65e+01 ± 7.06e-02 | 0.32 ± 0.04 | 2.63e+01 ± 6.09e-02 |
| **HGNN+** | 2 | 0.33 ± 0.03 | 1.00e+08 ± 7.85E+05 | 0.24 ± 0.16 | 9.84e+07 ± 8.18E+04 |
| | 4 | 0.24 ± 0.16 | 1.37e+13 ± 1.23e+10 | 0.34 ± 0.22 | 9.84e+07 ± 1.65E+06 |
| | 6 | 0.34 ± 0.23 | 1.46e+18 ± 6.62e+23 | 0.26 ± 0.07 | 9.84e+07 ± 2.65E+23 |

**Table 7: Main results of HGNN+ on ER and SBM datasets with $L \in \{2, 4, 6\}$**

| L | ER1 | | ER2 | | ER3 | | ER4 | | ER5 | | ER6 | |
|---|---|---|---|---|---|---|---|---|---|---|---|---|
| | Emp | Theory | Emp | Theory | Emp | Theory | Emp | Theory | Emp | Theory | Emp | Theory |
| 2 | 0.22 | 6.84E+07 | 0.30 | 1.56E+08 | 0.36 | 1.88E+08 | 0.26 | 3.58E+08 | 0.19 | 7.04E+08 | 0.25 | 9.84E+08 |
| 4 | 0.26 | 9.11E+12 | 0.28 | 3.52E+13 | 0.15 | 5.94E+13 | 0.27 | 2.11E+14 | 0.26 | 8.07E+14 | 0.31 | 1.64E+15 |
| 6 | 0.29 | 9.82E+17 | 0.19 | 7.40E+18 | 0.27 | 1.29E+19 | 0.37 | 1.08E+20 | 0.41 | 8.07E+20 | 0.26 | 2.48E+21 |

| L | ER7 | | ER8 | | ER9 | | ER10 | | ER11 | | ER12 | |
|---|---|---|---|---|---|---|---|---|---|---|---|---|
| | Emp | Theory | Emp | Theory | Emp | Theory | Emp | Theory | Emp | Theory | Emp | Theory |
| 2 | 0.20 | 1.05E+09 | 0.79 | 1.55E+09 | 0.79 | 2.48E+09 | 0.23 | 3.26E+09 | 0.32 | 3.28E+09 | 0.19 | 3.88E+09 |
| 4 | 0.24 | 1.94E+15 | 0.63 | 4.49E+15 | 0.54 | 1.01E+16 | 0.26 | 1.78E+16 | 0.25 | 1.82E+16 | 0.32 | 3.32E+16 |
| 6 | 0.26 | 3.07E+21 | 0.67 | 1.04E+22 | 0.57 | 3.42E+22 | 0.28 | 7.82E+22 | 0.21 | 8.69E+22 | 0.35 | 1.91E+23 |

| L | SBM1 | | SBM2 | | SBM3 | | SBM4 | | SBM5 | | SBM6 | |
|---|---|---|---|---|---|---|---|---|---|---|---|---|
| | Emp | Theory | Emp | Theory | Emp | Theory | Emp | Theory | Emp | Theory | Emp | Theory |
| 2 | 0.20 | 7.14E+07 | 0.26 | 1.47E+08 | 0.30 | 1.80E+08 | 0.29 | 3.74E+08 | 0.21 | 6.73E+08 | 0.26 | 1.03E+09 |
| 4 | 0.29 | 1.01E+13 | 0.36 | 3.66E+13 | 0.26 | 5.54E+13 | 0.31 | 2.19E+14 | 0.32 | 7.84E+14 | 0.23 | 1.76E+15 |
| 6 | 0.26 | 1.03E+18 | 0.30 | 8.37E+18 | 0.28 | 1.39E+19 | 0.32 | 1.15E+20 | 0.36 | 7.65E+20 | 0.25 | 2.67E+21 |

| L | SBM7 | | SBM8 | | SBM9 | | SBM10 | | SBM11 | | SBM12 | |
|---|---|---|---|---|---|---|---|---|---|---|---|---|
| | Emp | Theory | Emp | Theory | Emp | Theory | Emp | Theory | Emp | Theory | Emp | Theory |
| 2 | 0.33 | 1.09E+09 | 0.25 | 1.56E+09 | 0.32 | 2.42E+09 | 0.21 | 2.91E+09 | 0.80 | 3.09E+09 | 0.34 | 4.35E+09 |
| 4 | 0.34 | 2.08E+15 | 0.26 | 4.09E+15 | 0.35 | 9.97E+15 | 0.25 | 1.75E+16 | 0.71 | 1.89E+16 | 0.41 | 3.28E+16 |
| 6 | 0.30 | 3.37E+21 | 0.27 | 1.13E+22 | 0.35 | 3.30E+22 | 0.35 | 7.86E+22 | 0.74 | 8.39E+22 | 0.27 | 2.01E+23 |

Table 8: Addition results of UniGCN, M-IGN, and AllDeepSet.

| Model | L | ER3 | | ER4 | | ER7 | | ER8 | | ER11 | | ER12 | |
|---|---|---|---|---|---|---|---|---|---|---|---|---|---|
| | | Emp | Theory | Emp | Theory | Emp | Theory | Emp | Theory | Emp | Theory | Emp | Theory |
| UniGCN | 2 | 0.00 | 1.52E+09 | 0.07 | 3.65E+09 | 0.16 | 9.40E+09 | 0.14 | 1.13E+10 | 0.13 | 2.44E+10 | 0.23 | 3.25E+10 |
| | 4 | 0.02 | 1.61E+15 | 0.15 | 5.01E+15 | 0.56 | 5.43E+16 | 0.08 | 1.07E+17 | 0.14 | 5.77E+17 | 0.24 | 8.89E+17 |
| | 6 | 0.05 | 1.79E+21 | 0.05 | 1.34E+22 | 0.07 | 2.41E+23 | 0.37 | 5.75E+23 | 0.39 | 4.08E+24 | 0.28 | 1.07E+25 |
| M-IGN | 2 | 0.03 | 1.26E+13 | 0.02 | 1.13E+13 | 0.04 | 1.08E+14 | 0.12 | 1.09E+14 | 0.16 | 4.19E+14 | 0.16 | 4.42E+14 |
| | 4 | 0.02 | 2.18E+21 | 0.05 | 2.44E+21 | 0.10 | 2.04E+23 | 0.13 | 1.86E+23 | 0.16 | 2.70E+24 | 0.13 | 2.68E+24 |
| | 6 | 0.06 | 1.21E+29 | 0.05 | 1.68E+29 | 0.27 | 1.25E+32 | 0.12 | 1.02E+32 | 0.20 | 9.16E+33 | 0.24 | 1.02E+34 |
| AllDeepSet | 2 | 0.04 | 5.92E+08 | 0.08 | 2.32E+08 | 0.08 | 4.42E+08 | 0.09 | 6.86E+08 | 0.14 | 7.71E+08 | 0.22 | 1.54E+09 |
| | 4 | 0.02 | 4.13E+13 | 0.04 | 5.92E+13 | 0.12 | 1.52E+13 | 0.13 | 4.97E+13 | 0.27 | 6.19E+13 | 0.29 | 1.12E+14 |
| | 6 | 0.08 | 3.53E+16 | 0.08 | 4.29E+17 | 0.14 | 1.36E+19 | 0.15 | 1.54E+18 | 0.24 | 3.05E+18 | 0.31 | 8.23E+18 |

| Model | L | SBM3 | | SBM4 | | SBM7 | | SBM8 | | SBM11 | | SBM12 | |
|---|---|---|---|---|---|---|---|---|---|---|---|---|---|
| | | Emp | Theory | Emp | Theory | Emp | Theory | Emp | Theory | Emp | Theory | Emp | Theory |
| UniGCN | 2 | 0.01 | 1.80E+09 | 0.05 | 3.58E+09 | 0.04 | 9.57E+09 | 0.07 | 1.26E+10 | 0.06 | 3.40E+10 | 0.32 | 3.77E+10 |
| | 4 | 0.01 | 1.92E+15 | 0.08 | 5.30E+15 | 0.12 | 7.46E+16 | 0.18 | 1.28E+17 | 0.18 | 3.99E+17 | 0.33 | 6.03E+17 |
| | 6 | 0.07 | 1.13E+21 | 0.08 | 1.16E+22 | 0.25 | 2.56E+23 | 0.17 | 1.30E+24 | 0.33 | 7.74E+24 | 0.26 | 1.85E+25 |
| M-IGN | 2 | 0.03 | 1.05E+13 | 0.08 | 1.19E+13 | 0.16 | 1.01E+14 | 0.17 | 1.21E+14 | 0.11 | 4.58E+14 | 0.12 | 4.58E+14 |
| | 4 | 0.02 | 2.60E+21 | 0.13 | 3.00E+21 | 0.15 | 2.02E+23 | 0.13 | 1.64E+23 | 0.25 | 4.00E+24 | 0.24 | 3.74E+24 |
| | 6 | 0.23 | 1.92E+29 | 0.17 | 1.95E+29 | 0.13 | 7.90E+31 | 0.19 | 8.56E+31 | 0.13 | 7.69E+33 | 0.29 | 7.61E+33 |
| AllDeepSet | 2 | 0.02 | 5.93E+08 | 0.03 | 2.82E+08 | 0.11 | 3.93E+08 | 0.12 | 5.79E+08 | 0.24 | 8.56E+08 | 0.27 | 7.74E+08 |
| | 4 | 0.02 | 2.58E+13 | 0.05 | 8.56E+13 | 0.16 | 2.40E+13 | 0.22 | 4.25E+13 | 0.33 | 7.31E+13 | 0.26 | 1.34E+14 |
| | 6 | 0.02 | 1.62E+16 | 0.14 | 1.70E+17 | 0.25 | 3.04E+17 | 0.21 | 1.59E+18 | 0.25 | 4.69E+18 | 0.33 | 1.35E+19 |

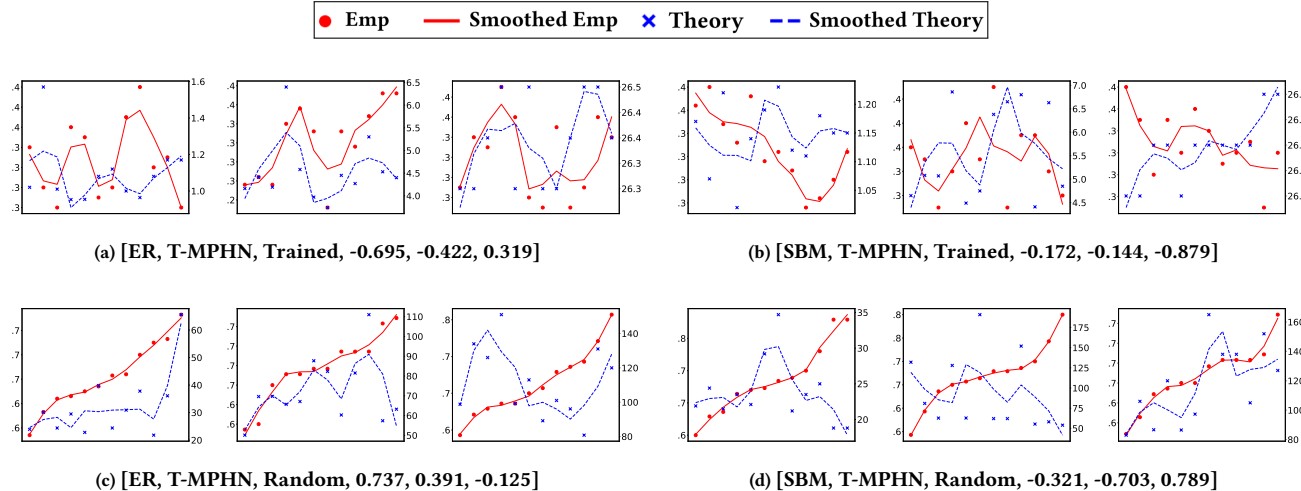

(a) [ER, T-MPHN, Trained, -0.695, -0.422, 0.319]    (b) [SBM, T-MPHN, Trained, -0.172, -0.144, -0.879]

(c) [ER, T-MPHN, Random, 0.737, 0.391, -0.125]    (d) [SBM, T-MPHN, Random, -0.321, -0.703, 0.789]

Figure 4: Consistency between empirical loss (Emp) and theoretical bounds (Theory). Each subgroup labeled by [graph type, model, parameter conditions, $r_2$, $r_4$, $r_6$] shows the empirical loss, theoretical bound, and their curves via Savitzky-Golay filter [53] of one graph type (i.e., ER or SBM) and one model (i.e., T-MPHN, and HGNN+) with trained ((a), (b), (e), and (f)) and random parameters ((c), (d), (g), and (h)), where each figure plots the results of twelve datasets; the figures, from left to right, show the results with 2, 4 and 6 propagation steps, where $r_2$, $r_4$, and $r_6 \in [-1, 1]$ are the Pearson correlation coefficients between the two sets of points in each figure – higher $r$ indicating stronger positive correlation.

**Table 9: Results of T-MPHN on ER datasets with $h \in \{64, 128\}$**

| h | L | ER1 | | ER2 | | ER3 | |
|---|---|-----|--|-----|--|-----|--|
| | | Emp | Theory | Emp | Theory | Emp | Theory |
| | 2 | 0.34 ± 0.04 | 1.00e+00 ± 1.38e-01 | 0.30 ± 0.03 | 9.53e-01 ± 3.32e-02 | 0.28 ± 0.03 | 1.02e+00 ± 7.85e-02 |
| 64 | 4 | 0.27 ± 0.07 | 4.41e+00 ± 4.90e-01 | 0.28 ± 0.04 | 6.40e+00 ± 1.20e+00 | 0.27 ± 0.06 | 4.19e+00 ± 2.74e-01 |
| | 6 | 0.29 ± 0.02 | 2.65e+01 ± 7.06e-02 | 0.34 ± 0.04 | 2.63e+01 ± 4.84e-02 | 0.33 ± 0.02 | 2.65e+01 ± 1.73e-01 |
| | 2 | 0.28 ± 0.03 | 6.22e-01 ± 3.90e-03 | 0.37 ± 0.02 | 6.99e-01 ± 1.03e-02 | 0.30 ± 0.00 | 5.84e-01 ± 1.32e-02 |
| 128 | 4 | 0.32 ± 0.02 | 7.69e+00 ± 1.37e+00 | 0.32 ± 0.03 | 3.78e+00 ± 1.84e-01 | 0.36 ± 0.01 | 3.07e+00 ± 1.63e-01 |
| | 6 | 0.36 ± 0.03 | 2.77e+01 ± 3.22e-02 | 0.32 ± 0.04 | 2.74e+01 ± 1.09e-01 | 0.30 ± 0.03 | 2.75e+01 ± 3.09e-02 |

| h | L | ER4 | | ER5 | | ER6 | |
|---|---|-----|--|-----|--|-----|--|
| | | Emp | Theory | Emp | Theory | Emp | Theory |
| | 2 | 0.36 ± 0.02 | 1.08e+00 ± 1.03e-01 | 0.35 ± 0.04 | 9.64e-01 ± 2.86e-02 | 0.29 ± 0.02 | 1.01e+00 ± 6.24e-02 |
| 64 | 4 | 0.35 ± 0.05 | 4.45e+00 ± 1.02e-01 | 0.37 ± 0.05 | 5.30e+00 ± 1.20e+00 | 0.34 ± 0.03 | 3.97e+00 ± 1.47e-01 |
| | 6 | 0.39 ± 0.03 | 2.64e+01 ± 7.20e-02 | 0.36 ± 0.07 | 2.65e+01 ± 1.49e-01 | 0.28 ± 0.07 | 2.64e+01 ± 7.97e-02 |
| | 2 | 0.27 ± 0.04 | 6.31e-01 ± 3.82e-03 | 0.35 ± 0.05 | 7.06e-01 ± 2.78e-03 | 0.34 ± 0.04 | 7.16e-01 ± 5.10e-03 |
| 128 | 4 | 0.37 ± 0.01 | 3.00e+00 ± 1.57e-01 | 0.33 ± 0.02 | 3.56e+00 ± 6.21e-02 | 0.32 ± 0.01 | 3.07e+00 ± 2.06e-01 |
| | 6 | 0.32 ± 0.02 | 2.75e+01 ± 7.15e-02 | 0.26 ± 0.03 | 2.74e+01 ± 2.56e-02 | 0.29 ± 0.10 | 2.78e+01 ± 1.74e-02 |

| h | L | ER7 | | ER8 | | ER9 | |
|---|---|-----|--|-----|--|-----|--|
| | | Emp | Theory | Emp | Theory | Emp | Theory |
| | 2 | 0.30 ± 0.03 | 9.54e-01 ± 3.70e-02 | 0.37 ± 0.04 | 1.17e+00 ± 4.99e-02 | 0.40 ± 0.04 | 1.17e+00 ± 5.13e-02 |
| 64 | 4 | 0.24 ± 0.09 | 4.16e+00 ± 3.19e-01 | 0.34 ± 0.04 | 3.74e+00 ± 3.80e-02 | 0.32 ± 0.05 | 4.58e+00 ± 2.83e-02 |
| | 6 | 0.27 ± 0.02 | 2.63e+01 ± 2.40e-02 | 0.35 ± 0.06 | 2.64e+01 ± 8.39e-02 | 0.27 ± 0.08 | 2.63e+01 ± 1.96e-02 |
| | 2 | 0.32 ± 0.06 | 6.45e-01 ± 3.66e-02 | 0.32 ± 0.05 | 6.76e-01 ± 1.82e-02 | 0.32 ± 0.03 | 6.91e-01 ± 4.52e-02 |
| 128 | 4 | 0.25 ± 0.08 | 7.21e+00 ± 2.77e-01 | 0.36 ± 0.02 | 2.92e+00 ± 1.76e-01 | 0.34 ± 0.02 | 2.62e+00 ± 2.22e-02 |
| | 6 | 0.38 ± 0.05 | 2.77e+01 ± 3.82e-02 | 0.27 ± 0.08 | 2.77e+01 ± 1.01e-02 | 0.29 ± 0.06 | 2.75e+01 ± 2.96e-02 |

| h | L | ER10 | | ER11 | | ER12 | |
|---|---|------|--|------|--|------|--|
| | | Emp | Theory | Emp | Theory | Emp | Theory |
| | 2 | 0.32 ± 0.04 | 1.08e+00 ± 1.42e-01 | 0.33 ± 0.04 | 1.12e+00 ± 1.26e-01 | 0.28 ± 0.05 | 1.57e+00 ± 3.66e-01 |
| 64 | 4 | 0.36 ± 0.05 | 4.27e+00 ± 3.10e-01 | 0.39 ± 0.05 | 4.53e+00 ± 1.06e+00 | 0.39 ± 0.06 | 4.40e+00 ± 7.55e-01 |
| | 6 | 0.29 ± 0.04 | 2.63e+01 ± 2.14e-02 | 0.36 ± 0.02 | 2.65e+01 ± 1.77e-01 | 0.34 ± 0.03 | 2.63e+01 ± 1.13e-01 |
| | 2 | 0.31 ± 0.04 | 6.93e-01 ± 3.15e-03 | 0.31 ± 0.04 | 6.81e-01 ± 3.12e-03 | 0.31 ± 0.03 | 6.58e-01 ± 1.76e-03 |
| 128 | 4 | 0.32 ± 0.04 | 3.16e+00 ± 4.55e-02 | 0.27 ± 0.07 | 2.87e+00 ± 2.34e-01 | 0.29 ± 0.03 | 3.22e+00 ± 1.32e-02 |
| | 6 | 0.28 ± 0.05 | 2.76e+01 ± 6.81e-02 | 0.37 ± 0.02 | 2.76e+01 ± 6.87e-02 | 0.29 ± 0.02 | 2.73e+01 ± 7.03e-02 |

**Table 10: Main results of T-MPHN on SBM datasets with $h \in \{64, 128\}$**

| h | L | SBM1 | | SBM2 | | SBM3 | |
|---|---|---|---|---|---|---|---|
| | | **Emp** | **theory** | **Emp** | **theory** | **Emp** | **theory** |
| | 2 | 0.38 ± 0.036 | 1.18e+00 ± 2.25e-01 | 0.40 ± 0.03 | 1.15e+00 ± 8.52e-02 | 0.36 ± 0.07 | 1.11e+00 ± 1.23e-01 |
| **64** | **4** | 0.36 ± 0.023 | 6.65e+00 ± 3.88e+00 | 0.35 ± 0.03 | 4.75e+00 ± 8.55e-01 | 0.31 ± 0.06 | 4.65e+00 ± 5.28e-01 |
| | 6 | 0.38 ± 0.047 | 2.65e+01 ± 1.21e-01 | 0.35 ± 0.02 | 2.64e+01 ± 7.61e-02 | 0.30 ± 0.04 | 2.63e+01 ± 4.05e-02 |
| | 2 | 0.30 ± 0.024 | 6.10e-01 ± 4.13e-02 | 0.29 ± 0.05 | 6.61e-01 ± 1.93e-03 | 0.35 ± 0.04 | 6.93e-01 ± 2.14e-02 |
| **128** | **4** | 0.35 ± 0.026 | 7.15e+00 ± 9.78e-01 | 0.28 ± 0.02 | 3.15e+00 ± 1.86e-01 | 0.37 ± 0.06 | 3.12e+00 ± 1.39e-01 |
| | 6 | 0.32 ± 0.022 | 2.75e+01 ± 6.77e-02 | 0.30 ± 0.02 | 2.75e+01 ± 2.83e-02 | 0.40 ± 0.06 | 2.73e+01 ± 1.62e-02 |

| h | L | SBM4 | | SBM5 | | SBM6 | |
|---|---|---|---|---|---|---|---|
| | | **Emp** | **theory** | **Emp** | **theory** | **Emp** | **theory** |
| | 2 | 0.34 ± 0.02 | 1.12e+00 ± 1.08e-01 | 0.39 ± 0.04 | 1.15e+00 ± 1.11e-01 | 0.32 ± 0.05 | 1.14e+00 ± 1.22e-01 |
| **64** | **4** | 0.34 ± 0.03 | 6.86e+00 ± 4.78e+00 | 0.38 ± 0.05 | 6.63e+00 ± 3.91e+00 | 0.35 ± 0.03 | 6.38e+00 ± 3.95e+00 |
| | 6 | 0.35 ± 0.08 | 2.64e+01 ± 3.05e-02 | 0.32 ± 0.04 | 2.64e+01 ± 1.57e-01 | 0.36 ± 0.03 | 2.65e+01 ± 1.68e-01 |
| | 2 | 0.27 ± 0.03 | 6.51e-01 ± 3.84e-03 | 0.28 ± 0.03 | 6.46e-01 ± 2.57e-03 | 0.26 ± 0.06 | 5.90e-01 ± 4.96e-02 |
| **128** | **4** | 0.30 ± 0.03 | 7.13e+00 ± 3.29e-01 | 0.35 ± 0.04 | 3.10e+00 ± 1.79e-01 | 0.34 ± 0.04 | 2.90e+00 ± 1.56e-01 |
| | 6 | 0.28 ± 0.05 | 2.76e+01 ± 4.48e-02 | 0.35 ± 0.02 | 2.76e+01 ± 6.08e-02 | 0.36 ± 0.03 | 2.77e+01 ± 7.68e-02 |

| h | L | SBM7 | | SBM8 | | SBM9 | |
|---|---|---|---|---|---|---|---|
| | | **Emp** | **Theory** | **Emp** | **Theory** | **Emp** | **Theory** |
| | 2 | 0.33 ± 0.034 | 1.19e+00 ± 2.03e-01 | 0.31 ± 0.03 | 1.02e+00 ± 4.36e-02 | 0.27 ± 0.06 | 1.17e+00 ± 1.76e-01 |
| **64** | **4** | 0.41 ± 0.030 | 4.85e+00 ± 9.76e-01 | 0.31 ± 0.03 | 5.08e+00 ± 1.06e+00 | 0.37 ± 0.04 | 6.80e+00 ± 3.79e+00 |
| | 6 | 0.34 ± 0.042 | 2.64e+01 ± 1.13e-01 | 0.31 ± 0.03 | 2.64e+01 ± 7.67e-02 | 0.32 ± 0.03 | 2.63e+01 ± 7.24e-02 |
| | 2 | 0.30 ± 0.034 | 6.68e-01 ± 4.19e-03 | 0.29 ± 0.02 | 6.63e-01 ± 1.08e-02 | 0.34 ± 0.08 | 6.72e-01 ± 2.97e-02 |
| **128** | **4** | 0.36 ± 0.048 | 2.95e+00 ± 1.57e-01 | 0.34 ± 0.03 | 3.17e+00 ± 1.65e-01 | 0.33 ± 0.02 | 2.97e+00 ± 2.70e-01 |
| | 6 | 0.32 ± 0.029 | 2.77e+01 ± 6.00e-02 | 0.35 ± 0.02 | 2.75e+01 ± 7.27e-02 | 0.31 ± 0.01 | 2.75e+01 ± 4.20e-02 |

| h | L | SBM10 | | SBM11 | | SBM12 | |
|---|---|---|---|---|---|---|---|
| | | **Emp** | **Theory** | **Emp** | **Theory** | **Emp** | **Theory** |
| | 2 | 0.28 ± 0.02 | 1.07e+00 ± 5.45e-02 | 0.30 ± 0.08 | 1.22e+00 ± 2.06e-01 | 0.33 ± 0.05 | 1.23e+00 ± 1.99e-01 |
| **64** | **4** | 0.37 ± 0.06 | 4.41e+00 ± 1.09e+00 | 0.34 ± 0.03 | 4.49e+00 ± 7.39e-01 | 0.32 ± 0.03 | 5.07e+00 ± 8.51e-01 |
| | 6 | 0.33 ± 0.05 | 2.64e+01 ± 1.08e-01 | 0.27 ± 0.04 | 2.63e+01 ± 8.27e-02 | 0.32 ± 0.04 | 2.64e+01 ± 1.35e-01 |
| | 2 | 0.41 ± 0.05 | 7.45e-01 ± 8.52e-03 | 0.33 ± 0.05 | 6.78e-01 ± 1.53e-02 | 0.30 ± 0.03 | 6.68e-01 ± 1.45e-03 |
| **128** | **4** | 0.37 ± 0.03 | 3.26e+00 ± 1.55e-01 | 0.32 ± 0.02 | 3.29e+00 ± 8.72e-02 | 0.32 ± 0.02 | 3.13e+00 ± 2.26e-01 |
| | 6 | 0.34 ± 0.02 | 2.74e+01 ± 5.96e-02 | 0.32 ± 0.05 | 2.76e+01 ± 3.67e-02 | 0.31 ± 0.04 | 2.75e+01 ± 5.64e-02 |

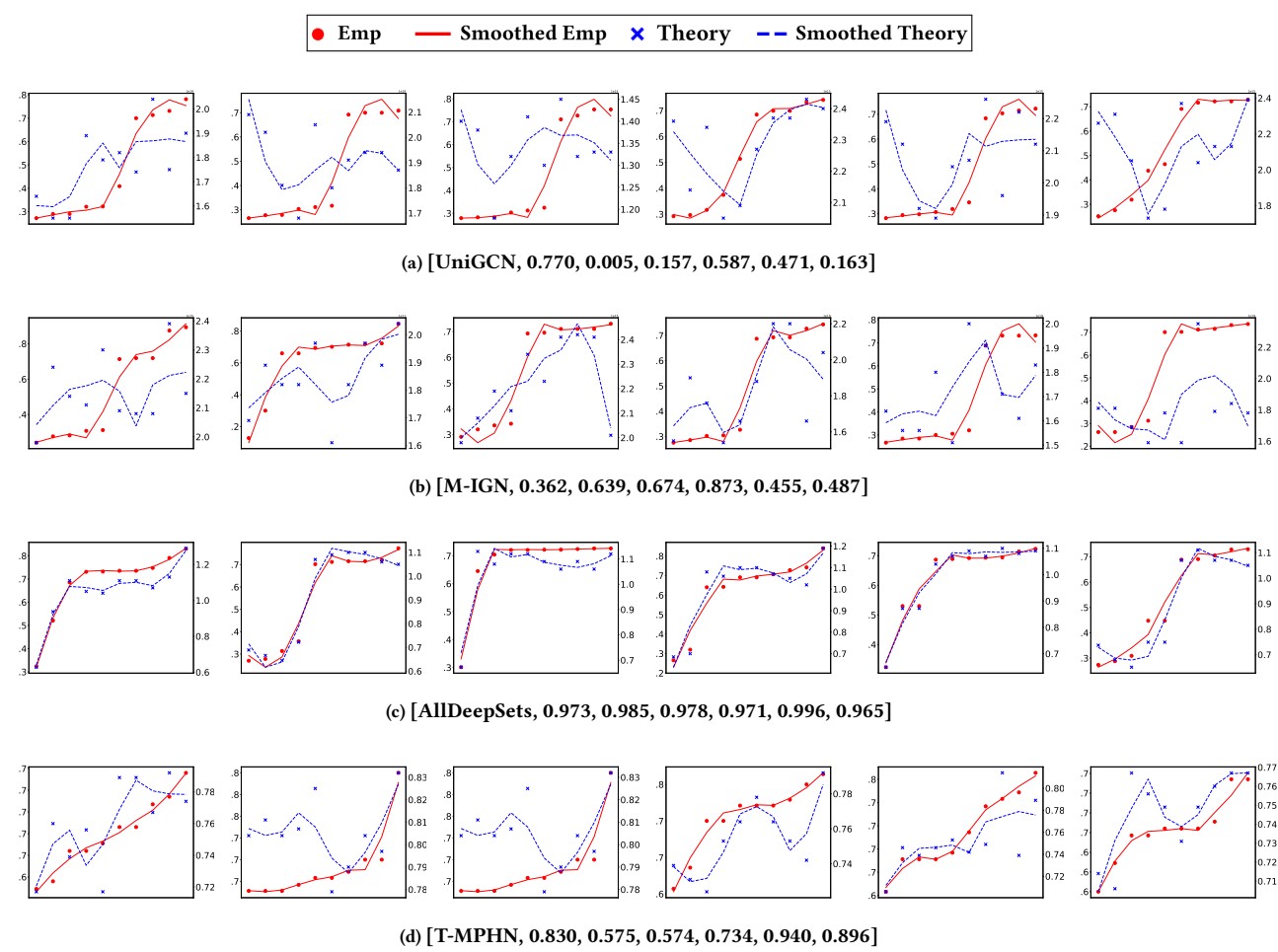

Figure 5: Results on DBLP. Each subgroup labeled by [model, $r_1, r_2, r_3, r_4, r_5, r_6$] shows the empirical loss, theoretical bound, and their curves via Savitzky-Golay filter; each figure plots the results over ten independent experiments with random initializations and such process is repeated for six times, where $r_i \in [-1, 1]$ for $i \in [6]$ is the Pearson correlation coefficient for the $i$-th figure (from left to right) – higher $r$ indicating stronger positive correlation.

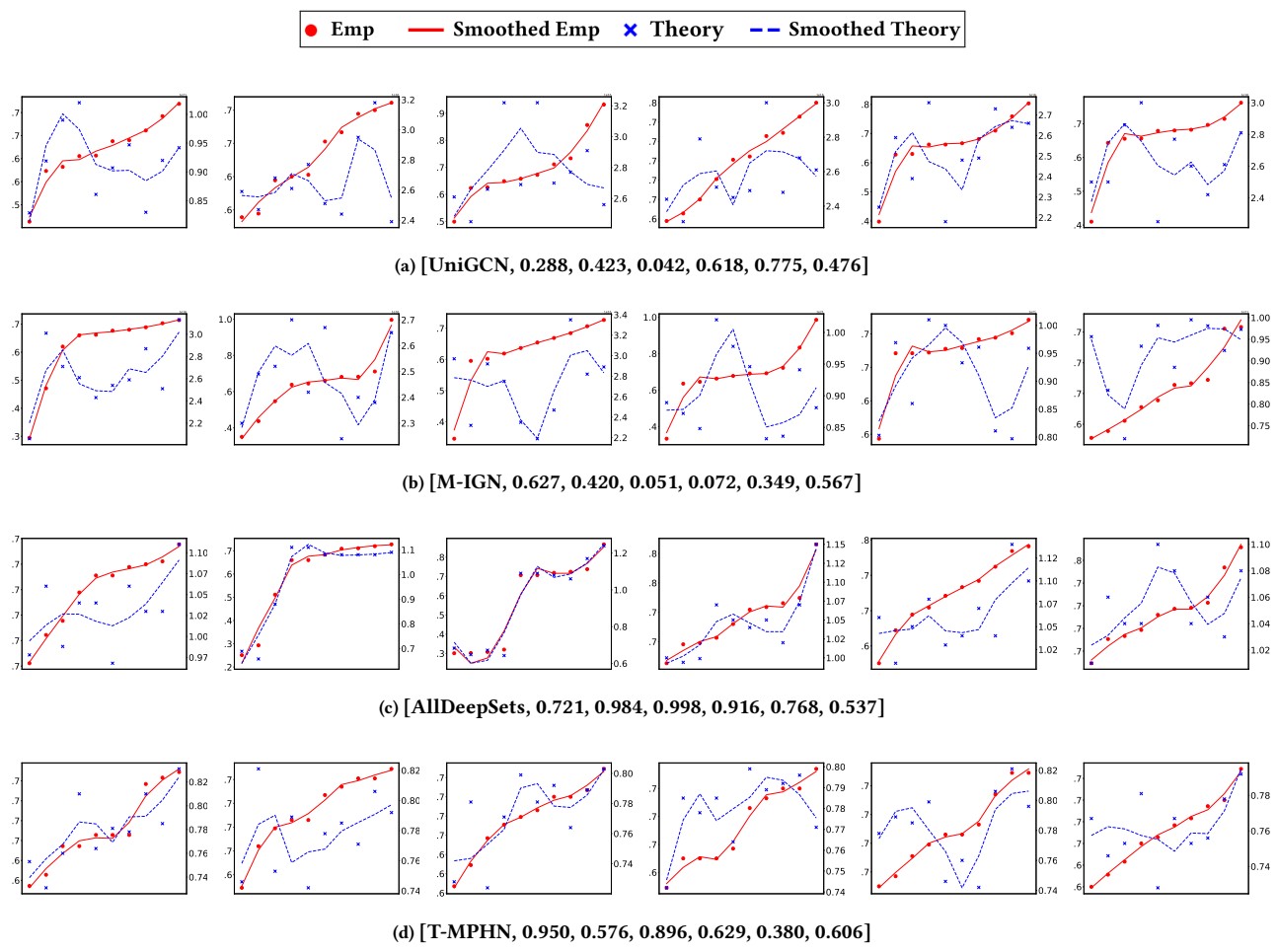

(a) [UniGCN, 0.288, 0.423, 0.042, 0.618, 0.775, 0.476]

(b) [M-IGN, 0.627, 0.420, 0.051, 0.072, 0.349, 0.567]

(c) [AllDeepSets, 0.721, 0.984, 0.998, 0.916, 0.768, 0.537]

(d) [T-MPHN, 0.950, 0.576, 0.896, 0.629, 0.380, 0.606]

Figure 6: Results on Collab. Each subgroup labeled by [model, $r_1, r_2, r_3, r_4, r_5, r_6$] shows the empirical loss, theoretical bound, and their curves via Savitzky-Golay filter; each figure plots the results over ten independent experiments with random initializations and such process is repeated for six times, where $r_i \in [-1, 1]$ for $i \in [6]$ is the Pearson correlation coefficient for the $i$-th figure (from left to right) – higher $r$ indicating stronger positive correlation.

