# OpenReview forum: "Generalization Performance of Hypergraph Neural Networks"
_ACM.org/TheWebConf/2025/Conference — WWW 2025 Oral_

### Official Review · Reviewer_axNx · 2024-11-29

**Novelty:** 6
**Technical Quality:** 6

**Review:**

This paper presents a theoretical analysis of the generalization performance of different types of hypergraph neural networks (HyperGNNs). Specifically, the authors adopt the PAC-Bayes framework to investigate the generalization bounds of four types of HyperGNNs based on GCNs, MPNNs, GIN, and Tensor-based. Moreover, they conduct an empirical study to verify the connections between the identified bounds and the empirical loss, showing a strong correlation between them.

Strengths:
- The authors provide an extensive examination of the theoretical bounds of HyperGNNs.
- The theoretical foundation is solid and thorough.
- Comprehensive analysis of the bounds' correlation with the models' empirical loss on real-world and synthetic datasets.

Weaknesses:
- There is no analysis of attention-based HyperGNNs, and the authors briefly discuss why they were not included.
- Some parts of the paper mentioned in the results are in the appendix (e.g., Figure 4 and Table 8), requiring readers to switch back and forth between the main text and the appendix, which disrupts the reading flow.

Minor:
- All methods analyzed are cited in their respective sections, but the citation for AllDeepSets is missing.

**Questions:**

- How can the bounds help improve architecture design?

- Can the authors briefly comment on the necessary changes to their approach to analyze attention-based methods? Is it necessary to adopt another framework instead of PAC-Bayes?

- Could the approach handle dynamic or evolving hypergraphs? If so, how might the bounds adapt? If not, what would be the challenges in such a scenario?

**Reviewer Confidence:**

3: The reviewer is confident but not certain that the evaluation is correct

**Scope:**

4: The work is relevant to the Web and to the track, and is of broad interest to the community

---

### Official Review · Reviewer_KjzA · 2024-11-29

**Novelty:** 6
**Technical Quality:** 6

**Review:**

This paper investigates the generalization performance of HyperGNNs using the PAC-Bayes framework. It first analyzes four representative HyperGNNs, including UniGCN, AllDeepSets, M-IGN, and T-MPHN, providing theoretical bounds on their generalization capabilities. Empirical results demonstrate a impressive correlation between theoretical bounds and empirical loss, suggesting that the bounds effectively capture the models' behavior. The study highlights the impact of hypergraph structure and spectral norms on generalization performance, offering insights for improving HyperGNNs.

Advantages:
1.The paper examines a diverse set of HyperGNN models, providing insights into the generalization properties of different architectural approaches.
2.Experiments on both synthetic and real-world datasets demonstrate the alignment between theoretical bounds and empirical performance.
3.The degradation of hypergraph theory can effectively reduce to the existing generalization work on graphs.

Disadvantages:
1. It would be better to verify how the correlation between empirical results and theory changes under different model hyperparameters, to illustrate whether the theoretical framework is robust.
2. Can the theory proposed in the paper be used to analyze whether certain types of HyperGNN designs exhibit better generalization potential, while others inherently lead to poorer performance?
3. The technique of deriving a generalization bound from the perturbation bound seems to be used frequently in the paper. It would be helpful if the authors could provide more preliminary details on this method.
4. Is it possible to improve the performance of existing models using this generalization bound? It would be useful if the authors discussed this possibility.

Minor Suggestions:
1.The figures are not clear enough, for example, the meaning of the axes in Figure 1 needs clarification.
2.The Related Works section could be organized into a table.
3.In Definition 1, the letters "i" and "I" seem to be used incorrectly.

**Questions:**

see disadvantages for details

**Reviewer Confidence:**

4: The reviewer is certain that the evaluation is correct and very familiar with the relevant literature

**Scope:**

4: The work is relevant to the Web and to the track, and is of broad interest to the community

---

### Official Review · Reviewer_d4KL · 2024-12-01

**Novelty:** 5
**Technical Quality:** 4

**Review:**

This paper investigates the generalization performance of Hypergraph Neural Networks (HyperGNNs) in hypergraph classification tasks, addressing the theoretical gap in understanding their learning capabilities.

Pros:
1. By extending the PAC-Bayes framework to handle high-order interactions and unique aggregation mechanisms of HyperGNNs, the work provides a foundational perspective on how structural properties and learned parameters influence model performance.
2. The work identifies key factors, such as spectral norms of weights, hyperedge size, and node degree, that impact generalization bounds. These insights provide actionable guidelines for designing and optimizing HyperGNNs, bridging the gap between theoretical research and practical implementation.
3.The paper is well-structured, with clear definitions of concepts, intuitive explanations of theoretical results, and logical connections between theoretical analysis and experimental outcomes. This clarity makes the paper accessible to both theoreticians and practitioners in the field of graph learning.

Cons:
1. The paper introduces numerous mathematical notations and symbols, some of which are used inconsistently or insufficiently explained, leading to potential confusion for readers. For example, the definitions of hypergraph statistics (e.g., D,M,R) and their relationships to different models could have been clarified further to avoid ambiguity across sections.
2. The empirical evaluation, while robust in correlating theoretical bounds with practical performance, lacks comparisons with alternative theoretical frameworks or models designed for similar tasks. This limits the ability to contextualize the contributions within the broader literature.
3. While the paper covers four types of HyperGNNs, the analysis for attention-based models (e.g., AllSetTransformer) is excluded due to challenges with non-linear dependencies and normalization issues. Addressing these gaps or discussing potential future extensions would have strengthened the paper.

**Questions:**

1. The paper derives generalization bounds for several HyperGNN architectures, emphasizing their dependence on hypergraph structure (e.g., node degree, hyperedge size, and shared hyperedges). However, in real-world scenarios, hypergraphs often exhibit diverse and dynamic structural properties that may not align with the fixed assumptions in the analysis. How does the proposed framework adapt to hypergraphs with highly heterogeneous or evolving structures, and how would the bounds account for such variability?

2. The analysis shows a strong correlation between theoretical bounds and empirical loss in experiments. However, for specific models like T-MPHN, the row-wise normalization introduces discrepancies in certain cases. How does the framework address such normalization effects, and could alternative strategies be explored to ensure that the theoretical bounds more accurately capture the behavior of models with complex normalization mechanisms?

**Reviewer Confidence:**

2: The reviewer is willing to defend the evaluation, but it is likely that the reviewer did not understand parts of the paper

**Scope:**

3: The work is somewhat relevant to the Web and to the track, and is of narrow interest to a sub-community

---

### Official Review · Reviewer_LQpF · 2024-12-03

**Novelty:** 7
**Technical Quality:** 6

**Review:**

#### **1. Quality**

This paper systematically studies the generalization performance of Hypergraph Neural Networks (HyperGNNs) using the PAC-Bayes framework. From theoretical analysis to experimental validation, the authors provide generalization bounds for several representative models (e.g., UniGCN, AllDeepSets, M-IGN, and T-MPHN) and reveal the relationships between key model attributes (e.g., parameter spectral norms and hyperedge sizes) and generalization capabilities. The research is of high quality, with rigorous mathematical derivations and conclusions that carry both theoretical and practical significance. Additionally, the experimental section thoroughly validates the theoretical results, covering both real-world and synthetic datasets with detailed and well-designed experimental setups.


#### **2. Clarity**

The paper is well-structured, with the theoretical sections unfolding model by model in a progressively logical manner. The experimental sections validate the theoretical predictions across diverse datasets. Symbols and terminology are clearly defined, and the mathematical expressions are precise. The experimental results are presented clearly using graphs and quantitative metrics. However, the discussion on deviations between theoretical bounds and empirical errors in specific cases (e.g., T-MPHN on certain datasets) is insufficient, which might affect readers’ confidence in the theoretical results.


#### **3. Originality**

This paper demonstrates high originality as it is the first to systematically analyze the generalization performance of HyperGNNs. By leveraging the PAC-Bayes framework, the authors propose a perturbation decomposition method tailored to HyperGNNs, extending existing generalization analysis techniques to models with complex structures. The analysis of four representative models (UniGCN, AllDeepSets, M-IGN, and T-MPHN) comprehensively covers mainstream HyperGNN architectures and provides a detailed investigation of how hypergraph structural properties (e.g., hyperedge sizes and node degrees) impact model generalization. Compared to previous works that focused only on empirical performance, this paper offers a theoretical framework for understanding HyperGNNs, opening new avenues for research in this area.


#### **4. Significance**

This paper holds significant value both academically and practically. From an academic perspective, the theoretical framework fills a critical gap in the understanding of HyperGNN generalization performance, systematically unveiling the influence of model parameters and hypergraph structural properties on generalization capabilities. This not only helps in understanding model behavior but also provides theoretical guidance for designing more efficient and stable HyperGNNs. From a practical perspective, HyperGNNs have broad applications in social network analysis, recommendation systems, and knowledge graphs. The findings of this paper offer valuable insights into model selection and optimization for these tasks. The consistency between theoretical results and experiments on various datasets further demonstrates its real-world applicability.


#### **5. Pros**

- The perturbation decomposition method combined with the PAC-Bayes framework provides the first systematic analysis of HyperGNN generalization, with conclusions that are theoretically robust.
- The paper covers four main HyperGNN architectures comprehensively, ensuring a wide-ranging exploration of the topic.
- The combination of real-world and synthetic datasets ensures a well-rounded validation of the theoretical results, with experiments that are well-designed and effective.


#### **6. Cons**

- Attention-based models (e.g., UniGAT) are not covered in this study. Although the challenges of analyzing such models are acknowledged, no attempts are made to extend the framework, limiting its comprehensiveness.
- In some experiments, the deviations between theoretical bounds and empirical errors (e.g., T-MPHN on certain datasets) are not thoroughly discussed, which might reduce readers’ confidence in the theoretical results.

**Questions:**

- This study theoretically explains the determining factors of the generalization performance of HyperGNNs, but the experiments have not explicitly clarified the practical implications of these theoretical results for model optimization.

- In the experimental section, while the overall consistency between the theoretical bounds and empirical errors is demonstrated, certain models (e.g., T-MPHN) exhibit deviations on specific datasets that are not sufficiently explained. For example, do these deviations stem from the normalization process of the model or the structural properties of the hypergraphs in the dataset? Could the paper provide a more detailed analysis of these deviations to enhance the persuasiveness of the results?

**Reviewer Confidence:**

3: The reviewer is confident but not certain that the evaluation is correct

**Scope:**

4: The work is relevant to the Web and to the track, and is of broad interest to the community